# HiTNet: Hippocampal-Thalamic Inspired Dual-Stream Network for Multimodal Sentiment Analysis under Missing Data

## Abstract

Multimodal sentiment analysis faces significant challenges under conditions of missing data, where simultaneous random frame-level missingness across all modalities results in fragmented emotional cues and heterogeneous data quality. Existing methods predominantly rely on cross-modal consistency for completion but often neglect residual intra-modal information and lack in assessing cross-modal reliability, leading to redundancy that degrades performance. Human cognitive systems exhibit remarkable robustness to incomplete perceptual input through two functional mechanisms: hippocampal memory systems that reconstruct missing content via pattern completion from stored semantic traces, and thalamic perceptual regulation that dynamically integrates multisensory inputs while filtering unreliable information. Inspired by the brain functions, we propose a Hippocampal-Thalamic dual-stream Network (HiTNet). Hippocampal-inspired intra-modal enhancement stream employs semantic memory modules with dynamic retrieval and sparse activation networks to mine modality-specific information and reconstruct missing features. Thalamic-inspired inter-modal regulation stream implements confidence perception and adaptive cross-modal completion modules to dynamically integrate high-quality cross-modal information while suppressing redundant interference. Comprehensive experiments on MOSI, MOSEI, and SIMS demonstrate that HiTNet achieves superior performance with 1.5%–2.0% average accuracy improvements over state-of-the-art methods across all missing rates and maintains 72.20% accuracy under extreme 90% missing conditions on MOSEI, validating the effectiveness of brain function-inspired design for robust multimodal sentiment analysis even under extreme missing data scenarios. Our code is available at: `https://anonymous.4open.science/r/HiTNet-8798/`.

## 1 Introduction

Multimodal sentiment analysis (MSA) provides a comprehensive analytical framework for understanding complex human emotions by collaboratively modeling multi-source data such as language, audio, and vision (Du et al., 2024; Georgiou et al., 2024; Wu et al., 2025). Nevertheless, multimodal data in real-world applications frequently suffer from missingness, which can be generally divided into two categories: modality-level missingness where an entire modality is absent (Sun et al., 2024a; Zhang et al., 2024b; Kim & Kim, 2025), and frame-level missingness, where partial content within a modality is lost due to noise, hardware malfunction, or transmission issues (Zhang et al., 2024a; Li et al., 2024a), as shown in Figure 1. Compared with modality-level missingness, random frame-level missingness across all modalities is more complex, causing fragmented emotional cues and discrepancies in data quality, making sentiment analysis more challenging.

Recent studies have made notable progress in addressing frame-level missingness data in multimodal sentiment analysis, primarily relying on cross-modal consistency for completion (Li et al., 2024b; Zhang et al., 2024a; Li et al., 2024a). For instance, UMDF (Li et al., 2024a) completes missing modalities by enforcing distributional consistency between heterogeneous modality missing samples. However, they still fail to exploit the residual semantic cues within each modality and overlook modality-specific characteristics. Moreover, different modalities often vary in confidence,

and treating cross-modal information equally makes it difficult to filter high-quality cues, which may introduce redundancy and lead to degraded sentiment analysis performance (Mai et al., 2024).

Unlike traditional artificial neural networks, humans have the capability for stable emotional understanding even when perceptual information is incomplete (Tanaka et al., 2014), which benefits from memory retrieval in the hippocampus and perceptual regulation in the thalamus under neuroscientific studies (Costa et al., 2025; Yang et al., 2025). Specifically, the hippocampus retrieves residual modality-specific information from memory via pattern completion to recover missing content, and

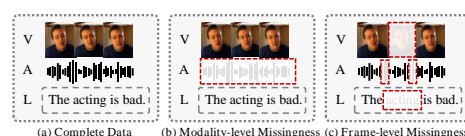

(a) Complete Data  (b) Modality-level Missingness  (c) Frame-level Missingness

Figure 1: Illustration of three data conditions in multimodal sentiment analysis.

further selectively activates task-relevant pathways, enabling goal-directed processing of critical information (Tanaka et al., 2014). Meanwhile, the thalamus plays a critical role in multimodal processing by receiving multisensory inputs, dynamically integrating information, gating behavior, and suppressing redundancy (Wolff et al., 2021; Yang et al., 2025).

Motivated by such brain function mechanisms, we propose the Hippocampal-Thalamic dual-stream Network (HiTNet). On one hand, the intra-modal enhancement stream simulates the completion process based on memory retrieval in the hippocampus. This memory retrieval process has been abstracted into classic computational models, specifically Sparse Distributed Memory (SDM) (Kanerva, 1988) and Hopfield Networks (Hopfield, 1982). These foundational works mathematically characterize memory as a high-dimensional associative process, capable of robustly reconstructing complete patterns from partial cues via content-based addressing. Drawing on the principles of these models, we have developed a robust method for addressing frame-level data missingness. First, a key-value storage-based semantic memory module is introduced to retrieve remaining modality-specific semantic memories through similarity matching. Crucially, an adaptive gating mechanism is designed to suppress irrelevant memories retrieved by corrupted queries. The results are then input into a sparse activation network, enabling fine-grained modeling of modality-specific semantics. On the other hand, the inter-modal regulation stream simulates the perceptual regulation process of the thalamus. A confidence-perception module is proposed to estimate the reliability of each modality and guide the cross-modal completion module to integrate high-quality information for frame-level missingness and suppress redundant interference. Furthermore, a hierarchical fusion module is employed to integrate the dual-stream output, enhancing the overall representational capacity of the model. Our contributions can be summarized as follows:

- We propose a brain function-inspired approach to overcome the performance bottleneck of existing methods, under the significant challenge posed by random frame-level missingness in sentiment analysis.

- We innovatively model hippocampal and thalamic functional mechanisms as the dual-stream network (HiTNet): the hippocampal-inspired intra-modal enhancement stream reconstructs modality-specific semantics, while the thalamic-inspired inter-modal regulation stream dynamically integrates cross-modal information and suppresses redundancy.

- The proposed HiTNet improves average accuracy by 1.5%-2.0% over state-of-the-art methods across all missing rates, maintaining high accuracy even with 90% missing data, demonstrating effectiveness of the HiTNet under extreme missing conditions.

## 2 RELATED WORK

**Multimodal Sentiment Analysis.** Conventional MSA approaches primarily focus on sophisticated fusion strategies under the assumption of complete data. To capture distinct modal properties, MISA (Hazarika et al., 2020) factorizes data into modality-invariant and specific subspaces. Information-theoretic approaches like MMIM (Han et al., 2021) maximize Mutual Information to maintain task-relevant cues, while Self-MM (Yu et al., 2021) employs self-supervised learning to refine unimodal representations. Recent attention-based models, such as CENET (Wang et al., 2023b) and TETFN (Wang et al., 2023a), utilize cross-modal attention to enhance non-verbal features with textual semantics. Furthermore, ALMT (Zhang et al., 2023) introduces an adaptive language-guided

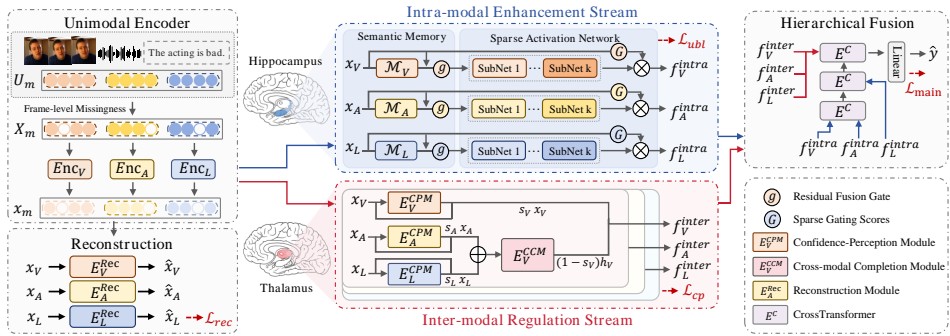

Figure 2: The overall of HiTNet. The modality information is first encoded under frame-level missing situation, then the brain function inspired dual streams enhance and regulate intra- and inter-modal features, and the hierarchically fused representations are finally used for sentiment analysis.

mechanism to suppress noise. While these methods achieve state-of-the-art performance on complete benchmarks, their reliance on full modality availability often leads to performance degradation in real-world missing scenarios.

**Approaches for Incomplete Multimodal Learning.** To address data incompleteness, recent studies have explored various robust learning paradigms. Reconstruction-based methods, such as TFR-Net (Yuan et al., 2021), employ Transformers to explicitly generate missing features. Targeting robustness against noise, LNLN (Zhang et al., 2024a) proposes a language-dominated learning framework to preserve text integrity, while P-RMF (Zhu et al., 2025) learns a robust proxy modality from latent Gaussian distributions. Recently, powerful generative models like Masked Autoencoders (MAE) (Tong et al., 2022; Dong et al., 2023) and Diffusion Models (Alcaraz & Strodthoff, 2023) have also been explored for imputation. These methods typically rely on global distribution priors to restore physical data integrity. However, they lack the capability to explicitly assess the intrinsic quality or information value of residual signals. Consequently, strictly generative approaches face the risk of semantic shift and propagating input noise into the completion results. Furthermore, while Key-Value memory networks (Lang et al., 2025; Pipoli et al., 2025) have been employed for general data completion tasks, these methods typically rely on direct lookup and feature replacement. However, an input query corrupted by frame missingness risks retrieving irrelevant memory, thereby leading to imputation errors. In contrast, our semantic memory module utilizes a residual gating mechanism to filter retrieval noise, providing a robust semantic prior for the subsequent sparse activation network.

**Brain-Inspired Approaches.** Inspired by neuroscience, recent studies have begun exploring brain-inspired architectures for robust multimodal perception (Wang et al., 2025b;a; Sun et al., 2024b; Wang et al., 2023c). For example, HIMM (Sun et al., 2024b) models high-order semantic relationships by mimicking the brain's neural connectivity structure, facilitating multimodal and multitask understanding. Current brain-inspired research primarily focuses on complete data scenarios. In this work, we propose a brain-inspired dual-stream completion network for missing data completion.

## 3 METHODOLOGY

### 3.1 PROBLEM DEFINITION

Given a video clip containing three modalities, visual (V), audio (A) and language (L), we denote the complete multimodal sequence as $U_m \in \mathbb{R}^{T_m \times D_m}$, where $T_m$ indicates the sequence length and $D_m$ refers to the dimensionality of each modal vector, and $m \in \{V, A, L\}$. Due to camera occlusion or channel noise, intra-modality partial data loss often occurs. For example, missing facial frames in a video or lost segments in audio. In this study, the missing data $X_m \in \mathbb{R}^{T_m \times D_m}$ is modeled by introducing randomly missing features at the frame level within modality sequences.

## 3.2 OVERALL FRAMEWORK

The overall architecture of HiTNet are depicted in Figure 2. First, the missing data sample $X_m$ is encoded by the modality encoder to obtain the unified feature $x_m$. Subsequently, $x_m$ is separately fed into an intra-modal enhancement stream inspired by hippocampal memory retrieval to obtain the intra-modal completion feature $f_m^{intra}$, and into an inter-modal regulation stream inspired by thalamic perceptual regulation to obtain the inter-modal completion feature $f_m^{inter}$. The two completion features are then hierarchically fused for sentiment prediction. During training, missing reconstruction is employed as an auxiliary supervision to further enhance model performance.

## 3.3 UNIMODAL ENCODER

We construct modal encoders $Enc_m(\cdot)$ for visual, audio, and language separately to extract and unify the modality representations. Specifically, the visual and audio modality encoders are composed of linearly transformed and Transformer encoder layers. The language modality is first extracted from the semantic representation by the pre-trained language model BERT, which is then fed into the Transformer layer to capture richer contextual dependencies. This process is expressed as:

$$x_m = Enc_m(X_m), \tag{1}$$

where $x_m$ is the final dimensionally consistent feature which is fed into the downstream modules.

## 3.4 INTRA-MODAL ENHANCEMENT STREAM

Existing methods primarily rely on cross-modal consistency to complete missing data, which limits their ability to capture modality-specific characteristics. In frame-level missing scenarios, however, the remaining features often retain rich modality-specific information, which the human brain can effectively leverage. In particular, memory retrieval in the hippocampus compensates for incomplete inputs by drawing on prior experiences to recover the missing modality-specific content (Tanaka et al., 2014). Inspired by this mechanism, we propose semantic memory modules (SMM) and sparse activation networks (SAN) to exploit intra-modal remaining information for self-completion.

**Semantic Memory Module.** For each modality $m \in \{V, A, L\}$, we introduce a learnable key-value memory matrix $\mathcal{M}_m = \{(\mathbf{k}_i^m, \mathbf{v}_i^m)\}_{i=1}^N$, where $\mathbf{k}_i^m \in \mathbb{R}^{D_m}$ and $\mathbf{v}_i^m \in \mathbb{R}^{D_m}$ denote the memory key and semantic content of the $i$-th memory unit, respectively, and $N$ is the total number of memory units. During training, the input features $x_m$ are mean-pooled and normalized to obtain the memory key, while the pooled representation is passed through a linear projection to generate the memory value. The new key–value pair replaces the least frequently accessed memory unit, enabling dynamic maintenance of the semantic memory.

For memory retrieval, we use the mean-pooled representation of the modality input $x_m$ as the query vector to retrieve the most relevant semantic memory unit based on cosine similarity:

$$i^* = \arg\max_i \frac{\langle \text{MeanPool}(x_m), \mathbf{k}_i^m \rangle}{\|\text{MeanPool}(x_m)\| \cdot \|\mathbf{k}_i^m\|}, \tag{2}$$

where $\mathbf{k}_i^m$ denotes the i-th memory key for modality m. The retrieved value $\mathbf{v}_{i^*}^m$ serves as the memory output, representing the most relevant prior semantic information for the current input. To adaptively integrate the retrieved memory with the original input, we employ a residual gating mechanism:

$$g_m = \sigma(W_r \cdot Concat(x_m, \mathbf{v}_{i^*}^m)), \quad \tilde{x}_m = x_m + g_m \odot \mathbf{v}_{i^*}^m, \tag{3}$$

where $W_r \in \mathbb{R}^{(2D_m) \times 1}$ is a learnable projection matrix, $\sigma(\cdot)$ is the sigmoid activation function, and $\tilde{x}_m$ is the memory-enhanced representation, $\odot$ indicates element-wise multiplication. This residual fusion facilitates integrating prior semantics without discarding original modality information.

**Sparse Activation Network.** To further model intra-modal diversity, we adopt a sparse activation network with a set of sub-networks $\{S_1, S_2, ..., S_n\}$. To ensure a consistent focus on key semantic information, we employ a sparse gating mechanism to select the top-$k$ sub-networks for each input:

$$G(x_m) = \text{Softmax}(\text{TopK}(x_m W_G, k)), \tag{4}$$

where $W_G \in \mathbb{R}^{D_m \times n}$ denotes the learnable parameters, and $\text{TopK}(\cdot, k)$ denotes the function to choose the top-$k$ highest values given the input $x_m$.

Each selected sub-network processes the memory-enhanced representation $\tilde{x}_m$, and their outputs are aggregated based on the gate scores to form the intra-modal completion feature:

$$f_m^{\text{intra}} = \sum_{j=1}^{s} G(x_m)_j \cdot S_j(\tilde{x}_m), \tag{5}$$

where $G(x_m)_j$ is the gate score assigned to the $j$-th sub-network and $S_j(\cdot)$ is composed of two MLP layers and a GELU activation function. To avoid over-reliance on a few sub-networks and promote balanced usage, we introduce a utilization balance loss:

$$\mathcal{L}_{ubl} = \sum_{m \in \{V,A,L\}} \left( \text{CV}^2(I_m) + \text{CV}^2(L_m) \right), \tag{6}$$

where $\text{CV}(\cdot)$ denotes the coefficient of variation, which measures the relative dispersion of sub-network usage. $I_m$ represents the total gating weights assigned to each sub-network for modality $m$, while $L_m$ counts the number of times each sub-network is selected.

### 3.5 Inter-modal Regulation Stream

While cross-modal information can help fill missing data, ignoring confidence differences across modalities may introduce redundancy. Biological studies have shown that the thalamus plays a key role in coordinating information integration and regulating behavioral gating in multimodal cognition (Wolff et al., 2021; Yang et al., 2025). Inspired by the mechanism of thalamic, we designed a confidence-perception module and a cross-modal completion module to assess modality confidence and guide adaptive cross-modal compensation.

**Confidence-Perception Module.** To assess the confidence of each modality, we introduce a confidence-perception module (CPM) for each modality. Each CPM consists of two Transformer encoder layers and a lightweight MLP classifier with a sigmoid activation to predict the confidence score. The confidence score $s_m$ for input $x_m$ is computed as:

$$s_m = E_m^{CPM}(x_m), \tag{7}$$

where $s_m$ quantifies the intrinsic completeness and confidence of modality $m \in \{V, A, L\}$, and serves as a crucial weight factor for subsequent cross-modal completion. To guide the model in learning meaningful confidence scores, we use L2 loss between the predicted score $s_m$ and a soft ground-truth completeness label $\hat{s}_m = 1 - r_m$, where $r_m$ is the missing ratio of modality $m$. The confidence perception loss is defined as:

$$\mathcal{L}_{cp} = \frac{1}{N_b} \sum_{k=1}^{N_b} \left\| s_m^k - \hat{s}_m^k \right\|_2^2, \tag{8}$$

where $N_b$ denotes the number of samples in a training batch.

**Cross-modal Completion Module.** To enable effective cross-modal completion, we introduce a Cross-Modal Completion Module (CCM) consisting of two Transformer encoder layers. Confidence scores $s_m$ are used as attention weights to guide the completion. Given a target modality $m \in \{V, A, L\}$, the CCM first constructs an intermediate representation $h_m \in \mathbb{R}^{T_m \times D_m}$ by aggregating cross-modal inputs from the remaining two modalities $m^{'}, m^{''}$.

$$h_m = E^{CCM}\left( Concat\left( h_m^0, s_{m'} x_{m'}, s_{m''} x_{m''} \right) \right), \tag{9}$$

where $h_m^0 \in \mathbb{R}^{T_m \times D_m}$ denotes a modality-specific learnable prompt sequence, shared across the batch, that serves as a semantic anchor for the target modality. Then, the original feature $x_m$ and the cross-modal complementary feature $h_m$ are dynamically integrated to form the final inter-modal completion feature $f_m^{inter}$:

$$f_m^{inter} = s_m \cdot x_m + (1 - s_m) \cdot h_m, \tag{10}$$

where $s_m$ indicates the confidence score, used to balance the contribution between intrinsic and complementary features.

## 3.6 FUSION AND RECONSTRUCTION

**Hierarchical Fusion and Prediction.** To enhance cross-modal interactions, we adopt a hierarchical fusion strategy based on a CrossTransformer ($E^C$), progressively integrating both intra-modal and inter-modal representations. The intra-modal completion features $f_m^{\text{intra}}$, which retain rich modality-specific structural and fine-grained information, are progressively fused via $E^C$. Placing the language modality last allows it to guide the final semantic integration, given its dominant role in conveying affective intent:

$$f^{\text{intra}} = E^C(E^C(f_V^{\text{intra}}, f_A^{\text{intra}}), f_L^{\text{intra}}), \tag{11}$$

where $f^{\text{intra}} \in \mathbb{R}^{T \times D}$ represents the fused intra-modal completion representation.

In contrast, inter-modal completion feature $f_m^{\text{inter}}$ already encodes complementary cross-modal cues, so we directly sum them to obtain $f^{\text{inter}}$. The $f^{\text{intra}}$ and $f^{\text{inter}}$ are integrated through a final $E^C$ to yield the comprehensive fused feature:

$$f^{\text{inter}} = f_V^{\text{inter}} + f_A^{\text{inter}} + f_L^{\text{inter}}, \quad F = E^C(f^{\text{intra}}, f^{\text{inter}}), \tag{12}$$

where $F \in \mathbb{R}^{T \times D}$ serves as the final multimodal embedding for sentiment regression.

Finally, the model generates sentiment predictions $\hat{y}$ by performing mean-pooling on $F$ and passing it through a linear regression head $\hat{y} = \text{Linear}(\text{MeanPool}(F))$, To optimize the model, we apply a standard L2 loss between the ground-truth sentiment label and the predicted score:

$$\mathcal{L}_{\text{main}} = \frac{1}{N_b} \sum_{k=1}^{N_b} \|y^k - \hat{y}^k\|_2^2, \tag{13}$$

where $y^k$ and $\hat{y}^k$ represent the ground-truth label and predicted score for the $k$-th sample, respectively, and $N_b$ denotes the number of samples in a training batch.

**Missing Information Reconstruction.** As noted in the LNLN (Zhang et al., 2024a), reconstructing missing information can significantly enhance the performance of regression tasks. Building on this finding, we introduce the missing information reconstruction module $E^{Rec}$, designed to reconstruct the missing features of each modality. The $E^{Rec}$ module consists of two Transformer encoder layers, which are specifically designed to restore the missing information in the input features. The reconstruction process is expressed as $\hat{x}_m = E_m^{Rec}(x_m)$. Where $\hat{x}_m$ is the reconstructed feature for modality $m$, and $x_m$ is the input feature with missing information. To optimize the performance of the reconstruction, we employ an L2 loss function between the encoded original feature $u_m = Enc_m(U_m)$ and the reconstructed feature $\hat{x}_m$:

$$\mathcal{L}_{\text{rec}} = \frac{1}{N_b} \sum_{k=1}^{N_b} \|u_m^k - \hat{x}_m^k\|_2^2, \tag{14}$$

where $u_m^k$ and $\hat{x}_m^k$ denote the encoded original and reconstructed features for the $k$-th sample.

## 3.7 OPTIMIZATION OBJECTIVES

In summary, the model is optimized via a weighted combination of four losses: the sentiment prediction loss $\mathcal{L}_{\text{main}}$, utilization balance loss $\mathcal{L}_{\text{ubl}}$, the confidence perception loss $\mathcal{L}_{cp}$, and the reconstruction loss $\mathcal{L}_{\text{rec}}$.

$$\mathcal{L}_{\text{total}} = \mathcal{L}_{\text{main}} + \alpha \cdot \mathcal{L}_{ubl} + \beta \cdot \mathcal{L}_{cp} + \gamma \cdot \mathcal{L}_{\text{rec}}, \tag{15}$$

where $\alpha$, $\beta$, and $\gamma$ are hyperparameters that balance the contributions of the corresponding losses.

## 4 EXPERIMENTS

### 4.1 DATASETS

To comprehensively assess the proposed HiTNet, we conduct experiments on three benchmark datasets for sentiment analysis: The **MOSI** dataset (Zadeh et al., 2016) consists of 2,199 opinion

video clips from 93 YouTube movie reviews, each annotated with a sentiment score ranging from -3 (strongly negative) to 3 (strongly positive), with aligned visual, audio, and text modalities. It is split into 1,284 training, 229 validation, and 686 testing samples. The **MOSEI** dataset (Bagher Zadeh et al., 2018) contains 22,856 video clips with annotations in -3 (strongly negative) to 3 (strongly positive). It is divided into 16,326 training, 1,871 validation, and 4,659 testing samples, and includes aligned visual, acoustic, and textual modalities. The **SIMS** dataset (Yu et al., 2020) is a Chinese multimodal sentiment benchmark comprising 2,281 video clips, annotated from -1 (negative) to 1 (positive). It includes 1,368 training, 456 validation, and 457 testing samples.

### 4.2 Missingness Settings and Evaluation Metrics

**Missingness Settings.** We follow LNLN (Zhang et al., 2024a) to simulate frame-level missingness for each sample. For the visual and audio modalities, missing segments are filled using zero vectors. In the language modality, missing segments are replaced with [UNK], representing unknown tokens in BERT (Devlin et al., 2019). During training, for each modality, a random missing rate is sampled independently for each sample, and then a Bernoulli process is applied across valid positions to determine which frames are masked. Additionally, to avoid overfitting to missing data, half of the samples for each modality are randomly set to have zero missing rate. During testing, missing rates are set from 0 to 0.9 with a step size of 0.1, and the missing positions for each modality are independently and randomly sampled.

**Evaluation Metrics.** For MOSI and MOSEI, we report the seven-class accuracy (Acc-7), five-class accuracy (Acc-5), binary classification accuracy (Acc-2), and the F1 score. For Acc-2 and F1, we adopt two calculation methods: negative/positive (left-side value of /) and negative/non-negative (right-side value of /). We also report the Mean Absolute Error (MAE) and Pearson correlation coefficient (Corr). For SIMS, the metrics include Acc-5, Acc-2, three-class accuracy (Acc-3), F1 score, MAE, and Corr. Except for MAE, higher values indicate better performance.

### 4.3 Implementation Details

Our model is implemented with the PyTorch toolbox and trained on NVIDIA L40S GPUs. The AdamW optimizer is adopted for parameter optimization. A fixed number of 200 training epochs is used across all datasets. The $E^{CCM}$ is configured with 2 layers, 8 heads, hidden dimension 128, input length 24, and token length 8. The $E^{CPM}$ shares the same head and dimension settings, but with shorter input and token lengths (8 and 1). The $E^C$ has 4 layers, 8 heads, and a hidden dimension of 128. In the sparse activation network, we set the total number of sub-networks to $n = 5$ and the number of activated sub-networks to $k = 3$. To ensure robustness and reliability, we repeat the experiment using three different random seeds and report the average results. The number of memory units is set as $N = 64$. For MOSI, MOSEI, and SIMS, the input vector length $T$ is 8, the vector dimension $d$ is 128, the batch size is 64, and the initial learning rate is $1 \times 10^{-4}$. The hyperparameters $\alpha$, $\beta$, and $\gamma$ vary across datasets: 10, 0.5, 0.1 for MOSI, 1.5, 0.9, 9.0 for MOSEI, and 10, 0.9, 0.1 for SIMS. All datasets employ warm up, cosine annealing, and early stopping strategies during training. In addition, the verification experiments of different values of the loss function weights are presented in Appendix B.1, and the sensitivity analysis of sparse activation parameters are presented in Appendix B.2.

### 4.4 Comparison with State-of-the-art Methods

We compare HiTNet against several strong baselines, including MISA (Hazarika et al., 2020), Self-MM (Yu et al., 2021), MMIM (Han et al., 2021), CENET (Wang et al., 2023b), TETFN (Wang et al., 2023a), TFR-Net, ALMT (Zhang et al., 2023), LNLN (Zhang et al., 2024a), and P-RMF (Zhu et al., 2025). The results of these baselines are reported as in LNLN, ensuring consistency in evaluation settings across all compared methods. As shown in Tables 1 and 2 (averaged across all missing rates), HiTNet consistently achieves state-of-the-art performance. It outperforms all existing methods across all metrics on MOSI and MOSEI, with Acc-2 and F1 improvements of 1.31% and 1.41% on MOSI, along with a substantial 2.56% gain in Acc-7 on MOSEI. On SIMS, HiTNet delivers state-of-the-art or highly competitive results, achieving a remarkable 4.53% improvement in Acc-3. This performance gain is attributed to our brain-inspired dual-stream mechanism. In addition, Figure 3 shows the trends of accuracy and MAE for all models as the missing rate increases. HiTNet

Table 1: Comparison of model performance on MOSI and MOSEI. Note: The lower MAE corresponds to superior results.

| Method | MOSI | | | | | | MOSEI | | | | | |
|--------|------|------|------|------|------|------|--------|------|------|------|------|------|
| | Acc-7 | Acc-5 | Acc-2 | F1 | MAE | Corr | Acc-7 | Acc-5 | Acc-2 | F1 | MAE | Corr |
| MISA | 29.85 | 33.08 | 71.49 / 70.33 | 71.28 / 70.00 | 1.085 | 0.524 | 40.84 | 39.39 | 71.27 / 75.82 | 63.85 / 68.73 | 0.780 | 0.503 |
| Self-MM | 29.55 | 34.67 | 70.51 / 69.26 | 66.60 / 67.54 | 1.070 | 0.512 | 44.70 | 45.38 | 73.89 / 77.42 | 68.92 / 72.31 | 0.695 | 0.498 |
| MMIM | 31.30 | 33.77 | 69.14 / 67.06 | 66.65 / 64.04 | 1.077 | 0.507 | 40.75 | 41.74 | 73.32 / 75.89 | 68.72 / 70.32 | 0.739 | 0.489 |
| CENET | 30.38 | 37.25 | 71.46 / 67.73 | 68.41 / 64.85 | 1.080 | 0.504 | 47.18 | 47.83 | 74.67 / 77.34 | 70.68 / 74.08 | 0.685 | 0.535 |
| TETFN | 30.30 | 34.34 | 69.76 / 67.68 | 65.69 / 63.29 | 1.087 | 0.507 | 30.30 | 47.70 | 69.76 / 67.68 | 65.69 / 63.29 | 1.087 | 0.508 |
| TFR-Net | 29.54 | 34.67 | 68.15 / 66.35 | 61.73 / 60.06 | 1.200 | 0.459 | 46.83 | 34.67 | 73.62 / 77.23 | 68.80 / 71.99 | 0.697 | 0.489 |
| ALMT | 30.30 | 33.42 | 70.40 / 68.39 | 72.57 / 71.80 | 1.083 | 0.498 | 40.92 | 41.64 | 76.64 / 77.54 | 77.14 / 78.03 | 0.674 | 0.481 |
| LNLN | 34.26 | 38.27 | 72.55 / 70.94 | 72.73 / 71.25 | 1.046 | 0.527 | 45.42 | 46.17 | 76.30 / 78.19 | 77.77 / 79.95 | 0.692 | 0.530 |
| P-RMF | 34.19 | 38.50 | 72.81 / 71.53 | 72.93 / 71.69 | **1.038** | 0.525 | 44.63 | 45.87 | 78.14 / 78.83 | **79.33** / 80.39 | **0.658** | 0.589 |
| **HiTNet** | **35.26** | **39.22** | **74.12 / 72.66** | **74.53 / 73.10** | 1.043 | **0.539** | **47.19** | **47.98** | **78.29 / 79.28** | 78.84 / **81.46** | 0.665 | **0.591** |

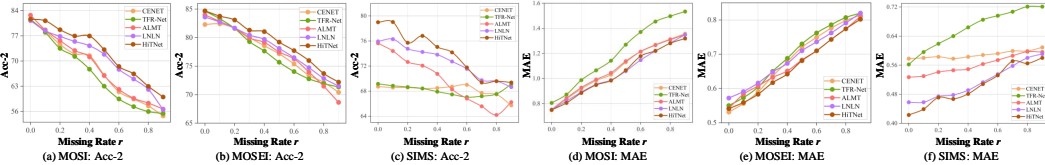

Figure 3: Visualization of performance under different missing rate (Acc-2↑ and MAE↓).

maintains high accuracy and low MAE even at high missing rates, significantly outperforming other baselines, further demonstrating the effectiveness and robustness of the dual-stream mechanism. Detailed performance in different missing rates is provided in Appendix B.3.

## 4.5 ABLATION STUDIES

**Contributions of Different Components.** To evaluate the contribution of each component in HiT-Net, we conduct a series of ablation studies by removing key modules and observing the performance changes, as shown in Table 3. The results show that removing the semantic memory module (w/o SMM) impairs the ability of the model to effectively retrieve and utilize residual semantic information stored in memory. Removing the confidence-perception module (w/o CPM) impairs dynamic assessment of modality reliability, leading to a substantial performance drop. Furthermore, removing the intra-modal enhancement stream (w/o Intra) results in noticeable declines across multiple metrics, demonstrating its positive role in recovering missing semantics and strengthening intra-modal representations. Removing the inter-modal regulation stream

Table 2: Comparison of model performance on SIMS. Note: The lower MAE corresponds to superior results.

| Method | Acc-5 | Acc-3 | Acc-2 | F1 | MAE | Corr |
|--------|-------|-------|-------|------|------|------|
| MISA | 31.53 | 56.87 | 72.71 | 66.30 | 0.539 | 0.348 |
| Self-MM | 32.28 | 56.75 | 72.81 | 68.43 | 0.508 | 0.376 |
| MMIM | 31.81 | 52.76 | 69.86 | 66.21 | 0.544 | 0.339 |
| CENET | 22.29 | 53.17 | 68.13 | 57.90 | 0.589 | 0.107 |
| TETFN | 33.42 | 56.91 | 73.58 | 68.67 | 0.505 | 0.387 |
| TFR-Net | 26.52 | 52.89 | 68.13 | 58.70 | 0.661 | 0.169 |
| ALMT | 20.00 | 45.36 | 69.66 | 72.76 | 0.561 | 0.364 |
| LNLN | 34.64 | 57.14 | 72.73 | **79.43** | 0.514 | 0.397 |
| P-RMF | 34.83 | 54.75 | 73.64 | 74.65 | **0.500** | **0.414** |
| **HiTNet** | **35.62** | **59.28** | **73.99** | 77.33 | 0.504 | 0.389 |

(w/o Inter) causes even greater performance degradation, emphasizing its central role in cross-modal completion and integration. In summary, each component plays an indispensable role in handling complex frame-level missing data for sentiment analysis, validating the effectiveness of the proposed dual-stream architecture.

**Contributions of Different Losses.** We assess the contribution of each loss component to HiT-Net's performance, as shown in Table 3. The experimental results reveal that excluding any of these losses leads to a noticeable performance degradation. Removing the utilization balance loss $\mathcal{L}_{ubl}$ disrupts the activation balance within the sparse activation network, resulting in over-reliance on certain computational paths and reduced diversity in learned representations. Omitting the confidence perception loss $\mathcal{L}_{cp}$ significantly impairs the ability of the model to dynamically perceive and weight the confidence of each modality, which is essential for robust multimodal fusion under partial modality data missing conditions. Furthermore, the absence of the reconstruction loss $\mathcal{L}_{rec}$ weakens

Table 3: Effects of different components and losses. Lower MAE indicates superior results.

| Method | MOSI | | | | | | SIMS | | | | | |
|---|---|---|---|---|---|---|---|---|---|---|---|---|
| | Acc-7 | Acc-5 | Acc-2 | F1 | MAE | Corr | Acc-5 | Acc-3 | Acc-2 | F1 | MAE | Corr |
| w/o SMM | 34.74 | 38.63 | 73.61 / 72.27 | 74.01 / 72.40 | 1.043 | 0.532 | 34.09 | 58.39 | 73.08 | 76.32 | 0.509 | 0.381 |
| w/o CPM | 34.87 | 38.57 | 73.72 / 72.08 | 74.48 / 72.51 | 1.044 | 0.531 | 34.68 | 59.19 | 73.90 | 76.59 | 0.508 | 0.385 |
| w/o Intra | 34.91 | 38.88 | 73.63 / 72.23 | 73.79 / 72.41 | 1.045 | 0.527 | 34.33 | 58.64 | 73.87 | 76.49 | 0.513 | 0.379 |
| w/o Inter | 33.98 | 37.75 | 73.25 / 72.06 | 73.57 / 72.46 | 1.062 | 0.499 | 33.52 | 58.17 | 73.04 | 76.83 | 0.525 | 0.348 |
| w/o $\mathcal{L}_{ubl}$ | **35.41** | **39.40** | 73.64 / 72.26 | 73.92 / 72.33 | 1.044 | 0.537 | 35.19 | 59.15 | 73.81 | 78.13 | 0.513 | 0.384 |
| w/o $\mathcal{L}_{cp}$ | 33.87 | 37.52 | 72.90 / 71.59 | 73.21 / 71.61 | 1.068 | 0.527 | 34.44 | 59.23 | 73.42 | 77.57 | 0.508 | 0.387 |
| w/o $\mathcal{L}_{rec}$ | 34.55 | 38.45 | 73.32 / 71.95 | 73.62 / 71.99 | 1.047 | 0.532 | 33.85 | 58.97 | 73.85 | **79.03** | 0.510 | 0.373 |
| **HiTNet** | 35.26 | 39.22 | **74.12 / 72.66** | **74.53 / 73.10** | **1.043** | **0.539** | **35.62** | **59.28** | **73.99** | 77.33 | **0.504** | **0.389** |

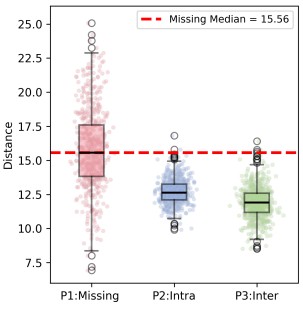

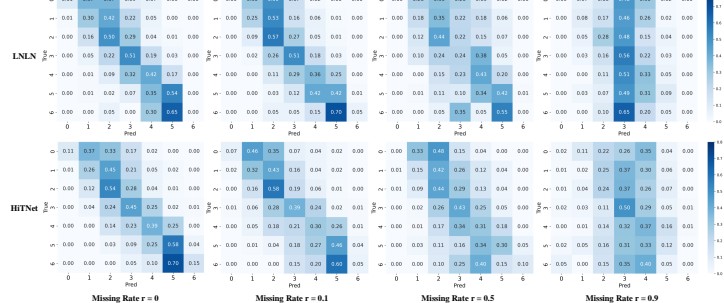

Figure 4: Boxplots showing Euclidean distances among missing, intra-, inter-modal, and complete features.

Figure 5: Confusion matrices of HiTNet and the baseline model LNLN on the MOSI dataset. Note: 0-6 denote strongly negative, negative, weakly negative, neutral, weakly positive, positive, and strongly positive, respectively.

the capacity of the model to reconstruct missing semantic content, thereby reducing the completeness of intra-modal and inter-modal feature integration. These findings demonstrate that each loss component plays a complementary and indispensable role in the optimization process. Additional ablation results and detailed analyses are provided in Appendix B.4, B.5, B.6, and B.7.

### 4.6 COMPLETION PERFORMANCE VISUALIZATION

To evaluate the effectiveness of the proposed completion methods, we evaluate feature distributions on the MOSI dataset under a 90% missing rate. Specifically, we measure the Euclidean distances between the features of the missing data (P1), intra-modal completion (P2), and inter-modal completion (P3) against their corresponding complete data, as illustrated in Figure 4. As shown in P1, missing data exhibit large deviations from the complete representations, with wide dispersion and a high median, indicating severe information loss. A few outliers with very small distances appear in P1, which stem from cases where non-critical features are missing. After completion, these outliers vanish, and in both P2 and P3 the distances to the complete features become smaller and the distributions more compact, demonstrating that intra- and inter-modal completion effectively recover missing information and align the representations closer to their complete counterparts.

### 4.7 CLASSIFICATION PERFORMANCE VISUALIZATION

To evaluate the classification performance of the proposed method, we present the confusion matrices of the baseline model LNLN and our HiTNet on the MOSI dataset under missing rates of 0, 0.1, 0.5, and 0.9 (Fig. 5). As the missing rate increases, both models suffer from class prediction bias, reflecting the difficulty of learning discriminative sentiment features under incomplete modalities and the emergence of prediction collapse. For instance, at a missing rate of 0.9, LNLN predictions concentrate almost exclusively on the neutral class, indicating reliance on a single dominant category. In contrast, the proposed HiTNet produces predictions distributed across multiple sentiment categories even under high missing rates, demonstrating stronger discriminative capacity. This ad-

vantage arises from its intra-modal enhancement and inter-modal regulation streams, which enable effective exploitation of residual information and thereby yield superior robustness under severe missing conditions. For brevity, we present only the MOSI results in the main text. Results on the SIMS dataset, showing consistent trends, are reported in Appendix B.8.

### 4.8 MODALITY-LEVEL MISSINGNESS ANALYSIS

To evaluate the performance of HiTNet under modality-level missing conditions, we conducted experiments on the MOSI dataset and reported the Acc-2 for each condition, as shown in Table 4. The results demonstrate that HiTNet outperforms or is comparable to baseline methods in most cases, especially when visual or audio modalities is present, where HiTNet achieves a 10% improvement over the second-best model. This indicates that HiTNet significantly enhances sensitivity to visual and audio modalities. In the human brain, the thalamus integrates and regulates various sensory information to ensure completeness and accuracy. It is precisely through the inspiration from this brain function that the thalamus-inspired inter-modal regulation stream

Table 4: Acc-2 under modality-level missing conditions on the MOSI dataset. Note: The modalities inside the { } are the present modalities.

| Method | {V} | {A} | {L} | {V,A} | {V,L} | {A,L} |
|--------|-----|-----|-----|-------|-------|-------|
| MMIM | 48.20 | 48.64 | 81.29 | 49.61 | 81.15 | 81.83 |
| CENET | 51.85 | 51.80 | 81.54 | 51.80 | 81.54 | 81.49 |
| TETFN | 55.25 | 55.25 | 81.05 | 55.25 | 81.00 | 81.10 |
| ALMT | 54.96 | 55.10 | 79.83 | 55.05 | 79.98 | 79.98 |
| LNLN | 49.03 | 49.03 | 82.48 | 49.03 | 82.21 | **82.26** |
| **HiTNet** | **59.33** | **59.29** | **82.49** | **59.04** | **82.26** | 81.90 |

in HiTNet effectively integrates and completes missing modality information, even in the absence of the language modality, thereby improving the model's sentiment analysis capability.

## 5 CONCLUSION

In this paper, we propose a Hippocampal-Thalamic inspired dual-stream completion Network (HiT-Net). On the one hand, the intra-modal enhancement stream simulates memory retrieval in the hippocampus to complete missing data by mining modality-specific information. On the other hand, the inter-modal regulation stream mimics thalamic perceptual regulation by estimating modality confidence to guide adaptive cross-modal completion. Finally, the outputs of both streams are integrated via a hierarchical fusion module. Experimental results demonstrate the effectiveness of HiTNet under missing data scenarios. In future work, we plan to investigate strategies for addressing classification imbalance, aiming to enhance model robustness under imbalanced scenarios.

## 6 ETHICS STATEMENT

This work adheres to the ICLR Code of Ethics. Our study does not involve direct experimentation with human or animal subjects. All datasets used, including MOSI, MOSEI, and SIMS, were accessed and utilized in compliance with their respective usage guidelines, ensuring no violation of privacy. We have taken care to avoid any biases or discriminatory outcomes in our research process. No personally identifiable information was used, and no experiments were conducted that could raise privacy or security concerns. We are committed to maintaining transparency and integrity throughout the research process.

## 7 REPRODUCIBILITY STATEMENT

We make every effort to ensure that the results presented in this paper are reproducible. All code and datasets have been made publicly available in an anonymous repository to facilitate replication and verification. The experimental setup, including training steps, model configurations, and hardware details, is described in detail in the paper. We have also provided a full description of HiTNet to assist others in reproducing our experiments. Additionally, the public datasets used in the paper, such as MOSI, MOSEI, and SIMS, are publicly available, ensuring consistent and reproducible evaluation results. We believe these measures will enable other researchers to reproduce our work and further advance the field.

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

## A  THE USE OF LARGE LANGUAGE MODELS (LLMS)

Large Language Models (LLMs) were employed solely to assist in polishing the manuscript. Specifically, an LLM was used to refine language, improve readability, and enhance clarity across various sections of the paper, including tasks such as sentence rephrasing and grammar checking.

Importantly, the LLM was not involved in the development of research ideas, methodology, or experimental design. All scientific concepts, analyses, and results were conceived and conducted entirely by the authors. The role of the LLM was strictly limited to improving the linguistic quality of the manuscript, without influencing the scientific content or data analysis. We have ensured that the use of LLMs adheres to ethical guidelines and does not contribute to plagiarism or scientific misconduct.

## B  ADDITIONAL EXPERIMENTS AND ANALYSIS

### B.1  HYPERPARAMETER SENSITIVITY ANALYSIS AND AUTOMATIC WEIGHTING STRATEGY

**Manual Tuning and Sensitivity Analysis.** We investigated the impact of different loss weightings on model performance by manually adjusting the relative weights assigned to the three loss components. Specifically, we adjust the relative weights assigned to the three loss components to study how the balance among them influences the final outcomes. Keeping the overall training process unchanged, we evaluate several combinations of loss weights on each dataset individually, as shown in Table 5, Table 6, and Table 7. The experimental results indicate that the configuration of loss weights plays a critical role in balancing different optimization objectives, thereby affecting the overall model performance. Notably, the optimal weight combinations vary across datasets. The optimal loss weights $(\alpha, \beta, \gamma)$ for the MOSI, MOSEI, and SIMS datasets are $(10.0, 0.5, 0.1)$, $(1.5, 0.9, 9.0)$, and $(10.0, 0.9, 0.1)$, respectively.

**Automatic Weighting Mechanism.** To avoid the complexity of manual hyperparameter tuning and ensure reproducible performance, we replace fixed weights with an automatic weighting mechanism based on homoscedastic uncertainty. Unlike the manual setting, where each loss term requires exhaustive search for an appropriate coefficient, this mechanism treats the weight of each auxiliary loss as a learnable parameter. During training, the model adaptively estimates the relative uncertainty associated with each auxiliary objective. Loss terms with higher uncertainty are automatically downweighted, while more reliable ones receive stronger emphasis. This allows the optimization process to dynamically balance the contribution of the main loss and the auxiliary regularization losses, resulting in a more stable and self-adjusting training procedure without extensive hyperparameter tuning.

We compared the performance of the best manually tuned configurations against the proposed automatic weighting strategy across all missing rate settings. As shown in Table 8, Table 9, and Table 10, the quantitative results demonstrate that the automatic weighting strategy consistently yields performance that is superior to, or competitive with, the manually tuned baselines. This confirms that our automatic approach not only eliminates the need for human intervention in hyperparameter search but also further unlocks the model's potential by automatically balancing the training objectives, ensuring robust reproducibility across diverse datasets.

### B.2  EFFECT OF THE SPARSE ACTIVATION MECHANISM

To comprehensively evaluate the effectiveness of the sparse activation mechanism, we designed two sets of comparative experiments. First, we replaced the entire sparse activation network module with a single sub-network to verify the impact of the sparse activation network on model performance. Second, to verify the model's sensitivity to the $k$ value within the sparse activation mechanism, we fixed the total number of sub-networks at $n = 5$ and retrained the model with varying $k$ values.

Table 5: Effect of loss weighting on model performance on the MOSI dataset. Note: The lower MAE corresponds to superior results.

| $\alpha, \beta, \gamma$ | MOSI | | | | | |
| | Acc-7 | Acc-5 | Acc-2 | F1 | MAE | Corr |
|---|---|---|---|---|---|---|
| 15.0, 0.5, 0.1 | 35.21 | 39.07 | 74.04 / 72.58 | 74.10 / 72.87 | 1.051 | 0.541 |
| 5.0, 0.5, 0.1 | 34.74 | 38.83 | 73.87 / 72.39 | 74.47 / 72.55 | 1.041 | 0.538 |
| 10.0, 0.9, 0.1 | 34.91 | 38.97 | 73.66 / 72.29 | 73.76 / 72.41 | 1.050 | **0.544** |
| 10.0, 0.5, 0.9 | 35.07 | **39.30** | 74.10 / 72.65 | 74.29 / 73.05 | **1.037** | 0.535 |
| 10.0, 0.5, 0.1 | **35.26** | 39.22 | **74.12 / 72.66** | **74.53 / 73.10** | 1.043 | 0.539 |

Table 6: Effect of loss weighting on model performance on the MOSEI dataset. Note: The lower MAE corresponds to superior results.

| $\alpha, \beta, \gamma$ | MOSEI | | | | | |
| | Acc-7 | Acc-5 | Acc-2 | F1 | MAE | Corr |
|---|---|---|---|---|---|---|
| 0.5, 0.9, 9.0 | 47.15 | 48.05 | 78.02 / 77.95 | 78.81 / 78.42 | 0.666 | 0.588 |
| 1.5, 5.0, 9.0 | 46.45 | **48.07** | 78.12 / 78.06 | 78.65 / 78.31 | 0.667 | 0.586 |
| 1.5, 0.1, 9.0 | 46.89 | 47.83 | 78.14 / 78.03 | **78.98** / 79.12 | 0.666 | 0.590 |
| 1.5, 0.9, 1.0 | 47.05 | 47.99 | 78.18 / 78.03 | 78.84 / 78.65 | 0.667 | 0.586 |
| 1.5, 0.9, 9.0 | **47.19** | 47.98 | **78.29 / 79.28** | 78.84 / **81.46** | **0.665** | **0.591** |

Table 7: Effect of loss weighting on model performance on the SIMS dataset. Note: The lower MAE corresponds to superior results.

| $\alpha, \beta, \gamma$ | SIMS | | | | | |
| | Acc-5 | Acc-3 | Acc-2 | F1 | MAE | Corr |
|---|---|---|---|---|---|---|
| 0.5, 0.9, 0.1 | 35.09 | 59.02 | 73.59 | 77.58 | 0.515 | 0.375 |
| 5.0, 0.9, 0.1 | 35.25 | 58.84 | 73.59 | 77.15 | 0.518 | 0.376 |
| 10.0, 0.5, 0.1 | 35.11 | 59.17 | 73.96 | 77.39 | 0.511 | 0.388 |
| 10.0, 0.9, 0.5 | 34.90 | 58.90 | 75.10 | **78.80** | 0.515 | 0.378 |
| 10.0, 0.9, 0.1 | **35.62** | **59.28** | **73.99** | 77.33 | **0.504** | **0.389** |

We conducted experiments across all missing rates and calculated the average performance. The results on the MOSI dataset are presented in Table 11.When the sparse activation network degenerates into a single sub-network ($n = k = 1$), the performance significantly declines. This indicates that multiple sub-networks provide diverse semantic enhancement pathways, which are crucial for capturing richer intra-modal feature patterns. Furthermore, with $n = 5$ fixed, the model performance gradually improves as $k$ increases, peaking at $k = 3$. This suggests that a moderate increase in the number of activated sub-networks helps integrate multi-perspective intra-modal semantic information, thereby enhancing robustness. However, when $k$ increases to 5 (full activation), the performance declines. This demonstrates that the full activation strategy introduces a substantial amount of redundant information irrelevant to the current sample, diminishing the advantages of sparse selection. The above analysis validates that the sparse activation mechanism plays a crucial role in intra-modal semantic enhancement.

### B.3 DETAILED PERFORMANCE UNDER DIFFERENT MISSING RATES

Tables 12, 13, and 14 present a detailed comparison of HiTNet and baseline models under varying missing rates across the three datasets. According to the experimental results, HiTNet consistently achieves the best performance on the MOSI and MOSEI datasets under all missing rate settings, and also demonstrates highly competitive results on the SIMS dataset. Additionally, CENET and TETFN achieve relatively strong performance on certain metrics. Overall, as the missing rate $r$ increases, the performance of all models declines, confirming the substantial impact of modality

Table 8: Performance comparison between Manual Tuning and Automatic Weighting strategies on the MOSI dataset. Note: The lower MAE corresponds to superior results.

| Method | Acc-7 | Acc-5 | Acc-2 | F1 | MAE | Corr |
|---|---|---|---|---|---|---|
| HiTNet (Manual) | **35.26** | 39.22 | 74.12 / 72.66 | 74.53 / 73.10 | 1.043 | **0.539** |
| HiTNet (Auto) | 35.02 | **39.24** | **75.73 / 74.15** | **75.93 / 74.31** | **1.037** | 0.537 |

Table 9: Performance comparison between Manual Tuning and Automatic Weighting strategies on the MOSEI dataset. Note: The lower MAE corresponds to superior results.

| Method | Acc-7 | Acc-5 | Acc-2 | F1 | MAE | Corr |
|---|---|---|---|---|---|---|
| HiTNet (Manual) | **47.19** | **47.98** | 78.29 / **79.28** | 78.84 / **81.46** | 0.665 | 0.591 |
| HiTNet (Auto) | 47.15 | 47.91 | **79.39** / 78.88 | **79.99** / 79.80 | **0.654** | **0.593** |

Table 10: Performance comparison between Manual Tuning and Automatic Weighting strategies on the SIMS dataset. Note: The lower MAE corresponds to superior results.

| Method | Acc-5 | Acc-3 | Acc-2 | F1 | MAE | Corr |
|---|---|---|---|---|---|---|
| HiTNet (Manual) | **35.62** | **59.28** | 73.99 | **77.33** | **0.504** | **0.389** |
| HiTNet (Auto) | 34.84 | 59.23 | **74.13** | 77.26 | 0.511 | 0.387 |

Table 11: Sensitivity analysis of the Top-$k$ hyperparameter in the sparse activation mechanism on the MOSI dataset. Note: The lower MAE corresponds to superior results.

| Method | Acc-7 | Acc-5 | Acc-2 | F1 | MAE | Corr |
|---|---|---|---|---|---|---|
| n=k=1 | 34.88 | 38.85 | 73.22 / 72.15 | 73.71 / 72.55 | 1.054 | 0.531 |
| n=5,k=1 | 34.03 | 37.42 | 73.25 / 72.09 | 73.63 / 72.28 | 1.067 | 0.537 |
| n=5,k=2 | 34.73 | 38.68 | 73.50 / 72.25 | 73.84 / 72.65 | 1.050 | 0.538 |
| **n=5,k=3** | **35.26** | **39.22** | **74.12 / 72.66** | **74.53 / 73.10** | **1.043** | 0.539 |
| n=5,k=4 | 33.72 | 37.53 | 73.58 / 72.11 | 73.97 / 72.85 | 1.057 | **0.541** |
| n=k=5 | 33.22 | 37.36 | 73.10 / 72.16 | 73.25 / 72.24 | 1.067 | 0.536 |

data incompleteness on multimodal sentiment analysis. Meanwhile, at higher missing rates, our model still achieves relatively good results, demonstrating the effectiveness of the brain-inspired mechanisms in handling incomplete data. Future work may further explore more robust and generalizable modality completion mechanisms to mitigate performance degradation under high missing rate scenarios.

### B.4 Effects of the Semantic Memory Module

As shown in Figure 6, we conduct a comparative analysis between the complete model (HiTNet) and its ablated variant without the semantic memory module (HiTNet w/o SMM) under different modality missings rates on the MOSI and SIMS datasets. Specifically, the figures present bar charts for the F1 metric and line charts for the MAE metric, providing an intuitive visualization of performance trends under varying missing rates. The experimental results demonstrate that the exclusion of the semantic memory module leads to a noticeable degradation in overall model performance. In most settings, HiTNet w/o SMM exhibits lower F1 and higher MAE values compared to the full model. These findings highlight the critical role of the semantic memory module in mitigating the effects of modality missingness by effectively preserving and exploiting residual information within each modality, thereby enhancing the robustness and generalization capability of the model.

### B.5 Effect of Memory Size on Model Performance

To assess the influence of memory size on model performance, we conduct a series of experiments on the MOSI and SIMS datasets under varying levels of data missingness. Specifically, we evaluate

Table 12: Robustness comparison of the overall performance on MOSI datasets. Note: The lower MAE corresponds to superior results.

| Method | Acc-7 | Acc-5 | Acc-2 | F1 | MAE | Corr | Method | Acc-7 | Acc-5 | Acc-2 | F1 | MAE | Corr |
|---|---|---|---|---|---|---|---|---|---|---|---|---|---|
| | | | Random Missing Rate $r = 0$ | | | | | | | Random Missing Rate $r = 0.5$ | | | |
| MISA | 43.05 | 48.30 | 82.78 / 81.24 | 82.83 / 81.23 | 0.771 | 0.777 | MISA | 28.14 | 30.61 | 70.53 / 69.34 | 70.50 / 69.20 | 1.124 | 0.519 |
| Self-MM | 42.81 | **52.38** | **85.22 / 83.24** | **85.19 / 83.26** | 0.720 | 0.790 | Self-MM | 26.97 | 31.39 | 67.43 / 67.54 | 64.27 / 66.81 | 1.129 | 0.503 |
| MMIM | **45.92** | 49.85 | 83.43 / 81.97 | 83.43 / 81.94 | 0.744 | 0.778 | MMIM | 28.33 | 29.89 | 68.09 / 66.52 | 66.15 / 64.59 | 1.128 | 0.501 |
| CENET | 43.20 | 50.39 | 83.08 / 81.49 | 83.06 / 81.48 | 0.748 | 0.785 | CENET | 28.33 | 30.90 | 72.46 / 66.08 | 71.10 / 63.50 | 1.130 | 0.496 |
| TETFN | 44.07 | 51.31 | 82.62 / 81.10 | 82.67 / 81.09 | **0.719** | **0.794** | TETFN | 27.55 | 31.34 | 67.23 / 65.06 | 64.30 / 61.78 | 1.157 | 0.492 |
| TFR-Net | 40.82 | 47.91 | 83.64 / 81.68 | 83.57 / 81.61 | 0.805 | 0.760 | TFR-Net | 25.85 | 30.71 | 64.83 / 63.02 | 58.04 / 56.64 | 1.270 | 0.443 |
| ALMT | 42.37 | 48.49 | 84.91 / 82.75 | 85.01 / 82.94 | 0.752 | 0.768 | ALMT | 28.42 | 31.25 | 68.24 / 65.94 | 69.74 / 68.54 | 1.138 | 0.485 |
| LNLN | 44.56 | 49.76 | 84.25 / 81.24 | 84.61 / 81.79 | 0.751 | 0.778 | LNLN | 33.92 | **38.39** | 73.37 / 71.86 | 73.70 / 72.30 | **1.059** | 0.536 |
| **HiTNet** | 44.31 | 50.73 | 83.84 / 81.63 | 84.18 / 81.61 | 0.749 | 0.779 | **HiTNet** | 33.97 | 37.76 | **74.54 / 73.18** | **74.68 / 73.41** | 1.066 | **0.553** |
| | | | Random Missing Rate $r = 0.1$ | | | | | | | Random Missing Rate $r = 0.6$ | | | |
| MISA | 40.28 | 46.21 | 80.18 / 79.01 | 80.21 / 78.97 | 0.847 | 0.721 | MISA | 24.68 | 27.12 | 66.97 / 65.84 | 66.94 / 65.69 | 1.200 | 0.441 |
| Self-MM | 40.33 | 49.03 | 81.40 / 80.03 | 81.19 / 80.03 | 0.812 | 0.728 | Self-MM | 24.34 | 27.31 | 63.47 / 63.36 | 58.94 / 62.07 | 1.209 | 0.425 |
| MMIM | 42.61 | 46.65 | 79.98 / 78.13 | 79.83 / 77.99 | 0.825 | 0.718 | MMIM | 25.41 | 27.11 | 63.67 / 62.49 | 60.87 / 59.48 | 1.208 | 0.418 |
| CENET | 40.13 | 46.60 | 80.08 / 78.38 | 79.91 / 78.20 | 0.837 | 0.719 | CENET | 24.54 | 26.53 | 67.58 / 61.47 | 64.87 / 57.86 | 1.215 | 0.415 |
| TETFN | 40.67 | 46.84 | 80.59 / 78.91 | 80.55 / 78.79 | 0.805 | 0.731 | TETFN | 25.12 | 27.99 | 64.42 / 61.23 | 58.68 / 56.08 | 1.238 | 0.417 |
| TFR-Net | 38.63 | 45.82 | 79.27 / 77.99 | 78.70 / 77.61 | 0.872 | 0.705 | TFR-Net | 24.05 | 28.33 | 61.64 / 59.47 | 52.44 / 50.53 | 1.371 | 0.363 |
| ALMT | 39.84 | 45.48 | 80.90 / 78.67 | 81.15 / 79.08 | 0.843 | 0.703 | ALMT | 25.41 | 27.36 | 64.53 / 62.15 | 66.81 / 65.87 | 1.214 | 0.407 |
| LNLN | 42.37 | 47.91 | 81.20 / 78.43 | 81.62 / 79.04 | 0.820 | 0.724 | LNLN | **30.37** | **34.35** | 69.00 / 67.69 | **69.19 / 67.99** | **1.147** | 0.458 |
| **HiTNet** | **45.48** | **50.87** | **82.77 / 81.20** | **83.07 / 81.21** | 0.805 | 0.733 | **HiTNet** | 30.17 | 33.24 | 69.05 / 68.51 | 69.09 / 68.62 | 1.178 | **0.459** |
| | | | Random Missing Rate $r = 0.2$ | | | | | | | Random Missing Rate $r = 0.7$ | | | |
| MISA | 36.25 | 41.55 | 77.54 / 76.34 | 77.58 / 76.30 | 0.939 | 0.654 | MISA | 21.14 | 23.27 | 65.09 / 63.89 | 65.07 / 63.74 | 1.257 | 0.381 |
| Self-MM | 36.64 | 43.98 | 78.15 / 76.48 | 77.76 / 76.51 | 0.901 | 0.660 | Self-MM | 20.70 | 23.81 | 61.74 / 61.46 | 55.11 / 58.97 | 1.271 | 0.339 |
| MMIM | 39.07 | 42.66 | 76.42 / 74.54 | 76.12 / 74.22 | 0.918 | 0.651 | MMIM | 22.35 | 24.00 | 61.23 / 59.18 | 57.15 / 54.36 | 1.267 | 0.342 |
| CENET | 38.00 | 42.32 | 77.49 / 74.64 | 77.35 / 74.28 | 0.916 | 0.654 | CENET | 22.35 | 59.86 | 63.82 / 59.43 | 53.79 / 54.22 | 1.269 | 0.335 |
| TETFN | 35.81 | 41.79 | 77.49 / 75.60 | 77.35 / 75.35 | 0.910 | 0.657 | TETFN | 23.13 | 25.27 | 61.13 / 58.65 | 53.79 / 50.77 | 1.293 | 0.337 |
| TFR-Net | 34.70 | 40.13 | 74.70 / 73.52 | 73.57 / 72.70 | 0.987 | 0.622 | TFR-Net | 23.71 | 26.92 | 59.91 / 57.34 | 48.41 / 45.48 | 1.454 | 0.276 |
| ALMT | 35.33 | 40.33 | 77.64 / 75.70 | 77.94 / 76.24 | 0.927 | 0.645 | ALMT | 23.71 | 24.97 | 61.84 / 59.67 | 65.30 / 65.19 | 1.266 | 0.336 |
| LNLN | 39.74 | 45.14 | 79.22 / 76.87 | 79.53 / 77.34 | 0.891 | 0.668 | LNLN | 27.79 | 31.19 | 65.95 / 65.01 | 65.95 / 65.14 | **1.219** | **0.383** |
| **HiTNet** | **41.69** | **46.94** | **80.79 / 78.72** | **80.86 / 78.68** | **0.885** | **0.682** | **HiTNet** | **29.01** | **31.78** | **66.85 / 66.45** | **66.70 / 66.40** | 1.221 | 0.379 |
| | | | Random Missing Rate $r = 0.3$ | | | | | | | Random Missing Rate $r = 0.8$ | | | |
| MISA | 34.60 | 38.97 | 75.76 / 74.54 | 75.82 / 74.51 | 0.989 | 0.618 | MISA | 19.92 | 20.99 | 63.56 / 62.24 | 63.16 / 61.67 | 1.311 | 0.321 |
| Self-MM | 34.89 | 40.67 | 76.37 / 74.98 | 75.68 / 74.94 | 0.967 | 0.614 | Self-MM | 19.29 | 22.11 | 59.55 / 58.26 | 49.98 / 53.56 | 1.313 | 0.282 |
| MMIM | 36.83 | 40.43 | 74.08 / 71.91 | 73.47 / 71.28 | 0.974 | 0.612 | MMIM | 20.26 | 21.77 | 58.33 / 55.30 | 52.46 / 47.89 | 1.314 | 0.287 |
| CENET | 34.74 | 38.97 | 76.83 / 72.01 | 76.56 / 71.30 | 0.983 | 0.605 | CENET | 21.14 | 21.67 | 60.93 / 57.53 | 54.68 / 50.80 | 1.314 | 0.274 |
| TETFN | 33.24 | 38.58 | 75.25 / 73.42 | 74.77 / 72.78 | 0.982 | 0.607 | TETFN | 22.01 | 23.76 | 59.40 / 56.85 | 48.73 / 45.59 | 1.337 | 0.274 |
| TFR-Net | 32.55 | 38.34 | 72.36 / 71.28 | 70.12 / 69.58 | 1.065 | 0.572 | TFR-Net | 23.23 | 27.70 | 58.49 / 55.98 | 44.70 / 41.88 | 1.497 | 0.155 |
| ALMT | 33.04 | 37.17 | 75.15 / 72.94 | 75.51 / 73.66 | 0.992 | 0.596 | ALMT | 23.13 | 23.66 | 60.37 / 58.31 | **65.45 / 66.14** | 1.310 | 0.273 |
| LNLN | 38.00 | 42.81 | 77.29 / 75.46 | 77.56 / 75.68 | 0.953 | 0.617 | LNLN | 26.34 | 28.23 | 62.75 / 62.10 | 62.56 / 62.03 | 1.283 | 0.314 |
| **HiTNet** | **38.78** | **43.44** | **78.81 / 76.97** | **78.91 / 77.16** | **0.948** | **0.643** | **HiTNet** | **26.82** | **28.86** | **64.18 / 62.97** | 64.12 /63.05 | **1.281** | **0.324** |
| | | | Random Missing Rate $r = 0.4$ | | | | | | | Random Missing Rate $r = 0.9$ | | | |
| MISA | 32.65 | 35.37 | 73.88 / 72.59 | 73.88 / 72.49 | 1.041 | 0.585 | MISA | 17.7 | 18.4 | 58.64 / 58.21 | 56.84 / 56.19 | 1.369 | **0.226** |
| Self-MM | 31.20 | 36.30 | 73.17 / 71.96 | 71.74 / 71.75 | 1.027 | 0.579 | Self-MM | 18.32 | 19.78 | 58.59 / 55.25 | 46.16 / 47.46 | 1.353 | 0.197 |
| MMIM | 33.38 | 35.76 | 70.84 / 68.90 | 69.69 / 67.80 | 1.034 | 0.576 | MMIM | 18.95 | 19.53 | 55.29 / 51.65 | 47.33 / 40.89 | 1.357 | 0.186 |
| CENET | 32.26 | 36.15 | 73.38 / 71.53 | 72.75 / 70.26 | 1.031 | 0.574 | CENET | 19.15 | 19.10 | 58.99 / 54.76 | 50.01 / 46.58 | 1.357 | 0.181 |
| TETFN | 30.66 | 35.28 | 72.05 / 70.07 | 70.79 / 68.58 | 1.051 | 0.571 | TETFN | 20.75 | 21.19 | 58.43 / 55.88 | 45.24 / 42.12 | 1.378 | 0.186 |
| TFR-Net | 30.17 | 35.76 | 68.75 / 67.74 | 64.71 / 64.41 | 1.142 | 0.537 | TFR-Net | 21.67 | 25.12 | 57.93 / 55.44 | 43.01 / 40.18 | 1.534 | 0.155 |
| ALMT | 31.44 | 35.03 | 73.12 / 71.14 | 73.85 / 72.47 | 1.045 | 0.560 | ALMT | 20.31 | 20.50 | 57.32 / 56.66 | 64.92 / **67.82** | 1.349 | 0.205 |
| LNLN | 36.49 | 41.11 | 76.01 / 74.25 | 76.31 / 74.67 | 0.987 | 0.594 | LNLN | 22.98 | 23.86 | 56.50 / 56.51 | 56.32 / 56.47 | 1.349 | 0.202 |
| **HiTNet** | **36.88** | **42.27** | **77.90 / 76.97** | **77.89 / 77.02** | **0.984** | **0.619** | **HiTNet** | **25.51** | **26.38** | **62.51 / 60.02** | **65.75 / 63.87** | **1.319** | 0.221 |

the F1 score across different memory size (number of key-value pairs) configurations, and visualize the results in Figure 7. The curves reveal that memory size substantially affects the model's ability to maintain discriminative power under incomplete input conditions. Among the evaluated settings, a memory size of 64 yields superior F1 scores across a range of missing rates, suggesting an optimal trade-off between representational capacity and regularization. Smaller memory sizes may lack sufficient capacity to store useful information, while larger sizes may introduce unnecessary redundancy or overfitting. These findings underscore the importance of appropriately configuring the memory module to enhance model robustness and generalization in the presence of partial data missing.

## B.6 EFFECT OF HIERARCHICAL FUSION STRATEGY

We propose a hierarchical fusion method based on a CrossTransformer to integrate the intra-modal and inter-modal completion features. To validate the effectiveness of the proposed hierarchical fusion strategy, we design a series of comparative experiments across multiple multimodal fusion schemes. For a fair comparison, we conducted experiments on each fusion method on the MOSI dataset and reported the average results of ten missing rates ranging from 0 to 0.9, as shown in Table 15.

Table 13: Robustness comparison of the overall performance on MOSEI datasets. Note: The lower MAE corresponds to superior results.

| Method | Acc-7 | Acc-5 | Acc-2 | F1 | MAE | Corr | Method | Acc-7 | Acc-5 | Acc-2 | F1 | MAE | Corr |
|---|---|---|---|---|---|---|---|---|---|---|---|---|---|
| | | | Random Missing Rate $r = 0$ | | | | | | | Random Missing Rate $r = 0.5$ | | | |
| MISA | 51.79 | 53.85 | 85.28 / 84.10 | 85.10 / 83.75 | 0.552 | 0.759 | MISA | 38.12 | 36.05 | 67.38 / 73.21 | 58.38 / 64.14 | 0.834 | 0.492 |
| Self-MM | 53.89 | 55.72 | 85.34 / 84.68 | 85.11 / 84.66 | 0.531 | 0.764 | Self-MM | 42.70 | 43.14 | 71.97 / 75.81 | 67.40 / 70.38 | 0.733 | 0.477 |
| MMIM | 50.76 | 53.04 | 83.53 / 81.65 | 83.39 / 81.41 | 0.576 | 0.724 | MMIM | 38.68 | 39.21 | 71.75 / 74.45 | 67.70 / 67.96 | 0.775 | 0.470 |
| CENET | **54.39** | **56.12** | 85.49 / 82.30 | 85.41 / 82.60 | **0.531** | 0.770 | CENET | 45.12 | 45.52 | 73.33 / 77.16 | 69.80 / 74.14 | 0.720 | 0.515 |
| TETFN | 44.07 | 55.96 | 82.62 / 81.10 | 82.67 / 81.09 | 0.719 | **0.794** | TETFN | 27.55 | 45.63 | 67.23 / 65.06 | 64.30 / 61.78 | 1.157 | 0.492 |
| TFR-Net | 53.71 | 47.91 | 84.96 / 84.65 | 84.71 / 84.34 | 0.550 | 0.745 | TFR-Net | 45.00 | 30.71 | 71.53 / 75.69 | 66.88 / 70.07 | 0.730 | 0.471 |
| ALMT | 52.18 | 53.89 | 85.62 / 83.99 | 85.69 / 84.53 | 0.542 | 0.752 | ALMT | 37.82 | 38.34 | 77.40 / 77.48 | 77.73 / 77.80 | 0.683 | 0.461 |
| LNLN | 50.66 | 51.94 | 84.14 / 83.61 | 84.53 / 84.02 | 0.572 | 0.735 | LNLN | 44.90 | 45.59 | 76.44 / 78.10 | 77.23 / 79.30 | 0.710 | 0.529 |
| **HiTNet** | 52.95 | 54.52 | **85.63 / 84.70** | **85.70 / 85.35** | 0.541 | 0.765 | **HiTNet** | **45.37** | **46.19** | **78.67 / 79.22** | **78.86/ 80.62** | **0.681** | **0.592** |
| | | | Random Missing Rate $r = 0.1$ | | | | | | | Random Missing Rate $r = 0.6$ | | | |
| MISA | 50.13 | 51.34 | 82.21 / 82.28 | 81.28 / 80.79 | 0.598 | 0.722 | MISA | 36.16 | 33.30 | 65.55 / 72.30 | 54.64 / 62.12 | 0.875 | 0.415 |
| Self-MM | 51.80 | 53.18 | 83.03 / **83.79** | 82.43 / 83.23 | 0.564 | 0.725 | Self-MM | 41.47 | 41.75 | 69.33 / 73.93 | 63.01 / 66.76 | 0.762 | 0.401 |
| MMIM | 49.09 | 51.19 | 82.00 / 81.09 | 81.57 / 80.15 | 0.602 | 0.696 | MMIM | 37.13 | 37.48 | 68.83 / 73.16 | 63.09 / 65.43 | 0.808 | 0.402 |
| CENET | **52.83** | 54.23 | 83.75 / 82.41 | 83.42 / 82.34 | **0.556** | 0.739 | CENET | 44.45 | 44.64 | 70.50 / 75.39 | 65.27 / 70.86 | 0.749 | 0.446 |
| TETFN | 40.67 | **54.28** | 80.59 / 78.91 | 80.55 / 78.79 | 0.805 | 0.731 | TETFN | 25.12 | 44.07 | 63.42 / 61.23 | 58.68 / 56.08 | 1.238 | 0.417 |
| TFR-Net | 52.29 | 45.82 | 82.92 / 83.31 | 82.25 / 82.40 | 0.573 | 0.715 | TFR-Net | 43.88 | 28.33 | 68.80 / 74.05 | 62.51 / 67.07 | 0.762 | 0.397 |
| ALMT | 49.98 | 51.38 | 84.14 / 82.84 | 84.23 / 83.04 | 0.583 | 0.718 | ALMT | 35.99 | 36.30 | 74.98 / 76.26 | 75.44 / 76.71 | 0.710 | 0.395 |
| LNLN | 49.96 | 51.25 | 83.32 / 82.73 | 83.66 / 82.91 | 0.591 | 0.712 | LNLN | 43.52 | 44.00 | 73.82 / 76.50 | 75.03 / 78.33 | 0.736 | 0.471 |
| **HiTNet** | 52.54 | 53.96 | **84.73 / 83.77** | **84.79 / 83.89** | 0.559 | **0.744** | **HiTNet** | **44.54** | **45.33** | **75.81 / 77.72** | **76.01 / 79.67** | 0.710 | **0.539** |
| | | | Random Missing Rate $r = 0.2$ | | | | | | | Random Missing Rate $r = 0.7$ | | | |
| MISA | 47.24 | 47.66 | 77.84 / 79.93 | 75.56 / 76.88 | 0.659 | 0.674 | MISA | 34.54 | 31.21 | 64.28 / 71.71 | 51.82 / 60.65 | 0.906 | 0.344 |
| Self-MM | 49.44 | 50.51 | 80.84 / 82.33 | 79.76 / 81.17 | 0.604 | 0.678 | Self-MM | 39.93 | 40.12 | 66.79 / 72.55 | 58.05 / 63.45 | 0.786 | 0.329 |
| MMIM | 46.27 | 47.99 | 79.93 / 79.66 | 79.08 / 77.68 | 0.642 | 0.653 | MMIM | 35.25 | 35.47 | 66.89 / 72.26 | 58.90 / 63.26 | 0.834 | 0.341 |
| CENET | 50.72 | 51.85 | 81.46 / 81.62 | 80.78 / 81.17 | 0.590 | 0.698 | CENET | 43.93 | 44.03 | 67.50 / 73.39 | 59.88 / 67.02 | 0.776 | 0.384 |
| TETFN | 35.81 | 52.35 | 77.49 / 75.60 | 77.35 / 75.35 | 0.910 | 0.657 | TETFN | 23.13 | 43.15 | 61.13 / 58.65 | 53.79 / 50.77 | 1.293 | 0.337 |
| TFR-Net | 51.04 | 40.13 | 80.47 / 81.61 | 79.29 / 79.99 | 0.604 | 0.672 | TFR-Net | 42.91 | 26.92 | 66.64 / 72.77 | 58.32 / 64.02 | 0.786 | 0.322 |
| ALMT | 46.61 | 47.82 | 82.71 / 81.65 | 82.82 / 81.83 | 0.607 | 0.669 | ALMT | 34.78 | 34.95 | 71.62 / 73.98 | 72.24 / 74.54 | 0.743 | 0.315 |
| LNLN | 48.75 | 49.95 | 81.70 / 81.68 | 81.95 / 81.89 | 0.616 | 0.677 | LNLN | 42.22 | 42.56 | 71.55 / 74.74 | **73.49 / 77.40** | 0.762 | 0.408 |
| **HiTNet** | 51.32 | 52.50 | **83.74/ 83.09** | **83.78 / 83.99** | 0.583 | 0.716 | **HiTNet** | **43.96** | **44.19** | **73.12 / 75.98** | 73.27 / 78.56 | 0.743 | **0.482** |
| | | | Random Missing Rate $r = 0.3$ | | | | | | | Random Missing Rate $r = 0.8$ | | | |
| MISA | 43.99 | 43.40 | 73.32 / 77.28 | 68.91 / 72.25 | 0.724 | 0.615 | MISA | 33.29 | 29.51 | 63.43 / 71.30 | 49.95 / 59.69 | 0.927 | 0.267 |
| Self-MM | 47.23 | 48.07 | 77.63 / 79.99 | 75.69 / 77.74 | 0.653 | 0.610 | Self-MM | 38.69 | 38.78 | 65.07 / 71.83 | 54.44 / 61.49 | 0.805 | 0.259 |
| MMIM | 43.25 | 44.73 | 77.08 / 77.79 | 75.46 / 74.49 | 0.690 | 0.597 | MMIM | 33.46 | 33.71 | 64.97 / 71.57 | 54.76 / 61.45 | 0.858 | 0.269 |
| CENET | 48.49 | 49.37 | 78.65 / 80.02 | 77.34 / 78.94 | 0.636 | 0.640 | CENET | **42.71** | 42.74 | 65.88 / 72.16 | 56.80 / 64.67 | 0.798 | 0.316 |
| TETFN | 33.24 | **50.23** | 75.25 / 73.42 | 74.77 / 72.78 | 0.982 | 0.607 | TETFN | 22.01 | 42.11 | 59.40 / 56.85 | 48.73 / 45.59 | 1.337 | 0.274 |
| TFR-Net | 48.75 | 38.34 | 77.48 / 79.29 | 75.43 / 76.52 | 0.650 | 0.604 | TFR-Net | 42.23 | 27.70 | 65.05 / 71.95 | 54.91 / 61.82 | 0.807 | 0.241 |
| ALMT | 43.04 | 44.05 | 80.94 / 79.94 | 81.15 / 80.20 | 0.632 | 0.598 | ALMT | 34.01 | 34.09 | 68.15 / 71.48 | 69.12 / 72.28 | 0.774 | 0.231 |
| LNLN | 47.36 | 48.40 | 80.11 / 80.45 | 80.44 / 80.91 | 0.648 | 0.629 | LNLN | 40.76 | 40.97 | 68.62 / 72.86 | 71.83 / 76.80 | 0.791 | 0.325 |
| **HiTNet** | 49.20 | 50.16 | **81.67 / 81.31** | **82.04 / 82.64** | 0.617 | **0.677** | **HiTNet** | 42.69 | **42.80** | **71.22 / 73.71** | **72.77 / 78.42** | 0.774 | **0.421** |
| | | | Random Missing Rate $r = 0.4$ | | | | | | | Random Missing Rate $r = 0.9$ | | | |
| MISA | 40.87 | 39.53 | 70.46 / 75.04 | 64.02 / 67.93 | 0.780 | 0.561 | MISA | 32.29 | 28.03 | 62.95 / 71.07 | 48.80 / 59.12 | 0.941 | 0.180 |
| Self-MM | 44.40 | 45.04 | 75.02 / 78.09 | 72.01 / 74.48 | 0.694 | 0.554 | Self-MM | 37.46 | 37.50 | 63.85 / 71.24 | 51.32 / 59.72 | 0.821 | 0.188 |
| MMIM | 40.84 | 41.86 | 74.56 / 76.15 | 71.98 / 71.40 | 0.732 | 0.542 | MMIM | 32.61 | 32.67 | 63.69 / 71.10 | 51.26 / 59.99 | 0.877 | 0.197 |
| CENET | 47.12 | 47.74 | 76.03 / 78.57 | 73.87 / 76.75 | 0.678 | 0.587 | CENET | **42.08** | **42.08** | 64.14 / 70.42 | 54.27 / 62.33 | 0.814 | 0.254 |
| TETFN | 30.66 | 47.62 | 72.05 / 70.07 | 70.79 / 68.58 | 1.051 | 0.571 | TETFN | 20.75 | 41.55 | 58.43 / 55.88 | 45.24 / 42.12 | 1.378 | 0.186 |
| TFR-Net | 46.70 | 35.76 | 74.74 / 77.65 | 71.67 / 73.71 | 0.688 | 0.548 | TFR-Net | 41.73 | 25.12 | 63.64 / 71.34 | 52.02 / 59.99 | 0.820 | 0.175 |
| ALMT | 40.40 | 41.21 | 79.40 / 79.16 | 79.68 / 79.50 | 0.651 | 0.536 | ALMT | 34.40 | 34.40 | 61.41 / 68.65 | 63.32 / 69.83 | 0.810 | 0.138 |
| LNLN | 45.99 | 46.88 | 78.49 / 79.70 | 78.98 / 80.46 | 0.673 | 0.592 | LNLN | 40.10 | 40.19 | 64.83 / 71.51 | 70.60 / 77.52 | 0.820 | 0.221 |
| **HiTNet** | 47.46 | 48.25 | **80.38 / 81.05** | **80.52 / 82.01** | 0.642 | 0.645 | **HiTNet** | 41.83 | 41.85 | **67.89 / 72.20** | 70.67 / 79.41 | 0.802 | **0.327** |

First, we evaluated the model's sensitivity to the fusion order of intra-modal completion features for the visual (V), audio (A), and language (L) modalities ($f_V^{\text{intra}}$, $f_A^{\text{intra}}$, and $f_L^{\text{intra}}$). We assessed six possible fusion orders and found that model performance is optimal when the language modality's enhanced features are placed at higher layers and involved in the final stage of fusion. This is because the language modality provides more critical semantic cues for sentiment recognition, and placing it in the later stage of hierarchical fusion helps to better integrate the complementary information from other modalities.

Additionally, we replaced the hierarchical fusion module with three common simple fusion methods: summation (Sum), concatenation (Concat), and attention-based fusion (Attention), and retrained the model to further validate the necessity and effectiveness of hierarchical fusion from a methodological perspective. As shown in Table 15, the performance of these variant models drops significantly, indicating that simple fusion methods cannot adequately capture the hierarchical dependencies between modalities. In contrast, hierarchical fusion more effectively facilitates the interaction of intra-modal and inter-modal complementary features, thereby improving model robustness and performance.

Table 14: Robustness comparison of the overall performance on SIMS datasets. Note: The lower MAE corresponds to superior results.

| Method | Acc-5 | Acc-3 | Acc-2 | F1 | MAE | Corr | Method | Acc-5 | Acc-3 | Acc-2 | F1 | MAE | Corr |
|---|---|---|---|---|---|---|---|---|---|---|---|---|---|
| | | | Random Missing Rate $r = 0$ | | | | | | | Random Missing Rate $r = 0.5$ | | | |
| MISA | 40.55 | 63.38 | 78.19 | 77.22 | 0.449 | 0.576 | MISA | 30.56 | 54.78 | 71.26 | 64.16 | 0.552 | 0.367 |
| Self-MM | 40.77 | 64.92 | 78.26 | 78.00 | **0.421** | 0.584 | Self-MM | 32.02 | 53.90 | 71.41 | 67.11 | 0.517 | 0.390 |
| MMIM | 37.42 | 60.69 | 75.42 | 73.10 | 0.475 | 0.528 | MMIM | 33.41 | 52.37 | 68.49 | 64.81 | 0.553 | 0.336 |
| CENET | 23.85 | 54.05 | 68.71 | 57.82 | 0.578 | 0.137 | CENET | 23.12 | 54.05 | 68.71 | 57.92 | 0.588 | 0.107 |
| TETFN | 41.94 | 65.86 | **80.23** | 79.25 | 0.424 | 0.589 | TETFN | 33.48 | 56.24 | 72.43 | 67.30 | 0.512 | 0.394 |
| TFR-Net | 33.85 | 54.12 | 69.15 | 58.44 | 0.562 | 0.254 | TFR-Net | 24.65 | 52.37 | 67.47 | 58.66 | 0.685 | 0.171 |
| ALMT | 23.41 | 54.78 | 75.64 | 76.27 | 0.527 | 0.536 | ALMT | 18.38 | 47.12 | 68.27 | 71.22 | 0.563 | 0.395 |
| LNLN | 38.66 | 63.97 | 75.93 | **79.89** | 0.458 | 0.570 | LNLN | **36.40** | 57.70 | 72.72 | **78.77** | 0.513 | **0.412** |
| **HiTNet** | **42.01** | 65.86 | 78.99 | 79.01 | 0.423 | **0.592** | **HiTNet** | 35.67 | **58.21** | **74.18** | 76.02 | **0.508** | 0.396 |
| | | | Random Missing Rate $r = 0.1$ | | | | | | | Random Missing Rate $r = 0.6$ | | | |
| MISA | 38.88 | 63.02 | 77.39 | 75.82 | 0.461 | 0.561 | MISA | 27.72 | 53.97 | 70.46 | 61.81 | 0.578 | 0.286 |
| Self-MM | 40.26 | 63.53 | 77.32 | 76.76 | 0.433 | 0.563 | Self-MM | 29.10 | 51.86 | 70.02 | 64.21 | 0.548 | 0.313 |
| MMIM | 37.27 | 60.90 | 74.25 | 72.08 | 0.473 | 0.529 | MMIM | 29.18 | 49.31 | 67.91 | 63.86 | 0.578 | 0.270 |
| CENET | 22.83 | 53.98 | 68.57 | 57.36 | 0.580 | 0.136 | CENET | 22.46 | 53.69 | 69.00 | 58.64 | 0.592 | 0.102 |
| TETFN | 41.36 | 64.62 | 78.92 | 77.70 | **0.432** | **0.578** | TETFN | 29.90 | 53.54 | 70.97 | 64.19 | 0.545 | 0.309 |
| TFR-Net | 30.12 | 53.25 | 68.85 | 59.38 | 0.596 | 0.203 | TFR-Net | 24.80 | 52.59 | 67.03 | 58.30 | 0.696 | 0.157 |
| ALMT | 22.10 | 55.14 | 74.40 | 75.19 | 0.530 | 0.537 | ALMT | 18.67 | 43.69 | 66.81 | 70.69 | 0.574 | 0.322 |
| LNLN | 38.51 | 62.73 | 76.29 | **80.07** | 0.458 | 0.562 | LNLN | **33.70** | 54.63 | 71.55 | **79.56** | 0.535 | **0.352** |
| **HiTNet** | **42.01** | **65.43** | **79.02** | 79.32 | 0.439 | 0.553 | **HiTNet** | 32.61 | **54.92** | **71.77** | 72.87 | **0.532** | 0.322 |
| | | | Random Missing Rate $r = 0.2$ | | | | | | | Random Missing Rate $r = 0.7$ | | | |
| MISA | 38.15 | 59.23 | 74.33 | 71.70 | 0.489 | 0.490 | MISA | 24.87 | 52.52 | 69.95 | 59.54 | 0.601 | 0.167 |
| Self-MM | 38.37 | 61.71 | 74.98 | 73.71 | 0.464 | 0.500 | Self-MM | 25.53 | 50.62 | 69.58 | 62.28 | 0.571 | 0.198 |
| MMIM | 37.27 | 57.33 | 72.36 | 69.80 | 0.504 | 0.460 | MMIM | 28.59 | 46.53 | 66.89 | 62.23 | 0.595 | 0.190 |
| CENET | 22.25 | 54.20 | 68.57 | 57.64 | 0.583 | 0.132 | CENET | 21.81 | 53.32 | 67.69 | 57.87 | 0.599 | 0.070 |
| TETFN | 39.46 | 61.56 | 75.49 | 73.59 | **0.457** | **0.527** | TETFN | 27.86 | 51.06 | 69.29 | 61.09 | 0.572 | 0.190 |
| TFR-Net | 29.03 | 53.61 | 68.64 | 59.74 | 0.619 | 0.191 | TFR-Net | 23.78 | 52.30 | 67.18 | 58.15 | 0.707 | 0.163 |
| ALMT | 21.08 | 53.17 | 72.65 | 73.90 | 0.541 | 0.485 | ALMT | 18.02 | 38.66 | 65.57 | 70.27 | 0.586 | 0.218 |
| LNLN | 38.88 | 61.78 | 74.76 | **78.53** | 0.474 | 0.513 | LNLN | **30.27** | 51.64 | **69.73** | 79.37 | 0.558 | **0.261** |
| **HiTNet** | **40.48** | **62.58** | **75.71** | 75.47 | 0.471 | 0.488 | **HiTNet** | 28.88 | **54.27** | 69.37 | 75.28 | 0.571 | 0.176 |
| | | | Random Missing Rate $r = 0.3$ | | | | | | | Random Missing Rate $r = 0.8$ | | | |
| MISA | 36.40 | 59.3 | 74.11 | 70.40 | 0.505 | 0.464 | MISA | 22.69 | 52.22 | 69.37 | 57.82 | 0.610 | 0.092 |
| Self-MM | 37.93 | 59.81 | 74.76 | 72.85 | 0.474 | 0.487 | Self-MM | 22.03 | 50.77 | 69.51 | 60.68 | 0.585 | 0.138 |
| MMIM | 37.71 | 58.06 | 72.36 | 69.52 | 0.512 | 0.436 | MMIM | 22.32 | 44.35 | 65.28 | 60.53 | 0.607 | 0.145 |
| CENET | 21.44 | 54.05 | 68.42 | 57.41 | 0.578 | 0.175 | CENET | 21.73 | 52.15 | 67.47 | 58.44 | 0.599 | 0.074 |
| TETFN | 38.80 | 61.92 | 75.86 | 73.28 | **0.463** | **0.521** | TETFN | 23.48 | 49.60 | **69.88** | 60.92 | 0.584 | 0.154 |
| TFR-Net | 27.64 | 52.30 | 68.42 | 59.88 | 0.640 | 0.182 | TFR-Net | 22.97 | 52.74 | 67.54 | 57.55 | 0.721 | 0.100 |
| ALMT | 20.35 | 50.62 | 72.06 | 73.64 | 0.546 | 0.469 | ALMT | 18.60 | 34.06 | 64.19 | 69.64 | 0.597 | 0.133 |
| LNLN | 38.37 | 60.98 | 74.25 | **78.60** | 0.478 | 0.509 | LNLN | 27.94 | 50.47 | 69.58 | 80.23 | 0.580 | 0.183 |
| **HiTNet** | **40.04** | **61.93** | **76.81** | 76.85 | 0.467 | 0.495 | **HiTNet** | **29.98** | **54.71** | 69.58 | **81.84** | **0.565** | **0.248** |
| | | | Random Missing Rate $r = 0.4$ | | | | | | | Random Missing Rate $r = 0.9$ | | | |
| MISA | 34.86 | 57.33 | 72.87 | 67.52 | 0.523 | 0.436 | MISA | 20.64 | 52.95 | 69.22 | 57.01 | 0.617 | 0.041 |
| Self-MM | 34.57 | 58.28 | 73.30 | 70.36 | 0.482 | 0.479 | Self-MM | 22.17 | 52.15 | 68.92 | 58.32 | 0.586 | 0.111 |
| MMIM | 34.57 | 55.36 | 69.95 | 66.49 | 0.533 | 0.399 | MMIM | 20.35 | 42.67 | 65.72 | 59.64 | 0.610 | 0.096 |
| CENET | 22.54 | 54.12 | 68.49 | 57.68 | 0.583 | 0.141 | CENET | 20.86 | 48.07 | 65.72 | 58.18 | 0.609 | -0.002 |
| TETFN | 35.81 | 58.93 | 73.81 | 70.66 | **0.473** | **0.504** | TETFN | 22.10 | 45.73 | 68.92 | 58.75 | 0.590 | 0.108 |
| TFR-Net | 25.31 | 51.86 | 67.91 | 59.16 | 0.664 | 0.176 | TFR-Net | 23.05 | 53.76 | 69.08 | 57.71 | 0.721 | 0.088 |
| ALMT | 19.91 | 49.45 | 70.75 | 72.97 | 0.549 | 0.470 | ALMT | 19.47 | 26.91 | 66.23 | 73.76 | 0.596 | 0.076 |
| LNLN | **37.49** | 60.03 | 73.81 | **78.82** | 0.491 | 0.481 | LNLN | 26.19 | 47.48 | 68.64 | 80.42 | 0.591 | 0.127 |
| **HiTNet** | 36.76 | **60.61** | **75.05** | 74.70 | 0.481 | 0.459 | **HiTNet** | 27.79 | **54.27** | **69.37** | **81.91** | **0.580** | **0.158** |

## B.7 EFFECTS OF CONFIDENCE-PERCEPTION MODULE

To verify the effectiveness of our learned context-aware reliability assessment, we designed a comparative experiment against a traditional signal-processing heuristic. We constructed a heuristic variant based on Signal-to-Noise Ratio (SNR-Gated). Instead of learning reliability scores via the network, this variant calculates the frame-level SNR for raw acoustic signals and applies a hard gating mechanism to filter out frames below a specific threshold. We evaluated the performance of both methods on the MOSI dataset across all missing rates (0.0 to 0.9) and reported the average results, as shown in Table 16.

Quantitative analysis indicates that our model consistently outperforms the SNR-gated variant. This comparison reveals that heuristic metrics like SNR only measure physical signal integrity, which

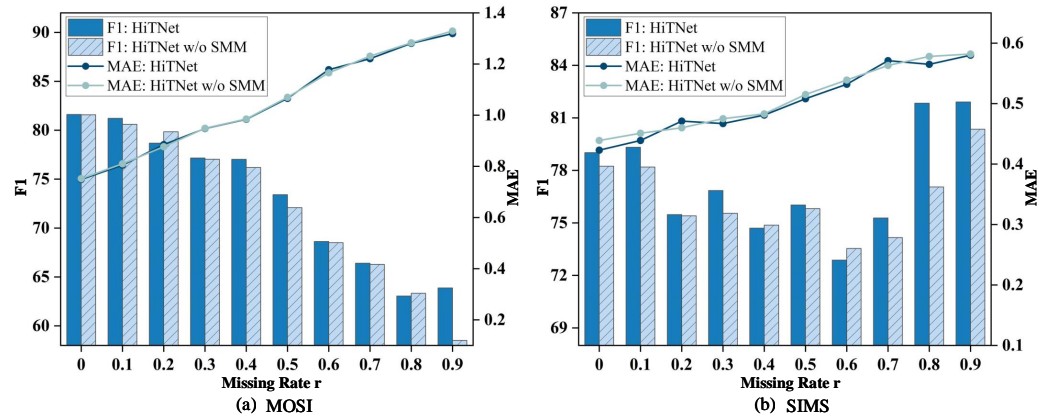

Figure 6: Comparison of HiTNet with and without the semantic memory module (SMM) on the MOSI (a) and SIMS (b) datasets. Bar charts indicate F1, and line plots represent MAE. Note: The lower MAE corresponds to superior results.

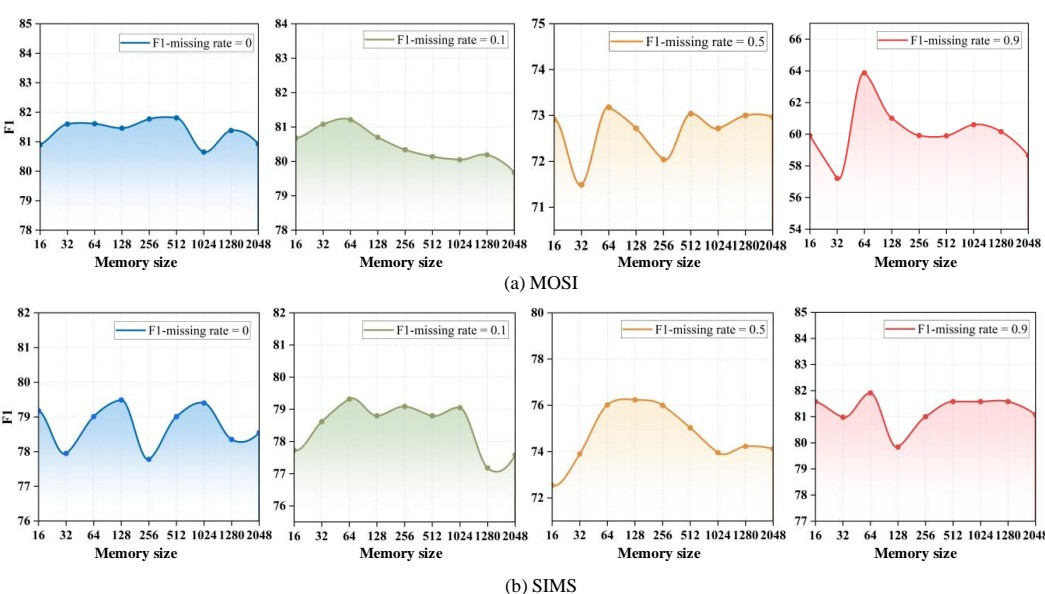

Figure 7: Performance of HiTNet with different memory sizes on the MOSI(a) and SIMS(b) datasets.

does not necessarily align with semantic information value. High-noise segments may still hold critical cues, while clear signals might be redundant. Unlike rigid heuristic filtering, our method learns context-aware reliability scores to selectively preserve and rectify semantically important data, ensuring superior robustness even under high missing rates.

### B.8 CLASSIFICATION PERFORMANCE VISUALIZATION

To analyze model performance under varying degrees of missing modalities, we present the confusion matrices of LNLN and HiTNet on the SIMS dataset at missing rates of 0, 0.1, 0.5, and 0.9, as shown in Figure 8. As the missing rate increases, the baseline LNLN model exhibits class prediction bias, showing a tendency to collapse toward the negative class, indicating over-reliance on a single dominant category under severe modality missingness. In contrast, HiTNet maintains predictions distributed across multiple sentiment categories even at high missing rates, demonstrating stronger discriminative capacity. These observations are consistent with trends seen on MOSI and further validate the robustness of HiTNet under extreme missing conditions.

Table 15: Performance comparison of different multimodal fusion strategies. Note: The lower MAE corresponds to superior results.

| Method | Acc-7 | Acc-5 | Acc-2 | F1 | MAE | Corr |
|---|---|---|---|---|---|---|
| LAV | 34.03 | 37.55 | 73.73 / 72.41 | 74.08 / 72.74 | 1.048 | 0.536 |
| ALV | 34.09 | 38.17 | 73.25 / 72.20 | 73.64 / 72.32 | 1.058 | **0.546** |
| LVA | 34.56 | 38.41 | 73.64 / 72.74 | 74.11 / 72.44 | 1.048 | 0.545 |
| VLA | 34.69 | 38.52 | 73.55 / 72.16 | 73.75 / 72.36 | 1.044 | 0.542 |
| AVL | 34.94 | 38.85 | 74.05 / 72.52 | 74.19 / 72.91 | 1.050 | 0.540 |
| sum | 32.73 | 36.84 | 72.39 / 71.22 | 73.92 / 71.25 | 1.067 | 0.519 |
| concat | 32.76 | 36.45 | 72.99 / 71.81 | 73.60 / 71.94 | 1.085 | 0.510 |
| attention | 33.55 | 37.50 | 72.71 / 71.37 | 73.26 / 71.87 | 1.073 | 0.527 |
| **HiTNet (VAL)** | **35.26** | **39.22** | **74.12 / 72.66** | **74.53 / 73.10** | **1.043** | 0.539 |

Table 16: Performance comparison with SNR-based heuristic gating on the MOSI dataset across varying missing rates. Note: The lower MAE corresponds to superior results.

| Method | Acc-7 | Acc-5 | Acc-2 | F1 | MAE | Corr |
|---|---|---|---|---|---|---|
| HiTNet (SNR-Gated) | 33.27 | 37.06 | 73.02 / 71.97 | 73.19 / 72.08 | 1.072 | 0.534 |
| **HiTNet** | **35.26** | **39.22** | **74.12 / 72.66** | **74.53 / 73.10** | **1.043** | **0.539** |

### B.9 COMPUTATIONAL EFFICIENCY AND SCALABILITY ANALYSIS

To conduct an analysis of computational efficiency and scalability, we conducted a comparative analysis of inference efficiency for the baseline model LNLN, the full HiTNet model ($k = 3$), HiTNet with the activation sparsity set to the minimum/maximum values ($k = 1, k = 5$), and HiTNet without the semantic memory module (w/o SMM), and we report key metrics including parameter count, FLOPs, GPU memory, and latency. The results are summarized in Table 17.

In terms of resource consumption, HiTNet's total parameter count (93.43M) and total FLOPs (627.45G) are on par with those of LNLN (91.99M / 622.91G), with only a slight increase, indicating that our model does not introduce significant additional computational burden. Secondly, in terms of deployment feasibility, HiTNet's peak GPU memory consumption is only 1.846GB, nearly identical to the baseline model, demonstrating its excellent resource efficiency and deployment feasibility on standard hardware. Finally, regarding the performance-latency trade-off, HiTNet's average inference latency (62.72 ms) is slightly higher than LNLN's (46.00 ms). This additional overhead arises from the semantic memory module and sparse activation network in HiTNet, which enable more complex memory retrieval and fine-grained semantic modeling. This moderate increase in inference latency is a justified trade-off, as it comes with significant performance improvements and enhanced robustness when handling the challenging scenario of severe frame-level missingness.

Although sparse activation introduces additional computations, the resulting increase in inference latency is moderate and still well within real-time processing constraints. Specifically, the inference latency only increases by 0.24 ms when scaling the activation sparsity from the minimum ($k = 1$, 62.65 ms) to the maximum ($k = 5$, 62.89 ms). In addition, comparing the full HiTNet ($k = 3$) and the HiTNet without the Semantic Memory Module (w/o SMM), we observe only a minor difference in both parameter count and FLOPs. This confirms the lightweight nature and low overhead of the memory retrieval mechanism itself. These findings demonstrate that both the memory retrieval and sparse activation mechanisms are lightweight and efficient. The design principles of HiTNet inherently support robust scalability. Our modules ensure stable scalability to longer sequences and larger multimodal datasets by selectively retrieving only the most relevant memories and applying a sparse activation mechanism.

### B.10 QUALITATIVE ANALYSIS

To intuitively demonstrate the model's completion capability, we performed a rigorous qualitative analysis in the feature space on the MOSI dataset, taking the visual modality as an example. As

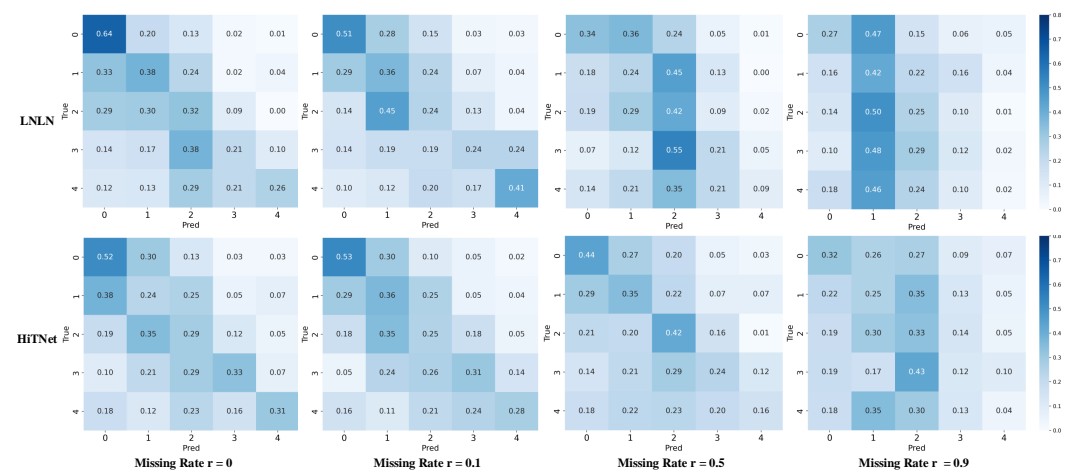

Figure 8: Confusion matrices of HiTNet and the baseline model LNLN on the SIMS dataset. Note: labels 0–4 denote negative, weakly negative, neutral, weakly positive, and positive, respectively.

Table 17: Comparative analysis of model inference efficiency.

| Evaluation Metric | LNLN | HiTNet ($k = 1$) | HiTNet ($k = 3$) | HiTNet ($k = 5$) | HiTNet (w/o SMM) |
|---|---|---|---|---|---|
| Params (M) | 91.99 | 93.43 | 93.43 | 93.43 | 93.38 |
| FLOPs (G) | 622.91 | 627.35 | 627.45 | 627.55 | 627.43 |
| GPU Memory (GB) | 1.840 | 1.846 | 1.846 | 1.846 | 1.845 |
| Latency (ms) | 46.00 | 62.65 | 62.72 | 62.89 | 61.59 |

shown in Figure 9, we visualize the feature sequence heatmap for a sample with a 0.2 missing rate, showing the missing visual modality features (a), the intra-modal completed features (b), and the original complete features (c), to demonstrate that our model achieves reliable semantic recovery in the feature space. The deep colored stripes in (a) clearly show the masked tokens, representing the loss of visual information. (b) shows that after processing by the Intra-modal Enhancement Stream, these originally empty stripes are effectively filled. Furthermore, compared to the original complete features in (c), (b) displays a highly consistent structural pattern. This visually proves that the Intra-modal Enhancement Stream successfully completes the lost underlying semantic patterns, thereby significantly restoring the data's semantic integrity within the feature space.

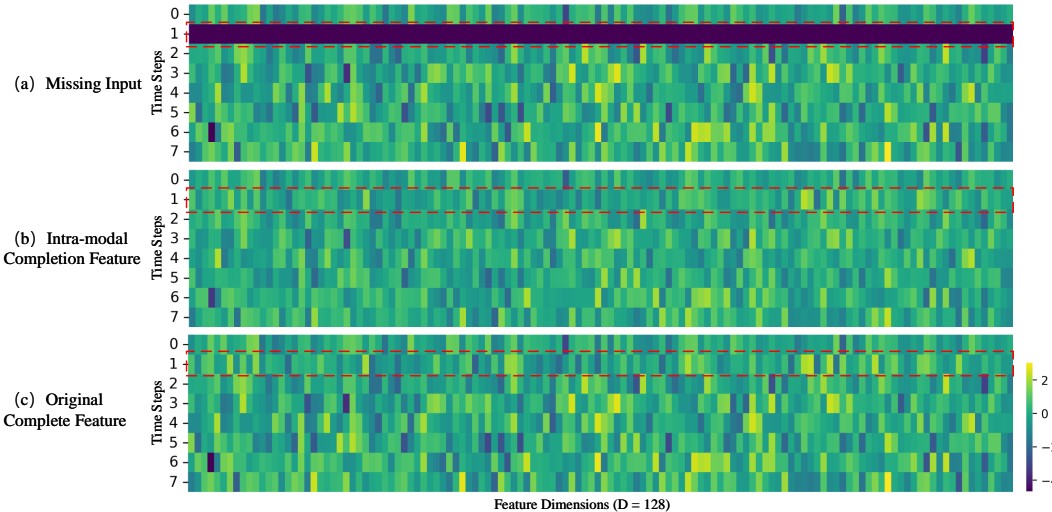

Figure 9: Qualitative analysis of Intra-modal semantic recovery in the feature space.

## B.11 Training Stability and Convergence Analysis

To more clearly illustrate the training dynamics of the model under multiple regularization losses, we trained the model with five independent random seeds and plotted the independent learning curves for all loss components, as shown in Figure 10.

The trajectories of all losses demonstrate an ideal convergence pattern. The loss exhibits a rapid descent in the initial training phase, followed by a smooth transition to the fine-tuning stage, and ultimately converges within a reasonable number of training steps. These results confirm that the multiple loss terms did not lead to training instability or noticeable degradation in training speed. Furthermore, to analyze the sensitivity of the training process to random initialization, we quantified the impact of different random seeds on the loss curves by calculating the standard deviation error band. The change in the error band highlights that while a wider fluctuation range is visible early on due to initialization differences, the band significantly narrows as training progresses, indicating that the model gradually eliminates the influence of randomness during convergence.

In conclusion, the training process of HiTNet is overall stable, with high convergence efficiency, low sensitivity to random seeds, and minimal loss fluctuations.

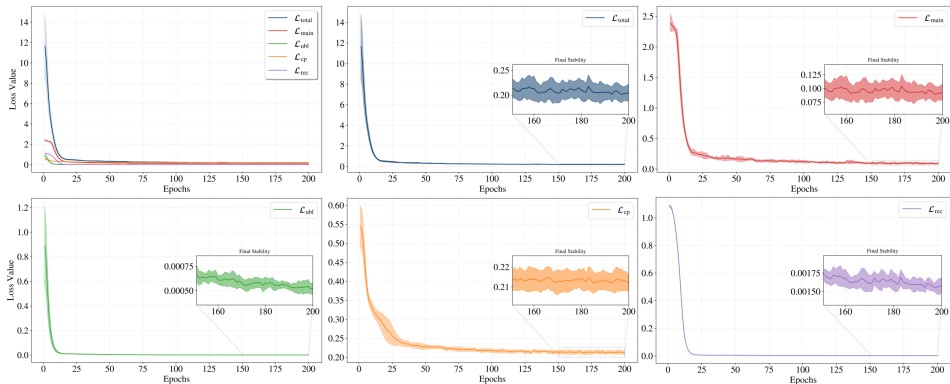

Figure 10: Training curves and seed sensitivity analysis of the HiTNet.

## B.12 Model Performance Analysis under Consecutive Frame-level Missingness

To evaluate the performance of HiTNet and the effectiveness of the Confidence Perception Module (CPM) under consecutive frame-level missingness, we modified the missingness settings on the MOSI dataset by replacing the random missingness mechanism with consecutive frame masking. Table 18 reports the results averaged across ten experimental settings with missing rates ranging from 0.0 to 0.9 for the baseline LNLN, HiTNet without CPM (w/o CPM), and the full HiTNet model.

The results demonstrate that the full HiTNet consistently outperforms the baseline LNLN across all metrics. For instance, it improves Acc-7 from 31.76% to 33.70% and reduces MAE from 1.085 to 1.066, confirming HiTNet's superior stability under severe data corruption. Furthermore, the critical role of the CPM is explicitly validated by the performance gap between the full model and the ablation variant. Notably, removing the CPM results in a performance drop, particularly in the Correlation metric, where the w/o CPM variant (0.505) falls to a level comparable to the baseline (0.506). In contrast, the full model achieves a significantly higher correlation of 0.525. This evidence confirms that the CPM is essential for gauging input reliability and mitigating the impact of unreliable frames, thereby ensuring robust performance in consecutive frame-level missingness scenarios.

Overall, HiTNet significantly outperforms the baseline model in consecutive frame-level missingness scenarios, and the Confidence Perception Module is verified to be a vital component for effective sentiment analysis under severe data corruption.

Table 18: Analysis of model performance under consecutive frame-level missingness. Note: The lower MAE corresponds to superior results.

| Method | Acc-7 | Acc-5 | Acc-2 | F1 | MAE | Corr |
|---|---|---|---|---|---|---|
| LNLN | 31.76 | 35.56 | 71.72 / 70.36 | 72.27 / 70.44 | 1.085 | 0.506 |
| HiTNet (w/o CPM) | 33.22 | 36.89 | 72.06 / 70.51 | 72.32 / 70.65 | 1.071 | 0.505 |
| **HiTNet** | **33.70** | **37.94** | **72.26 / 71.08** | **72.95 / 71.08** | **1.066** | **0.525** |

### B.13 QUANTITATIVE VERIFICATION OF THE BIOLOGICALLY INSPIRED MODULE

To further support our hippocampal–thalamic analogy, we have conducted quantitative verification of the Hippocampus-Inspired Module and Thalamus-Inspired Module.

**Quantitative Verification of the Hippocampus-Inspired Module.** The Semantic Memory Module (SSM) simulates hippocampal memory retrieval via a key–value retrieval mechanism. To visually demonstrate this functionality, we visualized the frame-level retrieval attention matrix within the SSM. This matrix shows, for each frame, the most relevant memory units activated by the model. As shown in Figure 11, the model still activates memory units for missing frames. More importantly, in the vertical direction, the memory activated by the missing frame is similar to that activated by the surrounding normal frames. This continuity in semantic trajectories demonstrates that the SSM successfully utilizes temporal context to retrieve the correct memory prototypes to fill in gaps, thereby verifying that the SSM effectively simulates hippocampal memory retrieval.

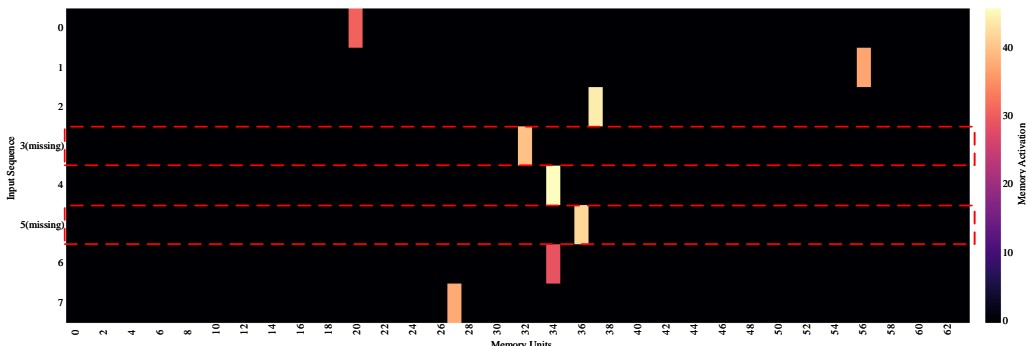

Figure 11: Visualization of frame-level retrieval attention in the semantic memory module.

**Quantitative Verification of the Thalamus-Inspired Module.** Our Confidence-Perception Module (CPM) simulates thalamic perceptual regulation by assigning different weights to each modality. To visually demonstrate that the CPM implements differential weighting, we analyzed the weights assigned to each modality on the test set. The results show that the model assigns different weights to the visual, audio, and language modalities (0.74, 0.71, and 0.81, respectively). Furthermore, to verify the effectiveness of this regulation, we conducted an ablation experiment in which all modality weights were forcibly set to 1, which implies the removal of CPM. The results are shown in Table 3 in the ablation study section. The performance degradation of this variant demonstrates the effectiveness of assigning different weights to each modality. Both the observed weight differences and the ablation results verify that CPM effectively simulated the perceptual regulatory function of the thalamus.

