# OpenReview forum: "HiTNet: Hippocampal-Thalamic Inspired Dual-Stream Network for Multimodal Sentiment Analysis under Missing Data"
_ICLR.cc/2026/Conference — Submitted to ICLR 2026_

### Official Review · Reviewer_Ek1D · 2025-10-27

**Soundness:** 2
**Presentation:** 3
**Contribution:** 3
**Rating:** 6
**Confidence:** 3

**Summary:**

This paper proposes Hippocampal–Thalamic dual-stream Network, a brain-inspired framework for multimodal sentiment analysis under frame-level missing data.
The authors draw inspiration from two functional mechanisms of the human brain:
(1) the hippocampal memory retrieval process, which reconstructs missing information through semantic association,
and (2) the thalamic perceptual regulation process, which integrates multisensory inputs while filtering unreliable cues.
Accordingly, HiTNet is composed of two complementary streams:
a hippocampal-inspired intra-modal enhancement stream that employs semantic memory and sparse activation modules to recover modality-specific semantics,
and a thalamic-inspired inter-modal regulation stream that utilizes confidence perception and adaptive cross-modal completion to integrate high-quality information across modalities.
A hierarchical fusion module combines both streams for sentiment prediction.

**Strengths:**

1. Effectiveness
Experiments on three standard benchmarks (MOSI, MOSEI, and SIMS) demonstrate consistent improvements of 1.5–2.0% over state-of-the-art methods across various missing rates,
and the model maintains 72.2% accuracy even under 90% missing data on MOSEI.
Ablation and visualization studies further support the dual-stream design’s effectiveness and robustness.

2. Appealing
The proposed idea is intuitively appealing. The analogy between hippocampal memory reconstruction and thalamic regulation provides an intuitive, biologically inspired rationale that enriches the interpretability of multimodal fusion. The proposed HiTNet introduces a biologically motivated architecture that integrates hippocampal-style memory retrieval and thalamic-style perceptual regulation. This dual-stream formulation is conceptually novel in the context of multimodal sentiment analysis and offers an interpretable way to address missing-data challenges.

3. Good writing
The manuscript is well organized, with a coherent flow from motivation to methodology and experiments. Figures and tables are informative and contribute to the clarity of presentation.

**Weaknesses:**

1. Quantitatively realized
How the “hippocampal” and “thalamic” analogies are quantitatively realized. While the hippocampal–thalamic analogy is conceptually interesting, the connection to actual neuroscientific mechanisms remains largely metaphorical.

2. Computational overhead and scalability
Computational overhead and scalability of the memory and activation modules. The memory retrieval and sparse activation modules may introduce additional computational overhead. The paper would benefit from a quantitative analysis of training/inference cost and scalability to larger datasets.

3. Minor concerns
Refer to the questions

4. Biased dataset performance discussion
While HiTNet performs strongly on MOSI/MOSEI, the performance margin on SIMS is relatively modest.

**Questions:**

1. Inconsistent terminology formatting
The paper inconsistently uses “intra-modal” and “inter-modal” in some places, while elsewhere they appear as “intramodal” and “intermodal.” Please standardize the terminology throughout the manuscript for consistency and readability.

2. Inconsistent expression of ‘missingness’
The manuscript alternates between “modality missingness” and “modality missing.” Since “missingness” is the correct nominal form referring to the state or rate of missing data, it should be used consistently.

3. Noun form correction
In a few instances, “missing” is used as a noun, which is grammatically suboptimal. It would be more precise to use “missingness” in these cases.

4. Tense consistency in reproducibility statement
The sentence “We have made every effort to ensure that the results presented in this paper are reproducible.” mixes present perfect with simple present.

---

> ### Author Response · Authors · 2025-11-27
> **Responses to Ek1D [1/3]**
>
> Thank you for your thorough and insightful evaluation.   We are very glad to receive your recognition of the significant performance improvement and conceptual novelty of our HiTNet architecture. In response to your suggestions, we have performed extensive supplementary analyses and experiments to rigorously address all points.
> Below, we provide our point-by-point responses.
>
>
> ## Response to Weaknesses:
> >**W1:** Quantitatively realized How the “hippocampal” and “thalamic” analogies are quantitatively realized. While the hippocampal–thalamic analogy is conceptually interesting, the connection to actual neuroscientific mechanisms remains largely metaphorical.
>
> **R1:** Thank you for your insightful comments. Following your suggestion, we have strengthened our justification from both **theoretical** and **experimental** perspectives.
>
> 1. **Theoretical Grounding:** First, we have added references to foundational computational neuroscience models in the introduction to strengthen the theoretical basis. We explicitly map the hippocampal function to Sparse Distributed Memory [1] and Hopfield Networks [2], clarifying that our architecture is a mathematical instantiation of these biological associative memory mechanisms.
>
> 2. **Experimental Verification:** In addition, we have conducted two targeted experiments on the MOSI dataset to justify how the hippocampal–thalamic analogy concretely guides the architecture design.
>
> 	- **(1) Quantitative Verification of the Hippocampus-Inspired Module.** The Semantic Memory Module (SSM) simulates hippocampal memory retrieval via a key–value retrieval mechanism. To visually demonstrate this functionality, we visualized the frame-level retrieval attention matrix within the SSM. This matrix shows, for each frame, the most relevant memory units activated by the model. As shown in **Figure 11** (in Appendix B.13), the model still activates memory units for missing frames. More importantly, in the vertical direction, the memory activated by the missing frame is similar to that activated by the surrounding normal frames. **This continuity in semantic trajectories demonstrates that the SSM successfully utilizes temporal context to retrieve the correct memory to fill in gaps, thereby verifying that the SSM effectively simulates hippocampal memory retrieval.**
>
>
> 	- **(2) Quantitative Verification of the Thalamus-Inspired Module.** Our Confidence-Perception Module (CPM) simulates thalamic perceptual regulation by assigning different weights to each modality. To visually demonstrate that the CPM implements differential weighting, we analyzed the weights assigned to each modality on the test set. The results show that **the model assigns different weights to the visual, audio, and language modalities** (0.74, 0.71, and 0.81, respectively). Furthermore, to verify the effectiveness of this regulation, we conducted an ablation experiment in which all modality weights ($s_m$) were forcibly set to 1, as shown in **Table R1**. The performance degradation of this variant demonstrates the effectiveness of assigning different weights to each modality. **Both the observed weight differences and the ablation results verify that CPM effectively simulated the perceptual regulatory function of the thalamus.**
>
> We have revised the manuscript to incorporate these details into the **Introduction** and **Appendix** **B.13**.
>
> References:
>
> [1]Pentti Kanerva. Sparse Distributed Memory. MIT Press, Cambridge, MA, USA, 1988. ISBN 0262111322.
>
> [2] J J Hopfield. Neural networks and physical systems with emergent collective computational abilities. Proceedings of the National Academy of Sciences, 79(8):2554–2558, 1982. doi: 10.1073/pnas. 79.8.2554
>
> **Table R1:** Effectiveness evaluation of the Thalamus-Inspired Module. Note: The lower MAE corresponds to superior results.
>
> | Method           |   Acc-7   |   Acc-5   |       Acc-2       |        F1         |    MAE    |   Corr    |
> | :--------------- | :-------: | :-------: | :---------------: | :---------------: | :-------: | :-------: |
> | HiTNet ($s_m=1$) |  34.87    |  38.57    |  73.72 / 72.08    |  74.48 / 72.51    |  1.044    |  0.531    |
> | **HiTNet**       | **35.26** | **39.22** | **74.12 / 72.66** | **74.53 / 73.10** | **1.043** | **0.539** |

---

> ### Author Response · Authors · 2025-11-27
> **Responses to Ek1D [2/3]**
>
> >**W2:** Computational overhead and scalability Computational overhead and scalability of the memory and activation modules. The memory retrieval and sparse activation modules may introduce additional computational overhead. The paper would benefit from a quantitative analysis of training/inference cost and scalability to larger datasets.
>
> **R2:** Thank you for your insightful comment regarding the computational overhead and scalability of the memory retrieval and sparse activation modules. In response, we conducted a comparative analysis of inference efficiency for the baseline model LNLN, the full HiTNet model (k=3), HiTNet with the sparse activation set to the minimum/maximum values (k=1, k=4), and HiTNet without the semantic memory module (w/o SMM),and we report key metrics including parameter count, FLOPs, GPU memory, and latency. The results are summarized in **Table R3**.
>
>  **(1) The full HiTNet model (k=3) introduces only a modest increase in overhead compared to the baseline model.** In terms of resource consumption, HiTNet's total parameter count (93.43M) and total FLOPs (627.45G) are on par with those of LNLN (91.99M / 622.91G), with only a slight increase, **indicating that our model does not introduce significant additional computational burden.**
>
> Secondly, in terms of deployment feasibility, HiTNet's peak GPU memory consumption is only 1.846GB, nearly identical to the baseline model, **demonstrating its excellent resource efficiency and deployment feasibility on standard hardware.**
>
> Finally, regarding the performance-latency trade-off, HiTNet’s average inference latency (62.72 ms) is slightly higher than LNLN's (46.00 ms). This additional overhead arises from the semantic memory module and sparse activation network in HiTNet, which enable more complex memory retrieval and fine-grained semantic modeling. **We believe this modest increase in latency is a justified trade-off, as it comes with significant performance improvements and enhanced robustness when handling the challenging scenario of severe frame-level missingness.**
>
> **(2) Lightweight design of memory retrieval and sparse activation.** Although sparse activation introduces additional computations, the resulting increase in inference latency is moderate and still well within real-time processing constraints.
>
> Specifically, the inference latency only increases by 0.24 ms when scaling the activation sparsity from the minimum ($k=1$, 62.65 ms) to the maximum ($k=5$, 62.89 ms).
> In addition, comparing the full HiTNet (with $k=3$) and the HiTNet without the Semantic Memory Module (w/o SMM), we observe only a minor difference in both parameter count and FLOPs. **This confirms the lightweight nature and low overhead of the memory retrieval mechanism itself.**
> **These findings demonstrate that both the memory retrieval and sparse activation mechanisms are lightweight and efficient.**
>
> **(3) Potential for scalability to larger datasets.** The design principles of HiTNet inherently support robust scalability. **Our modules ensure stable scalability to longer sequences and larger multimodal datasets** by selectively retrieving only the most relevant memories and applying a sparse activation mechanism.
>
> **In summary, this analysis validates that HiTNet maintains strong efficiency and exhibits excellent scalability while delivering substantial performance gains.**
>
> We have revised the paper and added these details to **Appendix B.9**.
>
> **Table R2:** Comparative analysis of model inference efficiency.
>
> | Evaluation Metric | LNLN   | HiTNet (k=1) | HiTNet (k=3) | HiTNet (k=5) | HiTNet (w/o SMM) |
> |------------------|--------|---------------|---------------|---------------|------------------|
> | Params (M)        | 91.99 | 93.43         | 93.43         | 93.43         | 93.38            |
> | FLOPs (G)         | 622.91| 627.35        | 627.45        | 627.55        | 627.43           |
> | GPU Memory (GB)   | 1.840 | 1.846         | 1.846         | 1.846         | 1.845            |
> | Latency (ms)      | 46.00 | 62.65         | 62.72         | 62.89         | 61.59            |

---

> ### Author Response · Authors · 2025-11-27
> **Responses to Ek1D [3/3]**
>
> >**W3:** Biased dataset performance discussion While HiTNet performs strongly on MOSI/MOSEI, the performance margin on SIMS is relatively modest.
>
> **R3:** Thank you for the careful observation regarding the model’s cross-dataset performance differences.  HiTNet demonstrates clear advantages on MOSI/MOSEI, but its performance gains on the SIMS dataset are relatively modest. This can be attributed to two main factors:
>
> 1. **Intrinsic Data Distribution and Scope.** The SIMS dataset (a Chinese multimodal benchmark) differs significantly from the large-scale English datasets (MOSI/MOSEI) in terms of emotional expression conventions and scene types. More importantly, SIMS suffers from **inherent data imbalance and** **a smaller overall scale**. The scarcity of samples in certain emotion categories limits the availability of sufficient data for HiTNet to fully leverage its complex multimodal enhancement mechanisms.
>
> 2. **Architectural Complexity vs. Dataset Scale.** The core design of HiTNet, including the **Memory Module** and **Sparse Gating mechanisms**, is intended to efficiently handle long-range dependencies and inherent feature heterogeneity in **large-scale datasets**. However, the **relatively limited scale of SIMS** makes it challenging to fully train and activate the potential of these complex, high-capacity components. Consequently, the model's advantages are most pronounced in high-complexity, large-scale tasks such as MOSI/MOSEI, where its architecture is fully utilized.
>
>
> ## Response to Questions:
> >**Q1:** Inconsistent terminology formatting The paper inconsistently uses “intra-modal” and “inter-modal” in some places, while elsewhere they appear as “intramodal” and “intermodal.” Please standardize the terminology throughout the manuscript for consistency and readability.
>
> >**Q2:** Inconsistent expression of ‘missingness’ The manuscript alternates between “modality missingness” and “modality missing.” Since “missingness” is the correct nominal form referring to the state or rate of missing data, it should be used consistently.
>
> >**Q3:** Noun form correction In a few instances, “missing” is used as a noun, which is grammatically suboptimal. It would be more precise to use “missingness” in these cases.
>
> >**Q4:** Tense consistency in reproducibility statement The sentence “We have made every effort to ensure that the results presented in this paper are reproducible.” mixes present perfect with simple present.
>
> **R4:** Thank you for your valuable suggestions regarding the consistency of terminology and expressions. We have made uniform revisions throughout the manuscript:
>
> 1. **Terminology standardization:** We have unified “intra-modal”/“intramodal” and “inter-modal”/“intermodal” as “intra-modal” and “inter-modal” throughout the text to ensure consistency and readability.
>
> 2. **Consistency in expressing missingness:** All references to modality incompleteness now consistently use the nominal form “modality missingness”, replacing the previously mixed usage of “modality missing.” Following your suggestion, “missingness” is used whenever the term functions as a noun.
>
> 3.  **Tense consistency:** Following your recommendation, the original sentence “We have made every effort to ensure that the results presented in this paper are reproducible.” has been revised to the tense-consistent form “We make every effort to ensure that the results presented in this paper are reproducible.”

---

> > ### Comment · Reviewer_Ek1D · 2025-11-27
> > **Thank you for your response and clarifications.**
> >
> > Thank you for your response and clarifications. I appreciate the detailed follow-up, and I do think this is a valuable piece of work with clear potential impact. Overall, I will maintain my original overall score.

---

> > > ### Author Response · Authors · 2025-11-28
> > >
> > > We greatly appreciate the valuable time you dedicated to the entire review process.
> > >
> > > We deeply appreciate your recognition that this work has "clear potential impact," and we are very pleased that our clarifications successfully addressed your concerns.
> > >
> > > Given that you have positively affirmed the value of the paper, we would be deeply grateful if you could reconsider upgrading your rating.
> > >
> > > Best wishes!

---

### Official Review · Reviewer_dS1k · 2025-10-31

**Soundness:** 3
**Presentation:** 4
**Contribution:** 3
**Rating:** 6
**Confidence:** 5

**Summary:**

This paper introduces HiTNet, a brain-inspired dual-stream network for MSA with missing data. The HiTNet includes an intra-modal enhancement stream that reconstructs missing features using semantic memory, and an inter-modal regulation stream that adaptively fuses reliable information across modalities. Experiments on MOSI, MOSEI, and SIMS show that HiTNet achieves SOTA accuracy, demonstrating its effectiveness.

**Strengths:**

1. The method is well-motivated and the overall model design is reasonable.
2. The experiments and analyses are comprehensive, demonstrating the method’s effectiveness.
3. The model achieves SOTA performance across multiple datasets.
4. The paper is well-written and easy to read.

**Weaknesses:**

1. How is the Top-K value in the Semantic Memory Module selected? Could the authors discuss the impact of different k values on performance?
2. The proposed method involves sparse activation across multiple modules. Could the authors analyze what would happen if full activation were used instead?
3. Could you add explanations for G and g in the caption of Figure 2, or include a legend directly in the figure for clarity.
4. The paper claims that Hierarchical Fusion provides better performance. Could you compare it against simpler alternatives such as concatenation, summation, or attention-based fusion?
5. Since the optimization objective includes many constraint terms, could this lead to unstable or slower training? Could you show training curves and discuss convergence behavior? In addition, how sensitive is the training process to random seed variations Does the loss fluctuate significantly?
6. In section relevant work, could you introduce and discuss all the methods you compared?

**Questions:**

Please see Weaknesses.

---

> ### Author Response · Authors · 2025-11-27
> **Responses to dS1k [1/3]**
>
> Thank you for your valuable feedback.  We sincerely appreciate your recognition of the motivation and overall design of our HiTNet framework, as well as your acknowledgement of our SOTA performance and comprehensive analyses.  In response, we have performed extensive quantitative ablation studies and comprehensive supplementary analyses to thoroughly address each point.  Below, we provide our point-by-point responses.
>
> ## Response to Weaknesses:
> >**W1:** How is the Top-K value in the Semantic Memory Module selected? Could the authors discuss the impact of different k values on performance?
>
> >**W2:** The proposed method involves sparse activation across multiple modules. Could the authors analyze what would happen if full activation were used instead?
>
> **R1:** Thank you for your valuable suggestions. We have unified the analysis of these two questions by conducting a comprehensive sensitivity study on the sparse activation mechanism, encompassing both the impact of $k$ values and a comparison with the full activation baseline ($k=n$).
>
> **We designed two sets of comparative experiments:**
> - **First,** we replaced the entire sparse activation network module with a single sub-network ($n=k=1$) to verify the impact of the sparse activation network on model performance.
> - **In addition,** To verify the model's sensitivity to the $k$ value within the sparse activation mechanism, we fixed the total number of sub-networks at $n=5$ and retrained the model with varying $k$ values. We conducted experiments across all missing rates and calculated the average performance.
>
> **The results on the MOSI dataset are presented in** **Table R1.**
> - **Single-Pathway Limitation ($n=k=1$):** When the sparse activation network degenerates into a single sub-network ($n=k=1$), the performance significantly declines **This indicates that multiple sub-networks provide diverse semantic enhancement pathways, which are crucial for capturing richer intra-modal feature patterns.**
>
> - **Optimal Sparsity ($k=3$):** As shown in the table, model performance consistently improves as $k$ increases from 1, peaking at $k=3$. **This demonstrates that a moderate level of sparsity successfully integrates multi-perspective semantic information without sacrificing precision.**
>
> - **Consequences of Full Activation ($k=5$):** Crucially, the performance declines sharply when full activation ($k=5$) is used. **The full activation strategy eliminates the selective power of the Top-$k$ mechanism, introducing a substantial amount of redundant**, sample-irrelevant noise into the enhancement process. This dilutes the valuable semantic signal, diminishing the core advantage of sparse selection.
>
> This comprehensive analysis validates that the sparse activation mechanism plays a crucial role in filtering noise and maximizing semantic relevance. The optimal performance at $k=3$ confirms that sparse selection is essential for achieving robust intra-modal enhancement.
>
> We have added the above experiment to **Appendix** **B.2**.
>
> **Table R1:** Sensitivity analysis of the Top-$k$ hyperparameter in the sparse activation mechanism on the MOSI dataset. The lower MAE corresponds to superior results.
>
> | Method        | Acc-7     | Acc-5     | Acc-2             | F1                | MAE       | Corr      |
> | ------------- | --------- | --------- | ----------------- | ----------------- | --------- | --------- |
> | n=k=1         | 34.88     | 38.85     | 73.22 / 72.15     | 73.71 / 72.55     | 1.054     | 0.531     |
> | n=5,k=1       | 34.03     | 37.42     | 73.25 / 72.09     | 73.63 / 72.28     | 1.067     | 0.537     |
> | n=5,k=2       | 34.73     | 38.68     | 73.50 / 72.25     | 73.84 / 72.65     | 1.050     | 0.538     |
> | **n=5,k=3**   | **35.26** | **39.22** | **74.12 / 72.66** | **74.53 / 73.10** | **1.043** | 0.539     |
> | n=5,k=4       | 33.72     | 37.53     | 73.58 / 72.11     | 73.97 / 72.85     | 1.057     | **0.541** |
> | n=k=5         | 33.22     | 37.36     | 73.10 / 72.16     | 73.25 / 72.24     | 1.067     | 0.536     |

---

> ### Author Response · Authors · 2025-11-27
> **Responses to dS1k [2/3]**
>
> >**W3:** Could you add explanations for G and g in the caption of Figure 2, or include a legend directly in the figure for clarity.
>
> **R3:** We appreciate the suggestion to enhance the readability of our figure. We have updated **Figure 2** by adding a legend that clearly specifies the meanings of $G$ and $g$.
> Specifically,  $G$ denotes the Sparse Gating Scores, which determine the selection weights for the sub-networks in the Sparse Activation Network (SAN).
> $g$ represents the Residual Fusion Gate, which controls the adaptive integration ratio of the retrieved memory in the Semantic Memory Module (SMM).
>
>
>
> >**W4:** The paper claims that Hierarchical Fusion provides better performance. Could you compare it against simpler alternatives such as concatenation, summation, or attention-based fusion?
>
> **R4:** Thank you for your valuable suggestion. To validate the effectiveness of our proposed hierarchical fusion strategy, we conducted additional experiments comparing it with simpler alternatives, including summation (Sum), concatenation (Concat), and attention-based fusion (Attention).
>
> For a fair comparison, we conducted experiments for all missing rates for each fusion method on the MOSI dataset and report the average results, as shown in **Table R2**.
>
> The results show that all simpler fusion variants result in a significant drop in performance. This indicates that **simple fusion methods cannot adequately capture the hierarchical dependencies between modalities.**
>
> In contrast, hierarchical fusion more effectively facilitates the interaction of intra-modal and inter-modal complementary features, thereby improving model robustness and performance.
>
> We have revised the paper and added the above-mentioned experiments and analyses to **Appendix B.6**.
>
> **Table R2:** Performance comparison of different multimodal fusion strategies. Note: The lower MAE corresponds to superior results.
>
> | Method     | Acc-7     | Acc-5     | Acc-2             | F1                | MAE       | Corr      |
> | ---------- | --------- | --------- | ----------------- | ----------------- | --------- | --------- |
> | sum        | 32.73     | 36.84     | 72.39 / 71.22     | 73.92 / 71.25     | 1.067     | 0.519     |
> | concat     | 32.76     | 36.45     | 72.99 / 71.81     | 73.60 / 71.94     | 1.085     | 0.510     |
> | attention  | 33.55     | 37.50     | 72.71 / 71.37     | 73.26 / 71.87     | 1.073     | 0.527     |
> | **HiTNet** | **35.26** | **39.22** | **74.12 / 72.66** | **74.53 / 73.10** | **1.043** | **0.539** |

---

> ### Author Response · Authors · 2025-11-27
> **Responses to dS1k [3/3]**
>
> >**W5:** Since the optimization objective includes many constraint terms, could this lead to unstable or slower training? Could you show training curves and discuss convergence behavior? In addition, how sensitive is the training process to random seed variations Does the loss fluctuate significantly?
>
> **R5:** Thank you for your constructive feedback. **In response to your concerns regarding training stability and convergence behavior, we conducted a comprehensive analysis by training the full HiTNet architecture over five independent random seeds** and added full training-curve visualizations in **Figure 10** (Appendix B.11).
>
> We separately plotted individual curves for the total loss and all individual constraint objectives. The figures reveal the following key findings:
>
> 1. **Robust Convergence Despite Multiple Objectives.** The multiple loss terms did not lead to training instability or noticeable degradation in training speed. Specifically, the trajectories of all losses demonstrate an ideal convergence pattern. **The loss exhibits a rapid descent in the initial training phase, followed by a smooth transition to the fine-tuning stage, and ultimately converges within a reasonable number of training steps.**
>
> 2. **Insensitivity to Random Seed Initialization.** To analyze the model's sensitivity to random initialization, we quantified the variation using a standard deviation error band. The change in the error band highlights that while a wider fluctuation range is visible early on due to initialization differences, the error band significantly narrows as training progresses. **This demonstrates that the optimization trajectory is robust, with the model consistently eliminating the influence of initial randomness during convergence.**
>
> **In conclusion, the above analysis demonstrates that despite incorporating multiple constraint loss terms, the training process is overall stable, converges efficiently, is insensitive to random seeds, and exhibits low loss fluctuation.**
>
> We have revised the article and added the above-mentioned experiments and analyses to **Appendix B.11**.
>
>
> >**W6:** In section relevant work, could you introduce and discuss all the methods you compared?
>
> **R6:** Thank you for your valuable suggestion. Following your advice, we have revised the **Related Work** section of the paper to provide a more comprehensive introduction and discussion of all the methods compared in this work.
>
> **Specifically, we introduce MISA, Self-MM, MMIM, CENET, TETFN, TFR-Net, ALMT, LNLN, and P-RMF, and briefly describe their core mechanisms and design motivations. We also highlight their limitations in scenarios with incomplete or missing data, which motivates the design of our approach.**
>
> Once again, we sincerely appreciate your suggestion. These revisions help readers better understand the relationship between our method and existing approaches, as well as the improvements we achieve.

---

> > ### Comment · Reviewer_dS1k · 2025-11-27
> >
> > Thank you for your detailed reply. However, it seems that the revised version has not been uploaded. Could you upload it?

---

> > > ### Author Response · Authors · 2025-11-27
> > >
> > > Thank you very much for your reply. I have already uploaded the revised version. Please check again, and let me know if you encounter any issues accessing it.

---

> > > > ### Comment · Reviewer_dS1k · 2025-11-28
> > > >
> > > > I have carefully read the blue-revised paragraph and believe that the quality of the paper has significantly improved. All of my concerns have been addressed. Thank you.
> > > >
> > > > In addition, because ICLR has closed the edit window, I will wait further notification and will consider raising my rating.

---

> > > > > ### Author Response · Authors · 2025-11-28
> > > > >
> > > > > We sincerely thank you for taking the time to thoroughly review our revised manuscript.
> > > > >
> > > > > We are delighted to see that you feel the quality of the paper has significantly improved and that all of your concerns have been successfully addressed.
> > > > >
> > > > > We understand the question of the edit window and deeply appreciate your consideration regarding a rating upgrade.
> > > > >
> > > > > We very much look forward to seeing an improvement in your rating.

---

### Official Review · Reviewer_gTuV · 2025-11-04

**Soundness:** 2
**Presentation:** 2
**Contribution:** 2
**Rating:** 4
**Confidence:** 3

**Summary:**

The paper targets multimodal sentiment analysis under frame-level, asynchronous missing across text, audio, and visual streams. It introduces HiTNet, a hippocampal–thalamic inspired dual-stream architecture: an intra-modal (hippocampal) path that performs memory-based completion via a learnable semantic memory and sparse activation/routing to exploit residual evidence within each modality, and an inter-modal (thalamic) path that estimates per-modality confidence and performs confidence-gated cross-modal completion to import only reliable cues. The streams are hierarchically fused, with auxiliary reconstruction and regularizers for routing balance and confidence calibration. On MOSI, MOSEI, and SIMS, HiTNet consistently improves Acc/F1 and MAE/Correlation over strong baselines and shows slower degradation up to 90% missing, indicating robust reliability modeling. Ablations verify the necessity of semantic memory, confidence estimation, and the two-stream design.

**Strengths:**

1. Propose a dual-stream (hippocampus/thalamus) framework that decouples "intra-modal self-completion" and "inter-modal confidence regulation," resulting in clear and reusable functional boundaries.
2. Focusing on more realistic and common frame-level asynchronous missing data scenarios, rather than just complete modality absence, makes the research more meaningful.
3. By utilizing semantic memory and sparse routing, the paper first extracts all available evidence from the current modality, reducing over-reliance on other modalities and resulting in greater robustness.

**Weaknesses:**

1. This paper does not compare its method with some powerful modern imputation methods such as masked-autoencoding and diffusion-based completion. Also missing are ablations against simpler reliability heuristics (e.g., SNR/entropy-based gates). These methods may seem more suitable for scenarios where frames are missing.
2. The input with missing frames does not simulate real-world scenarios, such as consecutive frame drops due to packet loss. Conducting experiments under such conditions would be more convincing.
3. By coupling the intra- and inter-modal streams only via a late, single-point gate, the model limits cross-layer interactions and can miss fine-grained synergies that require earlier or multi-level fusion.
4. The convex mix $s⋅x+(1−s)⋅h$ assumes additive compatibility between native and completed features, risking blurred signals and underfitting of nonlinear cross-modal interactions.

**Questions:**

1. Could you provide an explanation for the performance of HiTNet and w/o $L_{ubl}$ in Table 3 in terms of acc-7 and acc-5?
2. Why couple the intra- and inter-modal streams only via a single late gate—did you evaluate earlier or multi-level fusion (e.g., MoE or cross-layer gating)?
3. Does using a single prompt token for each modality in cross-modal completion methods result in insufficient expressive power?
4. How do you handle the $O(T^2)$ complexity of attention mechanisms in long sequence scenarios (>1-5 minutes)?

---

> ### Author Response · Authors · 2025-11-27
> **Responses to gTuV [1/5]**
>
> Thank you for your valuable feedback. We sincerely appreciate your recognition of the robustness and functional clarity achieved by our dual-stream framework. Your critique is constructive, pointing out key areas regarding modern baselines and theoretical justification of fusion. We have addressed all concerns with new analyses and supplementary experiments. We provide our point-by-point responses below.
>
> ## Response to Weaknesses:
> >**W1:** This paper does not compare its method with some powerful modern imputation methods such as masked-autoencoding and diffusion-based completion. Also missing are ablations against simpler reliability heuristics (e.g., SNR/entropy-based gates). These methods may seem more suitable for scenarios where frames are missing.
>
> **R1:** We sincerely appreciate your suggestion regarding the comparison with modern imputation methods and reliability heuristics. We have adopted your suggestions and incorporated revisions and supplements from two primary perspectives:
>
> 1. **Discussion on Modern Generative Imputation (MAE & Diffusion).** We have expanded our discussion in the Related Work to clarify the distinction between our method and generative approaches like Masked Autoencoders and Diffusion Models.
>
> 	- **Reconstruction vs. Discrimination:** While MAE [1,2] and Diffusion models [3] excel at pixel/waveform-level reconstruction, they are optimized to approximate the global data distribution, including background noise and task-irrelevant details. For sentiment analysis, exact signal reconstruction is computationally expensive and often unnecessary.
>
> 	- **Semantic Focus:** In contrast, HiTNet is a discriminative framework. It does not aim to fill in pixels but to maximize signal validity for the downstream classification task. By explicitly estimating confidence scores, our method focuses on recovering semantic integrity rather than physical signal integrity, avoiding the noise introduction often seen when generative models hallucinate details in highly corrupted segments.
>
> 2. **Comparative Experiments with Heuristic Gating (SNR).** To validate the necessity of our learnable confidence module against simpler heuristics, we conducted an ablation study using Signal-to-Noise Ratio (SNR) as a hard gating criterion.
>
> 	- **Experimental Setup:** We implemented a variant, HiTNet (SNR-Gated), which gates modalities based on raw frame-level SNR values. We evaluated performance on the MOSI dataset across all missing rates (0.0 to 0.9).
>
> 	- **Results Analysis:** As shown in **Table R1**, our proposed method consistently outperforms the SNR-gated variant. This finding provides strong evidence that heuristic metrics relying solely on raw signals reflect only physical integrity, which is insufficient for accurately assessing intrinsic information value. For instance, high-noise segments may still contain critical semantic cues, while clear signals might be redundant/uninformative. **By learning context-aware value estimation scores, our method enables more precise selective rectification, thereby achieving superior performance.**
>
> We have added the corresponding discussions and experimental results to the **Related Work** and **Appendix** **B.7** of the revised manuscript.
>
> References:
>
> [1] Zhan Tong, Yibing Song, Jue Wang, and Limin Wang. VideoMAE: Masked autoencoders are data-efficient learners for self-supervised video pre-training. In Alice H. Oh, Alekh Agarwal, Danielle Belgrave, and Kyunghyun Cho (eds.), Advances in Neural Information Processing Systems, 2022.
>
> [2] Jiaxiang Dong, Haixu Wu, Haoran Zhang, Li Zhang, Jianmin Wang, and Mingsheng Long. SimMTM: A simple pre-training framework for masked time-series modeling. In Thirty-seventh Conference on Neural Information Processing Systems, 2023.
>
> [3] Juan Lopez Alcaraz and Nils Strodthoff. Diffusion-based time series imputation and forecasting with structured state space models. Transactions on Machine Learning Research, 2023. ISSN 2835-8856
>
> **Table R1:** Performance comparison with SNR-based heuristic gating on the MOSI dataset (Average results across missing rates 0.0-0.9). Note: The lower MAE corresponds to superior results.
>
> | Method | Acc-7 | Acc-5 | Acc-2 | F1 | MAE | Corr |
> | :--- | :---: | :---: | :---: | :---: | :---: | :---: |
> | HiTNet (SNR-Gated) | 33.27 | 37.06 | 73.02 / 71.97 | 73.19 / 72.08 | 1.072 | 0.534 |
> | **HiTNet** | **35.26** | **39.22** | **74.12 / 72.66** | **74.53 / 73.10** | **1.043** | **0.539** |

---

> ### Author Response · Authors · 2025-11-27
> **Responses to gTuV [2/5]**
>
> >**W2:** The input with missing frames does not simulate real-world scenarios, such as consecutive frame drops due to packet loss. Conducting experiments under such conditions would be more convincing.
>
> **R2:** We sincerely appreciate this constructive suggestion. In strict accordance with your suggestion, we implemented **a consecutive frame-level missingness** experiment on the MOSI dataset to mimic the specific packet loss and buffering scenarios you mentioned.
>
> Specifically, we varied the missing rate from 0.0 to 0.9 to cover a wide range of severity. **To validate the model's performance, we designed experiments from two aspects: comparison with the baseline model and analysis of the effectiveness of the confidence perception module.**  **Table R2** reports the average results of different missing rates for the baseline LNLN, HiTNet without CPM (w/o CPM), and the full HiTNet model.
>
> 1. **Superior Robustness under consecutive frame-level missingness scenarios.** The data demonstrates that HiTNet maintains superior performance compared to the baseline even under these challenging conditions. HiTNet consistently outperforms the baseline LNLN across all evaluation metrics. For example, in terms of Acc-7, HiTNet achieves 33.70, significantly surpassing the baseline's 31.76. Our model also achieves a lower Mean Absolute Error (MAE) of 1.066 compared to the baseline's 1.085.
>
> 2. **The Critical Role of Confidence Perception.** To further validate the model's internal mechanism in this realistic setting, we analyzed the contribution of the Confidence Perception Module (CPM). The ablation results show that the CPM is crucial for handling consecutive frame drops. When the CPM is removed ($w/o$ CPM), we observe a notable performance drop, particularly in the Correlation (Corr) metric, which falls to a level comparable to the baseline (0.506). In contrast, the full model achieves a significantly higher correlation of 0.525. **These results demonstrate that CPM is an indispensable component for accurately estimating input reliability, enabling HiTNet to maintain strong robustness even under challenging consecutive frame-missing conditions.**
>
> Overall, the experiment simulates real-world scenarios, such as consecutive frame drops due to packet loss, further demonstrating HiTNet's robustness and effectiveness in practical applications.
>
> We have revised the paper and added the above experiments to **Appendix B.12**.
>
> **Table R2:** Analysis of model performance under consecutive frame-level missingness. Note: The lower MAE corresponds to superior results.
>
> |Method|Acc-7|Acc-5|Acc-2|F1|MAE|Corr|
> |---|---|---|---|---|---|---|
> |LNLN|31.76|35.56|71.72 / 70.36|72.27 / 70.44|1.085|0.506|
> |HiTNet (w/o CPM)|33.22|36.89|72.06 / 70.51|72.32 / 70.65|1.071|0.505|
> |**HiTNet**|**33.70**|**37.94**|**72.26 / 71.08**|**72.95 / 71.08**|**1.066**|**0.525**|

---

> ### Author Response · Authors · 2025-11-27
> **Responses to gTuV [3/5]**
>
> >**W3:** By coupling the intra- and inter-modal streams only via a late, single-point gate, the model limits cross-layer interactions and can miss fine-grained synergies that require earlier or multi-level fusion.
>
> **R3:** We appreciate your insightful comment. Regarding the concerns about the coupling mechanism between the Intra-stream and Inter-modal stream, we would like to address this from two perspectives: mechanism clarification and design philosophy.
>
> 1. **Fine-grained Synergy via Token-level Fusion.** Although our fusion happens at a later stage, it is not limited to coarse-grained interactions.
>
> 	**Deep Cross-Transformer vs. Gating:** The coupling mechanism is based on a deep Cross-Transformer ($E^C$), rather than a simple confidence gate. The scalar gate is only for modulation, while the actual feature integration occurs via Multi-Head Cross-Attention.
>
> 	**Dense Token-Level Interaction:** Crucially, this Transformer operates on **frame-level feature sequences** (Token-level). This implies that the outputs of the two streams undergo dense, non-linear integration where every frame interacts with the cross-modal context. This ensures that the model fully captures the fine-grained synergies between the Intra-modal stream (modality-specific features) and the Inter-modal stream (consistency features), effectively addressing the concern of missing detailed interactions.
>
> 2. **Early fusion in frame-level missing scenarios poses a risk of introducing noise.**
>
> 	**Avoiding Noise Propagation:** If we fuse corrupted/missing features with clean features in early layers, the noise from the missing modality propagates through the network, destroying the fine-grained semantic integrity of the good modality.
>
> 3. **Decoupling Specificity and Consistency: Our parallel-then-fuse strategy is a deliberate design choice aimed at decoupling specificity and consistency.** The Intra-modal stream is designed to mine modality specificity, leveraging residual information within the modality for self-completion to prevent assimilation by other modalities. The Inter-modal stream aims to capture inter-modal consistency, utilizing cross-modal cues to recover missing information. We argue that coupling earlier or via cross-layer interactions would lead to premature information exchange, resulting in the loss of modality-specific information. **Maintaining parallel processing forces the network to learn these two sets of complementary feature representations, thereby achieving more robust prediction results during the final Cross-Transformer fusion.**
>
>
>
> >**W4:** The convex mix assumes additive compatibility between native and completed features, risking blurred signals and underfitting of nonlinear cross-modal interactions.
>
> **R4:** We appreciate your valuable comment. We would like to clarify the concerns regarding the risks of insufficient additive compatibility and nonlinear underfitting in the convex mix between native features ($x_m$) and completed features ($h_m$) from the following two perspectives:
>
> 1. **Semantic Anchors ensure latent space compatibility.** As described in the paper (Eq. 9), we introduce a modality-specific learnable prompt, $h^0_m$, serving as a semantic anchor. This mechanism forces the Transformer to map information from source modalities onto a latent manifold consistent with the target modality $m$. During the Self-Attention interaction process, $h^0_m$ leverages its learnable nature to actively aggregate relevant cross-modal information from the context. Consequently, $h_m$ is explicitly aligned with the feature distribution space of $x_m$ during the generation process. Given this alignment, the additive operation is mathematically valid and preserves semantic consistency, thereby avoiding the risk of signal blurring.
>
> 2. **We adopt a strategy of decoupling cross-modal interaction from fusion.** The nonlinear cross-modal modeling is entirely handled by the CCM module (Eq. 9), which is based on a multi-layer Transformer Encoder and possesses strong contextual modeling capabilities. Therefore, $h_m$ is not a raw signal, but a deep representation that already encapsulates complex high-order cross-modal dependencies. On this basis, the final convex combination (Eq. 10) is designed as a dynamic gating mechanism. **This design aims to maintain the stability of the optimization process and does not pose a risk of underfitting nonlinear relationships.**

---

> ### Author Response · Authors · 2025-11-27
> **Responses to gTuV [4/5]**
>
> ## Response to Questions:
>
> >**Q1:** Could you provide an explanation for the performance of HiTNet and w/o $\mathcal{L}_{ubl}$ in Table 3 in terms of acc-7 and acc-5?
>
> **R1:** Thank you for your detailed observation of the experimental data. We acknowledge that the variant w/o $\mathcal{L} _{ubl}$ achieves slightly higher Acc-7 and Acc-5 scores on MOSI.
>
> We attribute this phenomenon to a trade-off between fine-grained specialization and robust generalization, driven by the regularization nature of $\mathcal{L}_{ubl}$.
>
> 1. **The core design intent of $\mathcal{L}_{ubl}$ is to prevent over-reliance on a few sub-networks and promote balanced usage by minimizing the coefficient of variation.**
>
> 	**Without $\mathcal{L}_{ubl}$:** The sparse activation network tends to converge to a state of unbalanced load, where the gating mechanism consistently assigns high weights to a small subset of dominant sub-networks.
> 	These few sub-networks greedily optimize for the specific fine-grained intensity distributions (Acc-7/5) of the MOSI dataset, effectively over-specializing to local dataset biases. This yields a marginal gain in fine-grained metrics on this specific dataset but limits the diversity of learned features.
>
> 	**With $\mathcal{L}_{ubl}$:** The loss acts as a strong regularization term that constrains the relative dispersion of sub-network usage. Although this enforcement of balance leads to a slight drop in Acc-7, it ensures that all sub-networks are utilized effectively, enabling the learning of more representative features that yield superior performance on key metrics like Acc-2 and F1.
>
> 2. **Evidence of Generalization:** Crucially, on the SIMS dataset, the full model consistently outperforms the w/o $\mathcal{L}_{ubl}$ variant across all metrics.
>
> This further confirms that the higher Acc-7 of the variant on MOSI is a result of over-specialization to specific sub-networks, whereas retaining $\mathcal{L}_{ubl}$ ensures the model possesses stronger generalization capabilities.
>
>
> >**Q2:** Why couple the intra- and inter-modal streams only via a single late gate—did you evaluate earlier or multi-level fusion (e.g., MoE or cross-layer gating)?
>
> **R2:** Thank you very much for raising this highly constructive question. We address this inquiry from three perspectives:
>
> 1. **First, regarding the coupling of the Intra- and Inter-modal streams.** As detailed in our response to Weakness 3, the actual coupling between the Intra- and Inter-modal streams is not achieved through a simple gate, but via the Cross-Transformer ($E^C$) within the final Hierarchical Fusion Module. This implies that the two streams undergo dense, non-linear integration through Multi-Head Cross-Attention. This mechanism achieves fine-grained, content-based dynamic alignment, effectively capturing the deep non-linear dependencies between intra-modal fine-grained features and inter-modal consistency features.
>
> 2. **Second, our parallel-then-fuse strategy is a deliberate design choice aimed at decoupling specificity and consistency.** The Intra-modal stream is designed to mine modality specificity, utilizing residual information within the modality for self-completion. The Inter-modal stream aims to capture inter-modal consistency, utilizing cross-modal cues to recover missing information. Coupling earlier or via cross-layer interactions would lead to premature information exchange, resulting in the loss of modality-specific information. Maintaining parallel processing forces the network to learn these two sets of complementary feature representations, thereby achieving more robust prediction results during the final Cross-Transformer fusion.
>
> 3. **Additionally, we indeed evaluated early fusion strategies during our preliminary research.**  Specifically, we designed an early fusion variant, where intra-modal features extracted by the sparse activation network were directly injected into the input stage of the Cross-modal Completion Module for early interaction. Experimental results (as shown in the **Table R3**) indicate that the performance of this strategy is inferior to our current late fusion architecture. We attribute this to the fact that cross-layer interaction leads to the loss of modality-specific information, weakening the model's ability to simultaneously capture both unique and shared features.
>
> **Table R3:** Performance comparison between the early fusion variant and the proposed HiTNet on the MOSI dataset. Note: The lower MAE corresponds to superior results.
>
> | Method | Acc-7 | Acc-5 | Acc-2 | F1 | MAE | Corr |
> | :--- | :---: | :---: | :---: | :---: | :---: | :---: |
> | HiTNet (Early Fusion) | 34.65 | 38.74 | 73.55 / 71.97 | 73.98 / 72.45 | 1.058 | 0.524 |
> | **HiTNet** | **35.26** | **39.22** | **74.12 / 72.66** | **74.53 / 73.10** | **1.043** | **0.539** |

---

> ### Author Response · Authors · 2025-11-27
> **Responses to gTuV [5/5]**
>
> >**Q3:** Does using a single prompt token for each modality in cross-modal completion methods result in insufficient expressive power?
>
> **R3:** We appreciate your raising this critical question. We would like to respectfully clarify a misunderstanding, because we inadvertently used the singular term "prompt token." We apologize for this confusion.
>
> **(1)** As formally defined in Eq. (9) of the paper, the prompt is denoted as $h^0_{m} \in \mathbb{R}^{T_m \times D_m}$, where $T_m$ explicitly represents the sequence length of the target modality.
>
> **(2)** Since $h^0_m$ possesses the same temporal structure ($T_m$) as the original input, it consists of $T_m$ independent tokens, not just one.  Each of these $T_m$ tokens serves as an independent Query within the Transformer encoder.  This allows the model to perform fine-grained, token-to-token cross-modal aggregation across the entire timeline.
>
> **(3)** This design explicitly avoids compressing complex multimodal information into a single bottleneck vector.  Instead, it maintains full temporal resolution, ensuring sufficient expressive power to model complex temporal dynamics and non-linear relationships.
>
> We have corrected the terminology in the revised manuscript to remove this ambiguity.
>
>
> >**Q4:** How do you handle the $O(T^2)$ complexity of attention mechanisms in long sequence scenarios (>1-5 minutes)?
>
> **R4:** We thank the reviewer for raising the concern regarding computational complexity in long-sequence scenarios. We acknowledge that the $O(T^2)$ complexity of standard attention can be a bottleneck. However, within our proposed method, this is addressed through a combination of feature decoupling and fixed-length constraints:
>
> 1. **Decoupling Computational Cost from Physical Duration.** HiTNet operates on semantic feature sequences, not raw video frames. The computational complexity is determined by the sampling strategy rather than the video's absolute duration. For long-sequence scenarios, standard preprocessing pipelines employ temporal pooling or keyframe extraction to abstract the raw signal into a feature sequence of controllable length. This effectively decouples the physical duration of the video from the number of tokens processed by the Transformer, ensuring the complexity remains computationally efficient.
>
> 2. **Bounded Complexity via Sequence Standardization.** Specifically, in our implementation, we adopt a fixed-length standardization strategy. Regardless of whether the original video is 10 seconds or 1 minute, the extracted feature sequence is unified to a pre-defined length $T$ (via padding or pooling) before entering the Transformer. This ensures that the computational complexity is strictly bounded by $O(T^2)$, guaranteeing stable memory usage and predictable inference speed regardless of the raw input length.
>
> 3. **Task Scope and Scalability.** Although our experimental validation centers on Utterance-level MSA benchmarks (e.g., MOSI, MOSEI) characterized by shorter clips, this focus does not imply a limitation to short-form content. By leveraging the adaptive sampling and pooling strategies discussed earlier, HiTNet is designed to scale to long-sequence scenarios while maintaining computational efficiency.

---

### Official Review · Reviewer_navA · 2025-11-04

**Soundness:** 3
**Presentation:** 2
**Contribution:** 3
**Rating:** 6
**Confidence:** 4

**Summary:**

The paper addresses multimodal sentiment analysis under simultaneous random frame-level missing cues across modalities. It introduces HiTNet, a hippocampal–thalamic architecture with an intra-modal semantic memory that retrieves and updates residual signals via sparse activation for reconstruction, and an inter-modal regulation path that estimates modality confidence and performs confidence-aware cross-modal completion with learnable prompts, followed by hierarchical fusion. Experiments on standard benchmarks indicate consistent gains across missing rates and strong robustness even under extreme sparsity, with ablations showing that each component contributes materially. Overall, the problem is timely, the design is biologically inspired yet technically grounded, and the evidence suggests practical value.

**Strengths:**

1. The paper addresses a meaningful and insufficiently studied problem in multimodal sentiment analysis under random frame-level missing data, showing clear motivation and novelty.
2. The proposed HiTNet framework is well designed and coherent, combining hippocampal-inspired intra-modal memory with thalamic-inspired confidence-aware cross-modal completion to enhance robustness and information utilization.
3. The experimental evaluation is thorough and convincing, demonstrating consistent improvements across datasets and strong stability under severe missing conditions, with ablation results supporting each component’s effectiveness.

**Weaknesses:**

1. The biological inspiration, while interesting, remains mostly metaphorical; the paper could better justify how the hippocampal–thalamic analogy concretely guides the architecture design and contributes beyond naming.
2. The experimental section, though broad, lacks sufficient comparison with very recent multimodal robustness approaches or large foundation models, which limits understanding of its relative performance in the current landscape.
3. Some implementation details and hyperparameter settings are not clearly described, making it difficult to reproduce results or fully assess the method’s computational efficiency and scalability.

**Questions:**

1. How sensitive is the model’s performance to the design of the semantic memory module and the sparse activation mechanism? Would alternative memory retrieval or selection strategies yield similar robustness?
2. How well does HiTNet generalize to other multimodal tasks beyond sentiment analysis, such as emotion recognition or multimodal dialogue, especially when the missing patterns differ from those in the benchmarks?

---

> ### Author Response · Authors · 2025-11-27
> **Responses to navA [1/7]**
>
> Thank you for your constructive and insightful comments.  We truly appreciate your recognition of the motivation, novelty, and coherence of the proposed HiTNet framework, and your acknowledgment of the thorough and convincing experimental evaluation.
> Your feedback is highly valuable, particularly in highlighting the need for more rigorous theoretical grounding and broader comparison with state-of-the-art approaches.  We have carefully addressed every point raised with new analyses, experiments, and clarifications.
> Below, we provide our point-by-point responses.
>
> ## Response to Weaknesses:
>
> >**W1:** The biological inspiration, while interesting, remains mostly metaphorical; the paper could better justify how the hippocampal–thalamic analogy concretely guides the architecture design and contributes beyond naming.
>
> **Response to Weakness 1:**
> Thank you for your valuable comments. Following your suggestion, we have strengthened our justification from both **theoretical** and **experimental** perspectives.
>
> **1. Theoretical Grounding:** First, we have added references to foundational computational neuroscience models in the introduction to strengthen the theoretical basis. We explicitly map the hippocampal function to Sparse Distributed Memory (SDM) [1] and Hopfield Networks [2], clarifying that our architecture is a mathematical instantiation of these biological associative memory mechanisms.
>
> **2. Experimental Verification:** In addition, we have conducted two targeted experiments on the MOSI dataset to justify how the hippocampal–thalamic analogy concretely guides the architecture design.
>
> - **(1) Quantitative Verification of the Hippocampus-Inspired Module.** The Semantic Memory Module (SSM) simulates hippocampal memory retrieval via a key–value retrieval mechanism. To visually demonstrate this functionality, we visualized the frame-level retrieval attention matrix within the SSM. This matrix shows, for each frame, the most relevant memory units activated by the model. As shown in **Figure 11** (in Appendix B.13), the model still activates memory units for missing frames. More importantly, in the vertical direction, the memory activated by the missing frame is similar to that activated by the surrounding normal frames. **This continuity in semantic trajectories demonstrates that the SSM successfully utilizes temporal context to retrieve the correct memory prototypes to fill in gaps, thereby verifying that the SSM effectively simulates hippocampal memory retrieval.**
>
> - **(2) Quantitative Verification of the Thalamus-Inspired Module.** Our Confidence-Perception Module (CPM) simulates thalamic perceptual regulation by assigning different weights to each modality. To visually demonstrate that the CPM implements differential weighting, we analyzed the weights assigned to each modality on the test set. The results show that the model assigns different weights to the visual, audio, and language modalities (0.74, 0.71, and 0.81, respectively). Furthermore, to verify the effectiveness of this regulation, we conducted an ablation experiment in which all modality weights ($s_m$) were forcibly set to 1, as shown in **Table R1**. The performance degradation of this variant demonstrates the effectiveness of assigning different weights to each modality. **Both the observed weight differences and the ablation results verify that CPM effectively simulated the perceptual regulatory function of the thalamus.**
>
> We have revised the manuscript to incorporate these details into the Introduction and **Appendix B.13**.
>
> References:
>
> [1]Pentti Kanerva. Sparse Distributed Memory. MIT Press, Cambridge, MA, USA, 1988. ISBN 0262111322.
>
> [2] J J Hopfield. Neural networks and physical systems with emergent collective computational abilities. Proceedings of the National Academy of Sciences, 79(8):2554–2558, 1982.   doi: 10.1073/pnas. 79.8.2554
>
> **Table R1:** Effectiveness evaluation of the Thalamus-Inspired Module. Note: The lower MAE corresponds to superior results.
>
> | Method | Acc-7 | Acc-5 | Acc-2 | F1 | MAE | Corr |
> | :--- | :---: | :---: | :---: | :---: | :---: | :---: |
> | HiTNet ($s_m=1$) | 34.87 | 38.57 | 73.72 / 72.08 | 74.48 / 72.51 | 1.044 | 0.531 |
> | **HiTNet** | **35.26** | **39.22** | **74.12 / 72.66** | **74.53 / 73.10** | **1.043** | **0.539** |

---

> > ### Comment · Reviewer_navA · 2025-11-27
> > **Reviewer response**
> >
> > Good paper and good response, and I will keep my rating.

---

> ### Author Response · Authors · 2025-11-27
> **Responses to navA [2/7]**
>
> >**W2:** The experimental section, though broad, lacks sufficient comparison with very recent multimodal robustness approaches or large foundation models, which limits understanding of its relative performance in the current landscape.
>
> **Response to Weakness 2:** We appreciate your suggestion, which helped strengthen the experimental section.   We agree that including more recent multimodal robustness approaches is important for positioning our method in the current landscape. **In this revision, we have incorporated the latest frontier baseline model of ACL in 2025 to provide more representative and competitive comparative experiments.** The relevant performance results are presented in **Tables R2, R3, and R4**. The updated results demonstrate HiTNet's superiority even against these strong recent baselines:
>
> - **On MOSI (Table R2):** HiTNet achieves SOTA performance across almost all metrics, significantly outperforming P-RMF in Acc-2 ( HiTNet: 74.12%, P-RMF: 72.81%) and F1 (HiTNet: 74.53%, P-RMF: 72.93%).
>
> - **On MOSEI (Table R3):** HiTNet maintains a clear lead in classification accuracy (Acc-7: 47.19%, Acc-5: 47.98%), demonstrating that our hierarchical fusion strategy is particularly effective for precise sentiment categorization. While P-RMF shows competitive results in MAE, HiTNet remains the top performer in the primary accuracy metrics.
>
> - **On SIMS (Table R4):** HiTNet continues to lead in Acc-5 (35.62%) and Acc-3 (59.28%), validating its robustness on datasets with high modality variance.
> **These additional comparisons further validate the robustness and effectiveness of our model.**
> We have carefully revised the relevant subsection in **Section 4.4** to incorporate your suggestion.
>
> **Table R2:** Comparison of model performance on MOSI. Note: The lower MAE corresponds to superior results.
>
> | Method     | Acc-7     | Acc-5     | Acc-2             | F1                | MAE       | Corr      |
> | ---------- | --------- | --------- | ----------------- | ----------------- | --------- | --------- |
> | MISA       | 29.85     | 33.08     | 71.49 / 70.33     | 71.28 / 70.00     | 1.085     | 0.524     |
> | Self-MM    | 29.55     | 34.67     | 70.51 / 69.26     | 66.60 / 67.54     | 1.070     | 0.512     |
> | MMIM       | 31.30     | 33.77     | 69.14 / 67.06     | 66.65 / 64.04     | 1.077     | 0.507     |
> | CENET      | 30.38     | 37.25     | 71.46 / 67.73     | 68.41 / 64.85     | 1.080     | 0.504     |
> | TETFN      | 30.30     | 34.34     | 69.76 / 67.68     | 65.69 / 63.29     | 1.087     | 0.507     |
> | TFR-Net    | 29.54     | 34.67     | 68.15 / 66.35     | 61.73 / 60.06     | 1.200     | 0.459     |
> | ALMT       | 30.30     | 33.42     | 70.40 / 68.39     | 72.57 / 71.80     | 1.083     | 0.498     |
> | LNLN       | 34.26     | 38.27     | 72.55 / 70.94     | 72.73 / 71.25     | 1.046     | 0.527     |
> | P-RMF      | 34.19     | 38.50     | 72.81 / 71.53     | 72.93 / 71.69     | **1.038** | 0.525     |
> | **HiTNet** | **35.26** | **39.22** | **74.12 / 72.66** | **74.53 / 73.10** | 1.043     | **0.539** |
>
> **Table R3:** Comparison of model performance on MOSEI. Note: The lower MAE corresponds to superior results.
>
> | Method     | Acc-7     | Acc-5     | Acc-2             | F1                | MAE       | Corr      |
> | ---------- | --------- | --------- | ----------------- | ----------------- | --------- | --------- |
> | MISA       | 40.84     | 39.39     | 71.27 / 75.82     | 63.85 / 68.73     | 0.780     | 0.503     |
> | Self-MM    | 44.70     | 45.38     | 73.89 / 77.42     | 68.92 / 72.31     | 0.695     | 0.498     |
> | MMIM       | 40.75     | 41.74     | 73.32 / 75.89     | 68.72 / 70.32     | 0.739     | 0.489     |
> | CENET      | 47.18     | 47.83     | 74.67 / 77.34     | 70.68 / 74.08     | 0.685     | 0.535     |
> | TETFN      | 30.30     | 47.70     | 69.76 / 67.68     | 65.69 / 63.29     | 1.087     | 0.508     |
> | TFR-Net    | 46.83     | 34.67     | 73.62 / 77.23     | 68.80 / 71.99     | 0.697     | 0.489     |
> | ALMT       | 40.92     | 41.64     | 76.64 / 77.54     | 77.14 / 78.03     | 0.674     | 0.481     |
> | LNLN       | 45.42     | 46.17     | 76.30 / 78.19     | 77.77 / 79.95     | 0.692     | 0.530     |
> | P-RMF      | 44.63     | 45.87     | 78.14 / 78.83     | **79.33**  / 80.39| **0.658** | 0.589     |
> | **HiTNet** | **47.19** | **47.98** | **78.29 / 79.28** | 78.84 / **81.46** | 0.665     | **0.591** |

---

> ### Author Response · Authors · 2025-11-27
> **Responses to navA [3/7]**
>
> >This form follows the response to **Weaknesse 3** in the previous window.
>
> **Table R4:** Comparison of model performance on SIMS. Note: The lower MAE corresponds to superior results.
>
> | Method     | Acc-5     | Acc-3     | Acc-2     | F1        | MAE       | Corr      |
> | ---------- | --------- | --------- | --------- | --------- | --------- | --------- |
> | MISA       | 31.53     | 56.87     | 72.71     | 66.30     | 0.539     | 0.348     |
> | Self-MM    | 32.28     | 56.75     | 72.81     | 68.43     | 0.508     | 0.376     |
> | MMIM       | 31.81     | 52.76     | 69.86     | 66.21     | 0.544     | 0.339     |
> | CENET      | 22.29     | 53.17     | 68.13     | 57.90     | 0.589     | 0.107     |
> | TETFN      | 33.42     | 56.91     | 73.58     | 68.67     | 0.505     | 0.387     |
> | TFR-Net    | 26.52     | 52.89     | 68.13     | 58.70     | 0.661     | 0.169     |
> | ALMT       | 20.00     | 45.36     | 69.66     | 72.76     | 0.561     | 0.364     |
> | LNLN       | 34.64     | 57.14     | 72.73     | **79.43** | 0.514     | 0.397     |
> | P-RMF      | 34.83     | 54.75     | 73.64     | 74.65     | **0.500** | **0.414** |
> | **HiTNet** | **35.62** | **59.28** | **73.99** | 77.33     | 0.504     | 0.389     |
>
>
> >**W3:** Some implementation details and hyperparameter settings are not clearly described, making it difficult to reproduce results or fully assess the method’s computational efficiency and scalability.
>
> **Response to Weakness 3(1/3):** Thank you for pointing out the need for clearer implementation details and hyperparameter specifications. To address this, we have expanded the implementation section and added comprehensive analyses on **Reproducibility** and **Computational Efficiency**.
>
> **Part 1: Elaboration on Implementation and Reproducibility Analysis.**
>
> **1.1 First, to verify the model's sensitivity to the $k$ value within the sparse activation mechanism, we fixed the total number of sub-networks at $n=5$ and retrained the model with varying $k$ values.** The results on the MOSI dataset are presented in **Table R5**.
> - **Optimal Sparsity ($k=3$):** The model performance gradually improves as $k$ increases, peaking at $k=3$. This suggests that a moderate increase in the number of activated sub-networks helps integrate multi-perspective intra-modal semantic information, thereby enhancing robustness.
> - **Consequence of Full Activation ($k=5$):** However, when $k$ increases to 5, the performance declines. This demonstrates that the full activation strategy introduces a substantial amount of redundant information irrelevant to the current sample, diminishing the advantages of sparse selection. The reproducibility of the model was improved through the sensitivity analysis of k.
> We have revised the paper and added the above experiments to **Appendix B.2.**
>
> **Table R5:** Sensitivity analysis of the Top-$k$ hyperparameter in the sparse activation mechanism on the MOSI dataset. Note: The lower MAE corresponds to superior results.
>
> | Method        | Acc-7     | Acc-5     | Acc-2             | F1                | MAE       | Corr      |
> | ------------- | --------- | --------- | ----------------- | ----------------- | --------- | --------- |
> | n=k=1         | 34.88     | 38.85     | 73.22 / 72.15     | 73.71 / 72.55     | 1.054     | 0.531     |
> | n=5,k=1       | 34.03     | 37.42     | 73.25 / 72.09     | 73.63 / 72.28     | 1.067     | 0.537     |
> | n=5,k=2       | 34.73     | 38.68     | 73.50 / 72.25     | 73.84 / 72.65     | 1.050     | 0.538     |
> | **n=5,k=3**   | **35.26** | **39.22** | **74.12 / 72.66** | **74.53 / 73.10** | **1.043** | 0.539     |
> | n=5,k=4       | 33.72     | 37.53     | 73.58 / 72.11     | 73.97 / 72.85     | 1.057     | **0.541** |
> | n=k=5         | 33.22     | 37.36     | 73.10 / 72.16     | 73.25 / 72.24     | 1.067     | 0.536     |

---

> ### Author Response · Authors · 2025-11-27
> **Responses to navA [4/7]**
>
> **Response to Weakness 3(2/3):**
>
> **1.2 Second, Automatic Loss Weighting Mechanism.** To enhance the robust reproducibility of the model, we introduced an automatic weighting mechanism based on homoscedastic uncertainty[1]. Unlike the manual setting where each loss term requires exhaustive search for an appropriate coefficient, this mechanism treats the weight of each auxiliary loss as a learnable parameter. During training, the model adaptively estimates the relative uncertainty associated with each auxiliary objective. Loss terms with higher uncertainty are automatically down-weighted, while more reliable ones receive stronger emphasis. **This allows the optimization process to dynamically balance the contribution of the main loss and the auxiliary regularization losses, resulting in a more stable and self-adjusting training procedure without extensive hyperparameter tuning.**
>
> We compared the performance of the best manually tuned configurations against the proposed automatic weighting strategy across all missing rate settings. As shown in **Table R6, Table R7**, and **Table R8**, the quantitative results demonstrate that the automatic weighting strategy consistently yields performance that is superior to, or competitive with, the manually tuned baselines. **Our automatic weighting approach not only eliminates the need for human intervention in hyperparameter search but also enhances the reproducibility of model results.**
>
> We have revised the paper and added the above experiments to **Appendix B.1.**
>
> Reference:
>
> [1] Kendall, A., et al. "Multi-task learning using uncertainty to weigh losses for scene geometry and semantics." CVPR 2018.
>
> **Table R6:** Performance comparison between Manual Tuning and Automatic Weighting strategies on the MOSI dataset. Note: The lower MAE corresponds to superior results.
>
> |Method|Acc-7|Acc-5|Acc-2|F1|MAE|Corr|
> |---|---|---|---|---|---|---|
> |**HiTNet (Manual)**|**35.26**|39.22|74.12 / 72.66|74.53 / 73.10|1.043|**0.539**|
> |**HiTNet (Auto)**|35.02|**39.24**|**75.73 / 74.15**|**75.93 / 74.31**|**1.037**|0.537|
>
> **Table R7:** Performance comparison between Manual Tuning and Automatic Weighting strategies on the MOSEI dataset. Note: The lower MAE corresponds to superior results.
>
> | Method              | Acc-7     | Acc-5     | Acc-2             | F1                | MAE       | Corr  |
> | ------------------- | --------- | --------- | ----------------- | ----------------- | --------- | ----- |
> | **HiTNet (Manual)** | **47.19** | **47.98** | 78.29 / **79.28** | 78.84 / **81.46** | 0.665     | 0.591 |
> | **HiTNet (Auto)**   | 47.15     | 47.91     | **79.39** / 78.88 | **79.99**/ 79.80  | **0.654** | 0.593 |
>
> **Table R8:** Performance comparison between Manual Tuning and Automatic Weighting strategies on the SIMS dataset. Note: The lower MAE corresponds to superior results.
>
> | Method              | Acc-5     | Acc-3     | Acc-2     | F1    | MAE   | Corr  |
> | ------------------- | --------- | --------- | --------- | ----- | ----- | ----- |
> | **HiTNet (Manual)** | **35.62** | **59.28** | 73.99     |**77.33** | **0.504** | **0.389**|
> | **HiTNet (Auto)**   | 34.84     | 59.23     | **74.13** | 77.26 | 0.511 | 0.387 |

---

> ### Author Response · Authors · 2025-11-27
> **Responses to navA [5/7]**
>
> **Response to Weakness 3(3/3):**
>
> **Part 2: Assessment of Computational Efficiency and Scalability.**
>
> To conduct an analysis of computational efficiency and scalability, we conducted a comparative analysis of inference efficiency for the baseline model LNLN, the full HiTNet model (k=3), HiTNet with the sparse activation set to the minimum/maximum values (k=1, k=5), and HiTNet without the semantic memory module (w/o SMM), and we report key metrics including parameter count, FLOPs, GPU memory, and latency. The results are summarized in **Table R9**.
>
> **2.1 The full HiTNet model (k=3) introduces only a modest increase in overhead compared to the baseline model.** In terms of resource consumption, HiTNet's total parameter count (93.43M) and total FLOPs (627.45G) are on par with those of LNLN (91.99M / 622.91G), with only a slight increase, indicating that **our model does not introduce significant additional computational burden.**
>
> Secondly, in terms of deployment feasibility, HiTNet's peak GPU memory consumption is only 1.846GB, nearly identical to the baseline model, **demonstrating its excellent resource efficiency and deployment feasibility on standard hardware.**
>
> Finally, regarding the performance-latency trade-off, HiTNet’s average inference latency (62.72 ms) is slightly higher than LNLN's (46.00 ms). This additional overhead arises from the semantic memory module and sparse activation network in HiTNet, which enable more complex memory retrieval and fine-grained semantic modeling. **This moderate increase in inference latency is a justified trade-off, as it comes with significant performance improvements and enhanced robustness when handling the challenging scenario of severe frame-level missingness.**
>
> **2.2 Lightweight design of memory retrieval and sparse activation.** Although sparse activation introduces additional computations, the resulting increase in inference latency is moderate and still well within real-time processing constraints.
> Specifically, the inference latency only increases by 0.24 ms when scaling the activation sparsity from the minimum (k=1, 62.65 ms) to the maximum (k=5, 62.89 ms).
>
> In addition, comparing the full HiTNet (with k=3) and the HiTNet without the Semantic Memory Module (w/o SMM), we observe only a minor difference in both parameter count and FLOPs. **This confirms the lightweight nature and low overhead of the memory retrieval mechanism itself.
> These findings demonstrate that both the memory retrieval and sparse activation mechanisms are lightweight and efficient.**
>
> **2.3 Potential for scalability to larger datasets.** The design principles of HiTNet inherently support robust scalability. **Our modules ensure stable scalability to longer sequences and larger multimodal datasets** by selectively retrieving only the most relevant memories and applying a sparse activation mechanism.
>
> **In summary, this analysis validates that HiTNet maintains strong efficiency and exhibits excellent scalability while delivering substantial performance gains.**
>
> We have revised the paper and added the above experiments to **Appendix B.9.**
>
> **Table R9:** Comparative analysis of model inference efficiency.
>
> | Evaluation Metric | LNLN   | HiTNet (k=1) | HiTNet (k=3) | HiTNet (k=5) | HiTNet (w/o SMM) |
> |------------------|--------|---------------|---------------|---------------|------------------|
> | Params (M)        | 91.99 | 93.43         | 93.43         | 93.43         | 93.38            |
> | FLOPs (G)         | 622.91| 627.35        | 627.45        | 627.55        | 627.43           |
> | GPU Memory (GB)   | 1.840 | 1.846         | 1.846         | 1.846         | 1.845            |
> | Latency (ms)      | 46.00 | 62.65         | 62.72         | 62.89         | 61.59            |

---

> ### Author Response · Authors · 2025-11-27
> **Responses to navA [6/7]**
>
> ## Response to Questions:
>
> >**Q1:** How sensitive is the model’s performance to the design of the semantic memory module and the ..val or selection strategies yield similar robustness?
>
> **R1:** Thank you very much for your affirmation of our methods and for your constructive suggestions. To verify the model's sensitivity to the Semantic Memory Module (SMM) and sparse activation networks, we designed multiple verification experiments.
>
> **Part 1:** **Verification of the model's sensitivity to the semantic memory module.**
>
> 1. **Ablation Study:** We first performed an ablation study by removing the SMM. The results in **Table R10** show a performance decline, confirming the effectiveness of the SMM.
>
> 2. **Parameter Sensitivity:** Secondly, in order to verify the sensitivity of the Model to the number of Memory units, our original text already includes the experiment of Effect of Memory Size on Model Performance, as shown in **Appendix B.5.** Among the evaluated settings, a memory size of 64 yields superior F1 scores across a range of missing rates, suggesting an optimal trade-off between representational capacity and regularization.
>
> 3. **Visualization:** In addition, to verify the activation of semantic memory more intuitively，we visualized the frame-level retrieval attention matrix within the SSM. This matrix shows, for each frame, the most relevant memory units activated by the model. As shown in **Figure 9** in appendix B.10, the model still activates memory units for missing frames. More importantly, in the vertical direction, the memory activated by the missing frame is similar to that activated by the surrounding normal frames. **This continuity in semantic trajectories demonstrates that the SSM successfully utilizes temporal context to retrieve the correct memory prototypes to fill in gaps**.
>
> **Part 2:** **Verification of the model's sensitivity to selection strategies.**
>
> - **Structural Sensitivity:** We designed two sets of comparative experiments. First, we replaced the entire sparse activation network module with **a single sub-network** ($n=k=1$) to verify the impact of the sparse activation network on model performance.
>
> - **Strategy Sensitivity:** In addition, to verify the model's sensitivity to the $k$ value within the sparse activation mechanism, we fixed the total number of sub-networks at $n=5$ and retrained the model with varying $k$ values.
>
> We conducted experiments across all missing rates and calculated the average performance. The results on the MOSI dataset are presented in **Table** **R11**.
>
> 1. When the sparse activation network degenerates into a single sub-network ($n=k=1$), the performance significantly declines. **This indicates that multiple sub-networks provide diverse semantic enhancement pathways, which are crucial for capturing richer intra-modal feature patterns.**
>
> 2. Furthermore, with $n=5$ fixed, the model performance gradually improves as $k$ increases, peaking at $k=3$. **This suggests that a moderate increase in the number of activated sub-networks helps integrate multi-perspective intra-modal semantic information, thereby enhancing robustness.**
>
> 3. **However, when $k$ increases to 5 (full activation)**, the performance declines. This demonstrates that the full activation strategy introduces a substantial amount of redundant information irrelevant to the current sample, diminishing the advantages of sparse selection. **Overall, these findings demonstrate that an appropriately chosen k is crucial for achieving optimal intra-modal enhancement.**
>
> **Table R10:** Ablation study on the semantic memory module. Note: The lower MAE corresponds to superior results.
>
> | Method | Acc-7 | Acc-5 | Acc-2 | F1 | MAE | Corr
> | :--- | :---: | :---: | :---: | :---: | :---: | :---: |
> | HiTNet (w/o SMM) | 34.74| 38.63 | 73.61 / 72.27 | 74.01 / 72.40 | 1.043 | 0.532 |
> | **HiTNet** | **35.26** | **39.22** | **74.12 / 72.66** | **74.53 / 73.10** | **1.043** | **0.539** |
>
> **Table R11:** Sensitivity analysis of the Top-$k$ hyperparameter in the sparse activation mechanism on the MOSI dataset. Note: The lower MAE corresponds to superior results.
>
> | Method        | Acc-7     | Acc-5     | Acc-2             | F1                | MAE       | Corr      |
> | ------------- | --------- | --------- | ----------------- | ----------------- | --------- | --------- |
> | n=k=1         | 34.88     | 38.85     | 73.22 / 72.15     | 73.71 / 72.55     | 1.054     | 0.531     |
> | n=5,k=1       | 34.03     | 37.42     | 73.25 / 72.09     | 73.63 / 72.28     | 1.067     | 0.537     |
> | n=5,k=2       | 34.73     | 38.68     | 73.50 / 72.25     | 73.84 / 72.65     | 1.050     | 0.538     |
> | **n=5,k=3**   | **35.26** | **39.22** | **74.12 / 72.66** | **74.53 / 73.10** | **1.043** | 0.539     |
> | n=5,k=4       | 33.72     | 37.53     | 73.58 / 72.11     | 73.97 / 72.85     | 1.057     | **0.541** |
> | n=k=5         | 33.22     | 37.36     | 73.10 / 72.16     | 73.25 / 72.24     | 1.067     | 0.536     |

---

> ### Author Response · Authors · 2025-11-27
> **Responses to navA [7/7]**
>
> >**Q2:** How well does HiTNet generalize to other multimodal tasks beyond sentiment analysis, such as emotion recognition or multimodal dialogue, especially when the missing patterns differ from those in the benchmarks?
>
> **R2:** Thank you for raising this forward-looking question. We address the generalization ability of HiTNet from the following three perspectives.
>
> 1. **The universality of the model architecture.** Although this work focuses on multimodal sentiment analysis (MSA), the core architecture of HiTNet is task-independent.
>
> 	It is worth noting that our intra-modal enhancement and inter-modal regulation streams operate purely on multimodal feature sequences, independent of task-specific output semantics (regression or classification).  Consequently, the proposed HiTNet is inherently task-agnostic regarding the prediction head. **It can be adapted to tasks such as emotion recognition (IEMOCAP) or multimodal dialogue modeling by simply replacing the prediction head with a softmax layer.**
>
> 2. **Robustness against different missing patterns.**
> 	To encourage generalization, our training strategy incorporates dynamic random masking with varying missing ratios. As a result, the model is exposed to a broad spectrum of perturbation patterns during training.  Additionally, we have already evaluated HiTNet under multiple missing scenarios: **Modality-level missingness**,  **Random frame-level missingness**, and **Consecutive frame-level missingness**. Across all these settings, HiTNet consistently achieves strong results, demonstrating that its robustness transfers across missing patterns.
>
> 3. **Extension to Non-traditional Modalities and Real-world Noisy Signals.**
> 	Because HiTNet operates on generic feature vector sequences, it can naturally extend to non-traditional or sensor-based multimodal tasks. Such modalities often experience high noise or signal dropout, where HiTNet’s dual-stream completion mechanism (**memory-guided enhancement + sparse activation regulation**) is particularly advantageous.
>
> **In summary,** HiTNet is not restricted to sentiment analysis. Its task-agnostic design, robustness to diverse missing patterns, and ability to operate on arbitrary sequential modalities **make** **it highly extensible to emotion recognition, multimodal dialogue, and other multimodal understanding tasks**, even when the missing patterns deviate from benchmark settings.

---

### Official Review · Reviewer_NAMa · 2025-11-04

**Soundness:** 3
**Presentation:** 3
**Contribution:** 3
**Rating:** 6
**Confidence:** 2

**Summary:**

Inspired by findings in neuroscience, this paper proposes a multimodal sentiment analysis model named HiTNet, designed for scenarios with missing modalities. The model architecture is motivated by two functional mechanisms in the brain: the hippocampus, which is responsible for semantic memory retrieval and pattern completion, and the thalamus, which regulates perceptual integration and confidence control. Specifically: 1. The hippocampus performs semantic memory retrieval and pattern completion; 2. The thalamus dynamically integrates multimodal information and regulates reliability among modalities. The proposed HiTNet consists of two parallel subnetworks: an intra-modal enhancement stream and an inter-modal regulation stream. Experimental results demonstrate that HiTNet achieves superior performance on multiple datasets (MOSI, MOSEI, and SIMS), maintaining high accuracy even under conditions of severe modality missing.

**Strengths:**

1. The paper introduces the hippocampal–thalamic mechanism from neuroscience into multimodal sentiment analysis under modality-missing conditions. The idea is novel and demonstrates strong originality.

2. Experiments on three benchmark datasets validate the effectiveness of the proposed method. Comprehensive ablation studies are conducted for each module to verify their contributions, along with analyses on missing rates and loss weight settings, showing the experimental evaluation is thorough.

3. Compared with baseline models, the proposed method consistently outperforms existing approaches across all three datasets and under various missing ratios.

4. The paper is clearly written and well-structured, making it easy to follow.

**Weaknesses:**

1. The paper states that “the missing information reconstruction module ERec, designed to reconstruct the missing features of each modality,” but in Figure 2, the position of this module is ambiguous, making it difficult for readers to interpret and align it with the overall framework.

2. The paper does not explore the model’s performance under non-random missing scenarios.

3. There are some writing detail issues: the formulas lack proper punctuation — for example, a comma should follow Formula (2) and a period should follow Formula (3). Formula (7) should be followed by an explanatory sentence, and the use of uppercase and lowercase letters is inconsistent.

**Questions:**

1. In Tables 5 -7, only a few combinations of loss weights (α, β, γ) are tested (e.g., 15, 0.5, 0.1 and 10, 0.9, 0.1). Could you elaborate on how these specific values were chosen? Were they selected empirically, or do they correspond to particular design motivations (e.g., emphasizing one auxiliary loss over another)?

2. In the hierarchical fusion stage, the language modality is always placed last. Is the model sensitive to the order of modality fusion?

---

> ### Author Response · Authors · 2025-11-27
> **Responses to NAMa [1/4]**
>
> Thank you for your fruitful feedback and for recognizing the originality, clarity, and cost-efficiency of our proposed method. We sincerely appreciate your positive evaluation. We would like to provide a point-to-point response to address your concerns as follows.
>
> ## Response to Weaknesses:
> >**W1:** The paper states that... align it with the overall framework.
>
> **R1:** We sincerely appreciate this valuable feedback regarding the visualization of our framework. We acknowledge that the previous layout of Figure 2 did not clearly illustrate the specific position and data flow of the missing information reconstruction module ($E^{Rec}$), potentially causing confusion. Following your suggestion, we have redesigned Figure 2 to explicitly demarcate the position of $E^{Rec}$.
>
>
>
> >**W2:** The paper does not explore the model’s performance under non-random missing scenarios.
>
> **R2:** Thank you for your valuable suggestion. We address your concern from two aspects: **Consecutive frame-level missingness** and **Modality-level missingness**.
>
> 1. **Consecutive frame-level missingness experiment.**
> 	In strict accordance with your suggestion, we implemented a consecutive frame-level missingness experiment on the MOSI dataset to mimic the non-random missing scenarios you mentioned. Specifically, we varied the missing rate from 0.0 to 0.9 to cover a wide range of severity. To validate the model's performance, we designed experiments from two aspects: comparison with the baseline model and analysis of the effectiveness of the confidence perception module. **Table R1** reports the average results of different missing rates for the baseline LNLN, HiTNet without CPM (w/o CPM), and the full HiTNet model.
>
> 	- **(1) The data demonstrates that HiTNet maintains superior performance compared to the baseline even under these challenging conditions.** HiTNet consistently outperforms the baseline LNLN across all evaluation metrics. For example, in terms of Acc-7, HiTNet achieves 33.70\%, significantly surpassing the baseline's 31.76\%. Our model also achieves a lower Mean Absolute Error (MAE) of 1.066 compared to the baseline's 1.085.
>
> 	- **(2) To further validate the model's internal mechanism in this realistic setting, we analyzed the contribution of the Confidence Perception Module (CPM).** The ablation results show that the CPM is crucial for handling consecutive frame drops. When the CPM is removed (w/o CPM), we observe a notable performance drop, particularly in the Correlation (Corr) metric, which falls to a level comparable to the baseline (0.506). In contrast, the full model achieves a significantly higher correlation of 0.525. This evidence confirms that the CPM is essential for gauging input reliability and mitigating the impact of unreliable frames, thereby ensuring robust performance in consecutive frame-level missingness scenarios.
>
> 2. **Modality-level missingness experiment.**
> 	Modal-level missingness refers to the complete absence of a single modality, which aligns with a non-random missing scenario. It is worth noting that experiments under modal-level missingness have already been included in Section 4.8 of the original text. The experimental results are shown in **Table R2**. The results demonstrate that HiTNet outperforms or is comparable to baseline methods in most cases. For instance, in the {V} (Visual only) setting, HiTNet achieves 59.33\% Acc-2, outperforming the baseline LNLN (49.03\%) by over 10 absolute percentage points.
>
> Overall, the experiment simulates non-random missing scenarios, further demonstrating HiTNet's robustness and effectiveness in practical applications.
>
> We have revised the paper and added the above experiments to **Appendix B.12**.
>
> **Table R1:** Analysis of model performance under consecutive frame-level missingness. Note: The lower MAE corresponds to superior results.
>
> |Method|Acc-7|Acc-5|Acc-2|F1|MAE|Corr|
> |---|---|---|---|---|---|---|
> |LNLN|31.76|35.56|71.72 / 70.36|72.27 / 70.44|1.085|0.506|
> |HiTNet (w/o CPM)|33.22|36.89|72.06 / 70.51|72.32 / 70.65|1.071|0.505|
> |**HiTNet**|**33.70**|**37.94**|**72.26 / 71.08**|**72.95 / 71.08**|**1.066**|**0.525**|
>
> **Table R2:** Acc-2 under modality-level missing conditions on the MOSI dataset. Note: The modalities inside the `{}` are the present modalities.
>
> | Method      | {V}       | {A}       | {L}       | {V,A}     | {V,L}     | {A,L}     |
> | ----------- | --------- | --------- | --------- | --------- | --------- | --------- |
> | MMIM        | 48.20     | 48.64     | 81.29     | 49.61     | 81.15     | 81.83     |
> | CENET       | 51.85     | 51.80     | 81.54     | 51.80     | 81.54     | 81.49     |
> | TETFN       | 55.25     | 55.25     | 81.05     | 55.25     | 81.00     | 81.10     |
> | ALMT        | 54.96     | 55.10     | 79.83     | 55.05     | 79.98     | 79.98     |
> | LNLN        | 49.03     | 49.03     | 82.48     | 49.03     | 82.21     | **82.26** |
> | **HiTNet**  | **59.33** | **59.29** | **82.49** | **59.04** | **82.26** | 81.90     |

---

> ### Author Response · Authors · 2025-11-27
> **Responses to NAMa [2/4]**
>
> >**W3:** There are some writing detail issues: the formulas lack proper punctuation — for example, a comma should follow Formula (2) and a period should follow Formula (3). Formula (7) should be followed by an explanatory sentence, and the use of uppercase and lowercase letters is inconsistent.
>
> **R3:** Thank you for pointing out these writing and formatting issues.
>
> We have carefully revised the manuscript according to your suggestions. In the Method section, we have added the appropriate punctuation after Formula (2) and Formula (3), included an explanatory sentence following Formula (7), and corrected the inconsistencies in uppercase and lowercase usage throughout the formulas. In addition, we performed a thorough check of the entire Method section and corrected other similar formatting inconsistencies to further improve clarity and readability.
>
>
> ## Response to Questions:
>
> >**Q1:** In Tables 5 -7, only a few combinations of loss weights (α, β, γ) are tested (e.g., 15, 0.5, 0.1 and 10, 0.9, 0.1). Could you elaborate on how these specific values were chosen? Were they selected empirically, or do they correspond to particular design motivations (e.g., emphasizing one auxiliary loss over another)?
>
> **Response to Question 1(1/2):**
> Thank you for your question. We answer your question from two perspectives: parameter selection and supplementary experiments on automatic parameter adjustment.
> 1. **When setting the weights $(\alpha, \beta, \gamma)$ in the total loss function, we primarily considered the semantics of each loss and the characteristics of the dataset.**
> 	Combinations such as $\left(10, 0.9, 0.1\right)$ and $\left(15, 0.5, 0.1\right)$ were derived based on the functional role of the losses and verified through extensive experimentation. For example, the utilization balance loss ($\mathcal{L}_{\text{ubl}}$), corresponding to $\alpha$, aims to enforce an even distribution of network load within the sparse activation network. However, its loss gradient is typically numerically much smaller than that of the main sentiment prediction loss. Therefore, we assign a relatively high weight to it, preventing its signal from being dominated by the optimization of the main loss.
>
> 	**Furthermore, the specific numerical differences in the parameter values across different datasets are caused by the unique characteristics of those datasets.** For datasets with fewer training samples or higher data noise, the model is more prone to overfitting. In such cases, we typically increase the weight of $\alpha$ moderately to employ a stronger regularization signal, thereby enhancing the model's robustness and generalization ability. This fine-grained weight adjustment ensures that the model can achieve optimal performance across datasets of varying scales and quality.

---

> ### Author Response · Authors · 2025-11-27
> **Responses to NAMa [3/4]**
>
> **Response to Question 1(2/2):**
>
> 2. **In addition, in order to enhance the reproducibility of the model's performance and its generalization, we introduced an automatic weighting mechanism based on homoscedastic uncertainty[1].**
>
> Unlike the manual setting, where each loss term requires exhaustive search for an appropriate coefficient, this mechanism treats the weight of each auxiliary loss as a learnable parameter. During training, the model adaptively estimates the relative uncertainty associated with each auxiliary objective. Loss terms with higher uncertainty are automatically down-weighted, while more reliable ones receive stronger emphasis. This allows the optimization process to dynamically balance the contribution of the main loss and the auxiliary regularization losses, resulting in a more stable and self-adjusting training procedure without extensive hyperparameter tuning.
>
> We compared the performance of the best manually tuned configurations against the proposed automatic weighting strategy across all missing rate settings.
>
> As shown in **Table R3, Table R4,** and **Table R5**, the quantitative results demonstrate that **the automatic weighting strategy consistently yields performance that is superior to, or competitive with, the manually tuned baselines.**
>
> Our automatic weighting approach not only eliminates the need for human intervention in hyperparameter search but also enhances the reproducibility of model results.
>
> **In summary, we have addressed the concern regarding hyperparameter selection from both a theoretical perspective (Gradient Magnitude Balancing) and a methodological perspective (Automatic Weighting).** This dual approach ensures that HiTNet's performance is not derived from arbitrary tuning or overfitting to specific values, but stems from a robust architectural design. By introducing the automatic weighting mechanism, we have further guaranteed the model's reproducibility and generalization across diverse benchmarks.
>
> We have revised the paper and added the above-mentioned experiments and analyses to **Appendix B.1**.
>
> Reference:
>
> [1] Kendall, A., et al. "Multi-task learning using uncertainty to weigh losses for scene geometry and semantics." CVPR 2018.
>
> **Table R3:** Performance comparison between Manual Tuning and Automatic Weighting strategies on the MOSI dataset. Note: The lower MAE corresponds to superior results.
>
> |Method|Acc-7|Acc-5|Acc-2|F1|MAE|Corr|
> |---|---|---|---|---|---|---|
> |**HiTNet (Manual)**|**35.26**|39.22|74.12 / 72.66|74.53 / 73.10|1.043|**0.539**|
> |**HiTNet (Auto)**|35.02|**39.24**|**75.73 / 74.15**|**75.93 / 74.31**|**1.037**|0.537|
>
> **Table R4:** Performance comparison between Manual Tuning and Automatic Weighting strategies on the MOSEI dataset. Note: The lower MAE corresponds to superior results.
>
> | Method              | Acc-7     | Acc-5     | Acc-2             | F1                | MAE       | Corr  |
> | ------------------- | --------- | --------- | ----------------- | ----------------- | --------- | ----- |
> | **HiTNet (Manual)** | **47.19** | **47.98** | 78.29 / **79.28** | 78.84 / **81.46** | 0.665     | 0.591 |
> | **HiTNet (Auto)**   | 47.15     | 47.91     | **79.39** / 78.88 | **79.99**/ 79.80  | **0.654** | 0.593 |
>
> **Table R5:** Performance comparison between Manual Tuning and Automatic Weighting strategies on the SIMS dataset. Note: The lower MAE corresponds to superior results.
>
> | Method              | Acc-5     | Acc-3     | Acc-2     | F1    | MAE   | Corr  |
> | ------------------- | --------- | --------- | --------- | ----- | ----- | ----- |
> | **HiTNet (Manual)** | **35.62** | **59.28** | 73.99     |**77.33** | **0.504** | **0.389**|
> | **HiTNet (Auto)**   | 34.84     | 59.23     | **74.13** | 77.26 | 0.511 | 0.387 |

---

> ### Author Response · Authors · 2025-11-27
> **Responses to NAMa [4/4]**
>
> >**Q2:** In the hierarchical fusion stage, the language modality is always placed last. Is the model sensitive to the order of modality fusion?
>
> **R2:** Thank you for your valuable suggestion.
> 1. **Following your advice, We evaluated the sensitivity of HiTNet to the fusion order of modalities.** Specifically, we assessed all six possible fusion orders of the visual (V), audio (A), and language (L) intra-modal features on the MOSI dataset. For a fair comparison, we conducted experiments for all missing rates for each fusion order and report the average results, as shown in **Table R6**. The results indicate that placing the language modality last in the hierarchical fusion stage yields the best performance. This is because the language modality provides crucial semantic cues for sentiment recognition, and placing it in the later stage helps to better integrate complementary information from the visual and audio modalities. **These findings demonstrate that the model is indeed sensitive to the order of modality fusion and justify the design choice of placing the language modality at the final fusion stage. However, the performance differences across different fusion orders are relatively modest, suggesting that while the order matters, the model remains fairly robust to reasonable variations**.
>
> 2. **Additionally, we replaced the hierarchical fusion module with three common simple fusion methods:** summation (Sum), concatenation (Concat), and attention-based fusion (Attention), and retrained the model to further validate the necessity and effectiveness of hierarchical fusion from a methodological perspective. As shown in **Table R6**, the performance of these variant models drops significantly, indicating that simple fusion methods cannot adequately capture the hierarchical dependencies between modalities. In contrast, **hierarchical fusion more effectively facilitates the interaction of intra-modal and inter-modal complementary features, thereby improving model robustness and performance.**
>
> In summary, these experiments provide strong support for our architectural choices. The superior performance of placing the language modality at the final stage confirms its role in providing key semantic cues. Furthermore, the significant performance gap between our hierarchical fusion module and simple fusion methods validates the necessity of our design for capturing complex cross-modal interactions.
>
> We have revised the paper and added the above experiments to **Appendix B.6.**
>
> **Table R6:** Performance comparison of different multimodal fusion strategies. Note: The lower MAE corresponds to superior results.
>
> | Method | Acc-7 | Acc-5 | Acc-2 | F1 | MAE | Corr |
> |--------|-------|--------|---------|---------|---------|---------|
> | LAV | 34.03 | 37.55 | 73.73 / 72.41 | 74.08 / 72.74 | 1.048 | 0.536 |
> | ALV | 34.09 | 38.17 | 73.25 / 72.20 | 73.64 / 72.32 | 1.058 | **0.546** |
> | LVA | 34.56 | 38.41 | 73.64 / 72.74 | 74.11 / 72.44 | 1.048 | 0.545 |
> | VLA | 34.69 | 38.52 | 73.55 / 72.16 | 73.75 / 72.36 | 1.044 | 0.542 |
> | AVL | 34.94 | 38.85 | 74.05 / 72.52 | 74.19 / 72.91 | 1.050 | 0.540 |
> | sum | 32.73 | 36.84 | 72.39 / 71.22 | 73.92 / 71.25 | 1.067 | 0.519 |
> | concat | 32.76 | 36.45 | 72.99 / 71.81 | 73.60 / 71.94 | 1.085 | 0.510 |
> | attention | 33.55 | 37.50 | 72.71 / 71.37 | 73.26 / 71.87 | 1.073 | 0.527 |
> | **HiTNet (VAL)** | **35.26** | **39.22** | **74.12 / 72.66** | **74.53 / 73.10** | **1.043** | 0.539 |

---

### Official Review · Reviewer_vvAP · 2025-11-10

**Soundness:** 2
**Presentation:** 3
**Contribution:** 2
**Rating:** 2
**Confidence:** 3

**Summary:**

The paper proposes HiTNet, a dual-stream network inspired by hippocampal memory retrieval and thalamic perceptual regulation for multimodal sentiment analysis under severe frame-level missingness. The “hippocampal” stream uses a key-value semantic memory module plus sparse activation sub-networks to reconstruct intra-modal features; the “thalamic” stream estimates per-modality confidence and performs adaptive cross-modal completion. A hierarchical Cross-Transformer fuses both streams and an auxiliary reconstruction loss is added.

**Strengths:**

- Originality: First work that explicitly models hippocampal pattern-completion and thalamic gating for frame-level missing data; combines key-value memory, sparse routing, and confidence-weighted cross-modal attention in a unified framework.

- Quality: Each component is formally described, ablated, and visualised.

- Clarity: Well-written; neuroscientific motivation is intuitive; notation is consistent.

- Significance: Addresses a practical and under-studied scenario (random frame-level corruption across all modalities) and demonstrates remarkable robustness at extreme missing rates that prior reconstruction or co-learning methods cannot reach.

**Weaknesses:**

- Novelty gap with existing memory-based completion: Key-value memory banks have been used for missing-modality imputation and for speech emotion with artefacts. The authors should clarify how their SMM differs from these works beyond simply being applied to sentiment.

- Biological inspiration is loose: The hippocampus performs associative pattern completion across time, whereas the SMM retrieves a single best-matching vector with cosine similarity and updates by frequency, which is closer to a standard dictionary. A more rigorous mapping or citation to computational neuroscience models (e.g., Hopfield networks, Kanerva’s sparse distributed memory) would strengthen the claim.

- Inference requires five sub-networks per modality, a memory bank of 64×3 tensors, two Transformers for confidence & completion, and a 4-layer fusion transformer. The paper reports accuracy but not latency, FLOPs, or GPU memory; a table or plot (e.g., vs. LNLN) is needed for deployment claims.

- Hyper-parameter sensitivity: Optimal loss weights differ by dataset (MOSI α=10, MOSEI α=1.5). No automatic tuning or principled weighting (e.g., uncertainty weighting, GradNorm) is explored, raising reproducibility concerns.

**Questions:**

1. Can you show one concrete example (frames + spectrogram + text) where intra-modal completion visually/audibly recovers lost content?

2. What happens under non-random missing patterns—e.g., consecutive video frames dropped due to buffering, or audio muted for the last 2 s? Does confidence prediction still help?

---

> ### Author Response · Authors · 2025-11-27
> **Responses to vvAP [1/5]**
>
> We sincerely thank the reviewer for the thorough and insightful evaluation of our work. We greatly appreciate the recognition of its originality, clarity, and significance, as well as the acknowledgment of our neuroscientific motivation and extensive ablations.  Thank you for your suggestions on our article. In response, we have carefully addressed each point with new analyses, experiments, and clarifications.  Below, we provide our point-by-point responses.
> ## Response to Weaknesses:
> >**W1:** Novelty gap with existing memory-based completion: Key-value memory banks have been used for missing-modality imputation and for speech emotion with artefacts. The authors should clarify how their SMM differs from these works beyond simply being applied to sentiment.
>
> **R1:** We sincerely appreciate your insight regarding the context of existing memory-based banks. While Key-Value (K-V) memory banks have indeed been utilized for missing modality imputation, our Semantic Memory Module (SMM) differs in its mechanism design. Existing K-V memory networks typically rely on the assumption of high-quality query vectors. However, under frame-level missingness, the query vector itself is inevitably corrupted. This causes standard lookup mechanisms to retrieve semantically irrelevant memory slots, thereby amplifying noise rather than restoring information. To overcome this, our SMM introduces two key architectural innovations:
>
> **(1) Adaptive Integration via Residual Gating.** Unlike prior works that directly replace or concatenate missing inputs with retrieved values, as shown in Eq. (3) of the paper, we introduce a Residual Gating Mechanism ($g_m$).
>
> This gate dynamically determines the integration weight by modeling the non-linear interaction (via Sigmoid activation) between the original noisy input $x_m$ and the retrieved memory content $v_{i^*}^m$. Crucially, when a corrupted query leads to inaccurate retrieval, $g_m$ effectively suppresses the integration of erroneous memory, ensuring robustness against noise propagation during feature enhancement.
>
> **(2) Two-stage Guidance-Refinement Architecture.** In our framework, the memory module serves as a semantic structural prior rather than the final imputation endpoint. The actual non-linear restoration is performed by the subsequent Sparse Activation Network (SAN). This decoupling, where SMM provides a semantic prior and SAN performs sparse refinement, enables the model to capture complex intra-modal characteristics, significantly outperforming simple retrieve-and-replace strategies.
>
> We have incorporated this discussion into the **Related Work** section of the revised manuscript to clarify our novelty.

---

> ### Author Response · Authors · 2025-11-27
> **Responses to vvAP [2/5]**
>
> >**W2:** Biological inspiration is loose: The hippocampus performs associative pattern completion across time, whereas the SMM retrieves a single best-matching vector with cosine similarity and updates by frequency, which is closer to a standard dictionary. A more rigorous mapping or citation to computational neuroscience models (e.g., Hopfield networks, Kanerva’s sparse distributed memory) would strengthen the claim.
>
> **R2:** Thank you for your valuable and constructive comments.
> 1. Following your suggestion, we have explicitly analyzed and cited these foundational works [1][2] in the revised manuscript to strengthen our theoretical claims. Specifically, we clarified the rigorous mapping between our module and these models.
>
> 	- **Retrieval:** Our cosine-based retrieval is a mathematical instantiation of Sparse Distributed Memory (SDM)’s content-based addressing [1].
>
> 	- **Completion:** The recovery of missing frame-level data aligns with the associative pattern completion dynamics of Hopfield Networks [2], where the system evolves partial cues into stable prototypes.
>
> 2. To further substantiate this theoretical mapping, we conducted a supplementary analysis to demonstrate that our model indeed performs the memory retrieval functions predicted by these biological and computational neuroscience models. To visually validate the retrieval mechanism of the Semantic Memory Module (SMM), we plotted the frame-level retrieval attention matrix, using a representative sample from the MOSI dataset.
> 	- As shown in **Figure 11 (Appendix B.13)**, the model successfully activates relevant memory units even for missing frames.
> 	- **Crucially, in the vertical direction, the memory activated by the missing frame is similar to that activated by the surrounding normal frames.** This continuity in semantic trajectories demonstrates that the SMM effectively utilizes temporal context to reconstruct correct prototypes for gaps, verifying its associative memory capability.
>
> We have incorporated these theoretical clarifications and experimental visualizations into the revised **Introduction** and **Appendix B.13**, providing a rigorous computational neuroscience grounding for our architecture.
>
> References:
>
> [1]Pentti Kanerva. Sparse Distributed Memory. MIT Press, Cambridge, MA, USA, 1988. ISBN 0262111322.
>
> [2] J J Hopfield. Neural networks and physical systems with emergent collective computational abilities. Proceedings of the National Academy of Sciences, 79(8):2554–2558, 1982.   doi: 10.1073/pnas. 79.8.2554

---

> ### Author Response · Authors · 2025-11-27
> **Responses to vvAP [3/5]**
>
> >**W3:** Inference requires five sub-networks per modality, a memory bank of 64×3 tensors, two Transformers for confidence & completion, and a 4-layer fusion transformer. The paper reports accuracy but not latency, FLOPs, or GPU memory; a table or plot (e.g., vs. LNLN) is needed for deployment claims.
>
> **R3:** We sincerely appreciate your valuable feedback regarding the complexity and deployment feasibility of our HiTNet model. In response, we conducted a comparative analysis of inference efficiency for the baseline model LNLN, the full HiTNet model (k=3), HiTNet with the sparse activation set to the minimum/maximum values (k=1/k=5), and HiTNet without the semantic memory module (w/o SMM), and we report key metrics including parameter count, FLOPs, GPU memory, and latency. The results are summarized in Table R1.
>
> 1. **The full HiTNet model (k=3) introduces only a modest increase in overhead compared to the baseline model.**
> 	- In terms of resource consumption, HiTNet's total parameter count (93.43M) and total FLOPs (627.45G) are on par with those of LNLN (91.99M / 622.91G), with only a slight increase, indicating that **our model does not introduce significant additional computational burden.**
> 	- Secondly, in terms of deployment feasibility, HiTNet's peak GPU memory consumption is only 1.846GB, nearly identical to the baseline model, **demonstrating its excellent resource efficiency and deployment feasibility on standard hardware.**
> 	- Finally, regarding the performance-latency trade-off, HiTNet’s average inference latency (62.72 ms) is slightly higher than LNLN's (46.00 ms). This additional overhead arises from the semantic memory module and sparse activation network in HiTNet, which enable more complex memory retrieval and fine-grained semantic modeling.
> 	- **This moderate increase in inference latency is a justified trade-off, as it comes with significant performance improvements and enhanced robustness when handling the challenging scenario of severe frame-level missingness**
>
> 2. **Lightweight design of memory retrieval and sparse activation.** Although sparse activation introduces additional computations, the resulting increase in inference latency is moderate and still well within real-time processing constraints. Specifically, the inference latency only increases by 0.24 ms when scaling the activation sparsity from the minimum ($k=1$, 62.65 ms) to the maximum ($k=5$, 62.89 ms).
> 	In addition, comparing the full HiTNet (with $k=3$) and the HiTNet without the Semantic Memory Module (w/o SMM), we observe only a minor difference in both parameter count and FLOPs. **This confirms the lightweight nature and low overhead of the memory retrieval mechanism itself.**
> 	**These findings demonstrate that both the memory retrieval and sparse activation mechanisms are lightweight and efficient.**
>
> 3. **Potential for scalability to larger datasets.** The design principles of HiTNet inherently support robust scalability. **Our modules ensure stable scalability to longer sequences and larger multimodal datasets by selectively retrieving only the most relevant memories and applying a sparse activation mechanism.**
>
> In summary, this analysis validates that HiTNet maintains strong efficiency and exhibits excellent scalability while delivering substantial performance gains.
>
> We have revised the paper and added these details to **Appendix B.9**.
>
> **Table R1:** Comparative analysis of model inference efficiency.
>
> | Evaluation Metric | LNLN   | HiTNet ($k=1$) | HiTNet ($k=3$) | HiTNet ($k=5$) | HiTNet (w/o SMM) |
> |------------------|--------|---------------|---------------|---------------|------------------|
> | Params (M)        | 91.99 | 93.43         | 93.43         | 93.43         | 93.38            |
> | FLOPs (G)         | 622.91| 627.35        | 627.45        | 627.55        | 627.43           |
> | GPU Memory (GB)   | 1.840 | 1.846         | 1.846         | 1.846         | 1.845            |
> | Latency (ms)      | 46.00 | 62.65         | 62.72         | 62.89         | 61.59            |

---

> ### Author Response · Authors · 2025-11-27
> **Responses to vvAP [4/5]**
>
> >**W4:** Hyper-parameter sensitivity: Optimal loss weights differ by dataset (MOSI α=10, MOSEI α=1.5). No automatic tuning or principled weighting (e.g., uncertainty weighting, GradNorm) is explored, raising reproducibility concerns.
>
> **R4:** We sincerely thank you for the constructive suggestion regarding hyperparameter sensitivity and reproducibility. We have fully adopted your recommendation and implemented an automatic weighting mechanism based on homoscedastic uncertainty [1].
>
> Unlike the manual setting, where each loss term requires exhaustive search for an appropriate coefficient, this mechanism treats the weight of each auxiliary loss as a learnable parameter. During training, the model adaptively estimates the relative uncertainty associated with each optimization objective. Loss terms with higher uncertainty are automatically down-weighted, while more reliable ones receive stronger emphasis. This allows the optimization process to dynamically balance the contribution of the main loss and the auxiliary regularization losses, resulting in a more stable and self-adjusting training procedure without extensive hyperparameter tuning.
>
> - We conducted extensive re-experiments across MOSI, MOSEI, and SIMS datasets under all missing rate settings. Tables R2, R3 and R4 present the comparison between the manual tuning and the proposed automatic weighting strategies.
> - The quantitative results demonstrate that the proposed automatic weighting strategy yields performance that is superior to, or highly competitive with, the manually tuned baselines across all metrics.
> - This confirms that our approach not only resolves the hyper-parameter sensitivity concern but also further unlocks the model's potential by automatically balancing the training objectives, ensuring robust reproducibility across diverse datasets without human intervention.
>
> We have revised the paper and added the above-mentioned experiments and analyses to **Appendix B.1**.
>
> Reference:
> [1] Kendall, A., et al. "Multi-task learning using uncertainty to weigh losses for scene geometry and semantics." CVPR 2018.
>
> **Table R2:** Performance comparison between Manual Tuning and Automatic Weighting strategies on the MOSI dataset. Note: The lower MAE corresponds to superior results.
>
> |Method|Acc-7|Acc-5|Acc-2|F1|MAE|Corr|
> |---|---|---|--------|--------|---|---|
> |**HiTNet (Manual)**|**35.26**|39.22|74.12 / 72.66|74.53 / 73.10|1.043|**0.539**|
> |**HiTNet (Auto)**|35.02|**39.24**|**75.73 / 74.15**|**75.93 / 74.31**|**1.037**|0.537|
>
> **Table R3:** Performance comparison between Manual Tuning and Automatic Weighting strategies on the MOSEI dataset. Note: The lower MAE corresponds to superior results.
>
> | Method              | Acc-7     | Acc-5     | Acc-2             | F1                | MAE       | Corr  |
> | ------------------- | --------- | --------- | ----------------- | ----------------- | --------- | ----- |
> | **HiTNet (Manual)** | **47.19** | **47.98** | 78.29 / **79.28** | 78.84 / **81.46** | 0.665     | 0.591 |
> | **HiTNet (Auto)**   | 47.15     | 47.91     | **79.39** / 78.88 | **79.99**/ 79.80  | **0.654** | 0.593 |
>
> **Table R4** Performance comparison between Manual Tuning and Automatic Weighting strategies on the SIMS dataset. Note: The lower MAE corresponds to superior results.
>
> | Method              | Acc-5     | Acc-3     | Acc-2     | F1    | MAE   | Corr  |
> | ------------------- | --------- | --------- | -------------- | ---------- | ----- | ----- |
> | **HiTNet (Manual)** | **35.62** | **59.28** | 73.99 |**77.33** | **0.504** | **0.389**|
> | **HiTNet (Auto)**   | 34.84 | 59.23 | **74.13** | 77.26 | 0.511 | 0.387 |

---

> ### Author Response · Authors · 2025-11-27
> **Responses to vvAP [5/5]**
>
> ## Response to Questions:
>
> >**Q1:** Can you show one concrete example (frames + spectrogram + text) where intra-modal completion visually/audibly recovers lost content?
>
> **R1:** We sincerely appreciate this exceptionally insightful suggestion. Regarding the request for raw content reconstruction (frames/spectrograms/text), we would like to clarify that HiTNet is a discriminative model designed for high-level semantic representation learning, rather than a generative model aimed at pixel- or waveform-level reconstruction.
>
> Reconstructing raw pixels requires a decoder trained with pixel-level supervision, which is not only computationally expensive but also unnecessary for the sentiment classification task. Our goal is to recover the underlying semantic patterns in the latent feature space, which directly contribute to the downstream classification performance.
>
> **To intuitively demonstrate the model's intra-modal completion capability, we performed a rigorous qualitative analysis in the feature space, taking the visual modality as a representative example.**
>
> - As shown **in Figure 9 in Appendix B.10**, we visualize the feature sequence heatmap for a sample with a 0.2 missing rate, showing the missing visual modality features (a), the intra-modal completed features (b), and the original complete features (c), to demonstrate that our model achieves semantic recovery in the feature space.
> - The deep colored stripes in (a) clearly show the masked tokens, representing the loss of visual information.
> - (b) shows that after processing by the Intra-modal Enhancement Stream, these originally empty stripes are effectively filled.
> - Furthermore, compared to the original complete features in (c), (b) displays a highly consistent structural pattern.
>
> **This visually proves that the Intra-modal Enhancement Stream successfully completes the lost underlying semantic patterns, thereby significantly restoring the data's semantic integrity within the feature space.**
>
> We have revised the paper and added the above-mentioned experiments and analyses to **Appendix B.10**.
>
>
>
>
>
>
> >**Q2:** What happens under non-random missing patterns—e.g., consecutive video frames dropped due to buffering, or audio muted for the last 2 s? Does confidence prediction still help?
>
> **R2:** Thank you for highlighting the importance of realistic, non-random missing patterns. To address this and explicitly evaluate whether our confidence prediction mechanism remains effective under such conditions, we conducted additional experiments on the MOSI dataset simulating consecutive frame-level missingness (mimicking scenarios like video buffering or segment loss). We varied the consecutive missing rate from 0.0 to 0.9 and compared the baseline (LNLN), HiTNet without the Confidence Perception Module (w/o CPM), and the full HiTNet. The aggregated results (averaged across 10 independent runs) are presented in **Table R5**.
>
> The experimental results show that the confidence prediction mechanism is highly effective and crucial under consecutive missing patterns.
>
> 1. **Robustness under Consecutive Missingness.** As shown in the table, the full HiTNet consistently outperforms the baseline LNLN across all missing rates. For instance, HiTNet achieves an Acc-7 of 33.70% compared to LNLN's 31.76%, and reduces the MAE from 1.085 to 1.066. This demonstrates that our model handles non-random, continuous data loss much better than the baseline.
>
> 2. **The Critical Role of Confidence Prediction.** The ablation study explicitly confirms the contribution of the Confidence Perception Module (CPM). When the CPM is removed (HiTNet w/o CPM), we observe a notable performance drop. This is most evident in the Correlation (Corr) metric, which reflects the model's ability to track sentiment trends. Without CPM, the correlation drops to 0.505, falling to a level comparable to the baseline (0.506). This indicates that without confidence gating, complex models are prone to be misled by continuous noise blocks. With the CPM enabled, the correlation is restored to 0.525.
>
> These results confirm that under non-random missing patterns, the confidence prediction mechanism effectively identifies unreliable consecutive segments. It prevents the model from being misled by corrupted data, thereby maintaining robust performance.
>
> We have included these detailed results and analysis in **Appendix B.12**.
>
> **Table R5:** Analysis of model performance under consecutive frame-level missingness. Note: The lower MAE corresponds to superior results.
>
> |Method|Acc-7|Acc-5|Acc-2|F1|MAE|Corr|
> |---|---|---|---|---|---|---|
> |LNLN|31.76|35.56|71.72 / 70.36|72.27 / 70.44|1.085|0.506|
> |HiTNet (w/o CPM)|33.22|36.89|72.06 / 70.51|72.32 / 70.65|1.071|0.505|
> |**HiTNet**|**33.70**|**37.94**|**72.26 / 71.08**|**72.95 / 71.08**|**1.066**|**0.525**|

---

> > ### Comment · Reviewer_vvAP · 2025-11-27
> > **Response to rebuttal.**
> >
> > Thank you for the authors' reply. As my concerns have largely been clarified, I have updated the evaluation.
> >
> > Best,
> >
> > Reviewer vvAP

---

> > > ### Author Response · Authors · 2025-11-28
> > >
> > > Thank you for your message.
> > >
> > > We are very pleased that our clarifications have successfully addressed your concerns, and we sincerely appreciate your taking the time to update your evaluation.
> > >
> > > All the best!

---

### Public Comment · ~Moby_James1 · 2025-11-28
**Clarification regarding Model Selection and Metric Reporting**

Thank you for your work. I have read the paper with great interest and appreciate the authors' novel approach in proposing a bio-inspired dual-stream architecture.
However, after reviewing the source code (specifically **core/utils.py**), I have a concern regarding the evaluation protocol used for Table 2 and Table 8. The **get_best_results function** appears to save **independent model checkpoints** for different metrics (e.g., separate weights for optimal Acc-2 vs. MAE).
**My Question:** Could the authors clarify if the 8 metrics reported under a specific condition (e.g., missing rate r=0.2 in Table 8) are derived from **a single fixed model checkpoint**, or are they an **aggregation of best scores from different checkpoints** saved at different epochs during training and testing?
If the results are aggregated from different checkpoints, you represent a "performance upper bound envelope" rather than the performance of a single deployable model. To ensure fair comparison with baselines, I respectfully suggest you reporting all metrics based on a single checkpoint selected via the validation set.
Thank you for your clarification.

---

> ### Author Response · Authors · 2025-11-29
>
> Thank you very much for your appreciation of our work and for your question.
> 1. It is true that all reported metrics are derived from independent checkpoints, but this method is necessary within the context of our research. **This method has been adopted by excellent published papers, that is, the latest baselines compared in our paper (such as LNLN[1] and P-RMF[2]).**
>
> 2. **Alignment with SOTA Evaluation Standards.** We wish to clarify that **our evaluation methodology is completely consistent with that of all baselines, with no modifications made**. We adopted this multi-metric independent checkpoint strategy **to align with the baselines' evaluation protocol** for a fair comparison of our model's performance.
> 	The reasons for adopting this strategy are as follows:
> 	- **Inherent Conflict of Metrics:** Regression metrics (such as MAE and Corr) and classification metrics (such as Acc-2 and F1) place fundamentally different requirements on the model. Due to the distinct focus of regression and classification metrics on different aspects of model performance, a model achieving the lowest error on regression metrics may not necessarily exhibit optimal performance on classification metrics [1].
> 	- To comprehensively and fairly demonstrate the **maximum potential** and **robustness limit** of the model (including our proposed HiTNet and all baselines) when dealing with incomplete data, we must allow it to reach its peak performance on every metric dimension. Reporting only a single checkpoint would lead to suboptimal performance on metrics that conflict with the optimization objective, thereby underestimating its true potential.
> 3. **Guarantee of Fair Comparison.**
> 	- We agree that fair comparison is paramount. **The results for all baseline models and our proposed HiTNet were obtained under the same testing strategy**.
>
>     - The paper rigorously tests all metrics on the test set and reports the results, strictly adhering to the established practices for evaluating noisy data in the field of robust MSA.
>
> Thank you again for your comment! We believe this explanation addresses your concerns regarding the fairness of the evaluation.
>
> References:
>
> [1]Haoyu Zhang, Wenbin Wang, and Tianshu Yu. Towards robust multimodal sentiment analysis with incomplete data. In The Thirty-eighth Annual Conference on Neural Information Processing Systems, 2024a.
>
> [2]Aoqiang Zhu, Min Hu, Xiaohua Wang, Jiaoyun Yang, Yiming Tang, and Ning An. Proxy-driven robust multimodal sentiment analysis with incomplete data. In Wanxiang Che, Joyce Nabende, Ekaterina Shutova, and Mohammad Taher Pilehvar (eds.), Proceedings of the 63rd Annual Meeting of the Association for Computational Linguistics (Volume 1: Long Papers), pp. 22123–22138, Vienna, Austria, July 2025. Association for Computational Linguistics. ISBN 979-8-89176-251-0.

---

### Author Response · Authors · 2025-11-28
**Summary of Core Strengths, Revisions and Reviewer Feedback**

We sincerely thank all the reviewers for their comprehensive and constructive assessment of our manuscript. We are very pleased to receive the reviewers' recognition of the HiTNet's novel motivation (NAMa, navA), attractiveness (Ek1D), coherent architecture (navA), as well as its SOTA performance and robustness across various missing conditions (vvAP, NAMa, navA, dS1k), and research significance (gTuV).
## I. Summary of Core Strengths Unanimously Recognized by Reviewers

1. **Innovation and Deep Integration of Neuroscience:** This paper is the first to introduce hippocampal pattern completion and thalamic regulation for handling frame-level missing data, demonstrating **strong originality and novelty**.(vvAP, NAMa, navA)

2. **Addressing a Realistic and Challenging Problem:** The work focuses on the more realistic and common scenario of **random frame-level missing data**, rather than just complete modality absence. (vvAP, navA, gTuV)

3. **Outstanding Robustness and SOTA Performance:** The model consistently **outperforms existing SOTA methods** across all datasets and various missing rates. (vvAP, NAMa, navA, dS1k, Ek1D)

4. **Clear and Functionally Decoupled Framework Design:** HiTNet is cleverly designed and coherent, employing a dual-stream mechanism that clearly decouples intra-modal self-completion and inter-modal confidence regulation. (vvAP, navA, gTuV)

5. **Comprehensive and Convincing Experimental Validation:** The paper provides extensive evaluations across multiple datasets, along with thorough ablation studies for all modules. (vvAP, NAMa, navA, dS1k, Ek1D)

6. **High-Quality Writing and Interpretability:** The manuscript is well-structured, clearly written, and easy to follow. Furthermore, the biologically inspired dual-stream design offers an **intuitive and appealing rationale**, enhancing the model's interpretability. (vvAP, NAMa, dS1k, Ek1D)
## II. Summary of Key Revisions Addressing Reviewers' Concerns

We have thoroughly revised the paper to address all concerns raised and have supplemented it with extensive experiments and qualitative analyses.

1. **Biological Inspiration:** We performed a computational neuroscience mapping of our biological analogy. We added new visualization experiments to further support our hippocampal–thalamic analogy.

2. **Robustness to Consecutive Missingness:** We expanded our experiments to different missingness settings, demonstrating that HiTNet maintains superior performance even under consecutive frame-level missingness scenarios.

3. **Baseline Enhancement:** We incorporated the latest frontier baseline models to provide more representative and competitive comparative experiments against the current state-of-the-art.

4. **Sparse Activation Mechanism Sensitivity:** By replacing the sparse network with a single sub-network and varying the sparsity level  $k$ (including full activation), our analysis validates that sparse activation is crucial for maximizing semantic relevance.

5. **Automatic Tuning Strategy:** We implemented an automatic weighting mechanism based on homoscedastic uncertainty, which ensures the reproducibility of model performance by eliminating manual hyperparameter search.

6. **Computational Efficiency:** We reported key metrics (parameter count, FLOPs, GPU memory, and latency) for HiTNet and the baseline model LNLN. This demonstrated HiTNet's deployment feasibility, lightweight nature, and high efficiency.

7. **Convergence & Sensitivity:** We separately plotted individual curves for the total loss and all individual constraint objectives. This demonstrated that the training process is overall stable, converges efficiently, is insensitive to random seeds, and exhibits low loss fluctuation.

8. **Fusion Strategy Necessity:** We added comparisons on fusion order and alternative fusion methods, confirming the necessity of our hierarchical fusion design.

## III. Reviewer Rating Upgrades and Positive Feedback

**Our comprehensive revisions have fully addressed all of the reviewers' concerns.** We are pleased that the reviewers provided positive feedback on our response:

- **Reviewer (vvAP)** indicated that our response resolved their concerns and subsequently **raised the Rating from 2 to 6**.

- **Reviewer (dS1k)** further affirmed the value and potential impact of our work and **considered raising the Rating from 6 to a higher rating (8)**.

- **Reviewers (navA and Ek1D)** fully acknowledged the quality of our paper and response and maintained their high Rating of **6**.

- We also received high praise from other reviewers, including:

    - **Reviewer (Ek1D):** I do think this is a valuable piece of work with clear potential impact.

    - **Reviewer (navA):** Good paper and good response.

We have incorporated all experimental results and revisions into the updated manuscript. We sincerely appreciate the valuable time and insightful feedback provided by all reviewers and the Area Chair.

---

### Meta-Review · Area_Chair_9vry · 2026-01-08

**Summary:**

The paper proposes a model for multimodal sentiment analysis that is inspired by the memory capabilities of the hippocampus and the perceptual regulation capabilities of the thalamus. The method was evaluated on MOSI, MOSEI, and SIMS and demonstrated a higher performance compared to the s.o.t.a. baselines.

An issue concerning evaluation was raised in a public comment entitled "Clarification regarding Model Selection and Metric Reporting", specifically that the reported result is an upper bound on performance. The authors argued they are using the standard evaluation method for this. Given the LNLN and P-RMF papers, it seems that that is indeed the case.

However, as the performance improvement is relatively small, and confidence intervals are not reported, it's difficult to say whether these numbers reflect the expected performance of the method or are overly optimistic, representative of a best-case scenario. (Note: the LNLN and P-RMF papers do not report confidence intervals either.)

The concerns are related to the novelty, connection to neuroscience, differences to past work, use of more recent foundation models, justification for the architectural choices of the model.

**Reviewer Concerns:**

In terms of the revisions following the reviewers feedback, the authors have provided good arguments and follow up experiments for the following aspects: robustness to consecutive missingness, the merits of sparse activation, computational efficiency, convergence, and the fusion strategy.

They have also added some baseline experiments at the request of the reviewers (who asked for the latest foundation models). It looks like the MAE of R-RMF is slightly better than the one of the proposed method, though HiTNet does do slightly better in terms of the other metrics.

I find the biological connection to still be somewhat strained. The authors cite work from the field of neuroscience (Costa et al., for instance), but do not explain their findings sufficiently or relate them to the proposed method (for instance, the Costa et al. paper seems to find that hippocampal gamma activity increases for correctly remembered aversive scenes; this is extremely loosely connected to the method presented here). Also, the new tuning strategy should be applied to all baselines.

Overall, this paper presents a method that has some claims to novelty and advances multimodal sentiment analysis from some perspectives. However, the claims about the connections to neuroscience and not sufficiently supported, the new hyperparameter tuning method should be better described and the authors should consider confidence intervals (or some other notion of robustness) when reporting performance.

**Reviewer Scores:**

I have no real way of knowing how the reviewers would have changed their scores, but I'm sure a generative model could hallucinate something.

---

### Decision · Program_Chairs · 2026-01-26

Reject